# SlideSparse: Fast and Flexible (2N-2):2N Structured Sparsity

Yingbo Hao [* 1 3]   Hanyong Shao [* 2 3]   Ting Song [3]   Yan Xia [3]   Di Zhang [2 3]   Shaohan Huang [3]   Xun Wu [3]
Songchen Xu [3 4]   Le Xu [3 5]   Li Dong [3]   Zewen Chi [3]   Yi Zou [1]   Furu Wei [3]

## Abstract

NVIDIA's 2:4 Sparse Tensor Cores deliver $2\times$ throughput but demand strict 50% pruning—a ratio that causes severe accuracy loss in LLMs. Milder $(2N-2):2N$ patterns (e.g., 6:8, 25% pruning) preserve accuracy far better—within 0.4–1.8 average points of dense in our Qwen2.5-7B/14B study—yet receive *NO* hardware support and fall back to dense execution. We present SLIDESPARSE, the first system to unlock Sparse Tensor Core acceleration for the $(2N-2):2N$ model family on commodity GPUs. Our *Sliding Window Decomposition* rewrites any $(2N-2):2N$ weight block into $N-1$ overlapping 2:4-compliant windows without changing the underlying dot product; in addition, our *Activation Lifting* fuses the corresponding activation rearrangement into per-token quantization at low marginal cost. Integrated into vLLM, SLIDESPARSE is evaluated across various GPUs (A100, H100, B200, RTX 4090, RTX 5080, DGX-spark), precisions (FP4, INT8, FP8, BF16, FP16), and model families (Llama, Qwen, BitNet). On compute-bound workloads, the measured speedup $(1.33\times)$ matches the theoretical upper-bound $N/(N-1) = 4/3$ at 6:8 weight sparsity in Qwen2.5-7B, establishing $(2N-2):2N$ as a practical path to better accuracy–speedup trade-offs in LLM acceleration. Code available at https://github.com/bcacdwk/vllmbench.

## 1. Introduction

NVIDIA's Sparse Tensor Cores deliver $2\times$ throughput for 2:4 structured sparsity (Mishra et al., 2021), but they im-

*Equal contribution  [1]South China University of Technology [2]Peking University [3]Microsoft Research [4]Shanghai Jiao Tong University [5]The Hong Kong University of Science and Technology. Correspondence to: Yi Zou <zouyi@scut.edu.cn>, Furu Wei <fuwei@microsoft.com>.

*Proceedings of the $43^{rd}$ International Conference on Machine Learning*, Seoul, South Korea. PMLR 306, 2026. Copyright 2026 by the author(s).

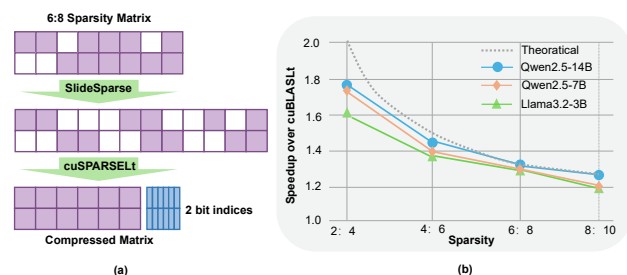

*Figure 1.* **SlideSparse extends 2:4 Sparse Tensor Cores to the $(\mathbf{2N-2}) : \mathbf{2N}$ sparsity family.** (a) SlideSparse transforms 6:8 weights into 2:4-compliant blocks, enabling sparsity acceleration. (b) End-to-end speedup on A100 (INT8, 8K tokens) approaches the theoretical limit $S_{\max} = N/(N-1)$, i.e. $3/2, 4/3, 5/4$ for $N = 3, 4, 5$ (§3).

pose a rigid constraint: 50% of weights must be pruned. For LLMs, this pruning ratio exceeds compression tolerance (Frantar & Alistarh, 2023; Sun et al., 2024)—especially on reasoning tasks where accuracy degrades catastrophically. Researchers thus face a stark choice: sacrifice accuracy for speed, or preserve accuracy with *no acceleration*.

Milder structured sparsity patterns provide a better trade-off. $(2N-2):2N$ patterns, such as 4:6 (33% sparsity) and 6:8 (25%), preserve accuracy far better than 2:4 sparsity. Across Qwen2.5-7B and Qwen2.5-14B (Qwen et al., 2025) with both Wanda (Sun et al., 2024) and SparseGPT (Frantar & Alistarh, 2023), 6:8 stays within 0.4–1.8 average accuracy points of dense, while 2:4 loses 14.9–16.5 points (§2). Yet these patterns receive *zero* hardware acceleration: Sparse Tensor Cores support only the 2:4 format (Mishra et al., 2021), and cuSPARSELt (NVIDIA, 2021) provides no API for alternatives. Consequently, inference engines such as vLLM (Kwon et al., 2023) and TensorRT-LLM (NVIDIA, 2023) must treat these $(2N-2):2N$ sparse weights as dense, wasting the sparsity entirely.

We introduce SLIDESPARSE, a system that accelerates $(2N-2):2N$ sparsity on existing GPUs with mathematically lossless transformation and no hardware modifications. Our key insight: any $(2N-2):2N$ block *decomposes losslessly* into overlapping 2:4-compliant windows via *Sliding Window Decomposition*. This transformation converts any $(2N-2):2N$ weights into 2:4 sparsity format (NVIDIA,

2021), unlocking $2\times$ acceleration from Sparse Tensor Cores. The required activation rearrangement fuses into per-token quantization (Dettmers et al., 2022; Xiao et al., 2023) at low marginal cost.

Figure 1 shows end-to-end speedup on A100 across models from 1B to 14B parameters. As model size grows, GEMM becomes compute-bound and speedup approaches the theoretical limit $S_{\max} = N/(N-1)$ (§3). For 6:8 sparsity on Qwen2.5-7B, SlideSparse achieves $1.33\times$—exactly matching this upper-bound.

Our contributions are as follows:

- **Sparsity–accuracy characterization.** We show across Qwen2.5-7B/14B and both Wanda/SparseGPT that 6:8 stays within 0.4–1.8 average accuracy points of dense, while 2:4 loses 14.9–16.5 points (§2).
- **Sliding Window Decomposition.** We prove that $N-1$ stride-2 windows are *necessary and sufficient* for lossless $(2N-2):2N \rightarrow 2{:}4$ transformation, achieving optimal expansion factor $\gamma = (2N-2)/N$ (§3).
- **SlideSparse system.** We design a three-phase pipeline: offline packer, initial compression, and online fused kernel. Activation lifting piggybacks on per-token quantization at low marginal cost (§4).
- **Empirical validation.** On six GPUs (A100, H100, B200, RTX 4090, RTX 5080, DGX-spark) and five precisions (FP4, INT8, FP8, BF16, FP16), speedup approaches the theoretical $N/(N-1)$ bound on compute-bound workloads—Qwen2.5-7B with 6:8 sparsity achieves exactly $1.33\times$, matching the theoretical speedup bound (§5).

**Broader Impact.** SlideSparse bridges the gap between accuracy-conscious pruning (Frantar & Alistarh, 2023; Sun et al., 2024) and hardware efficiency, offering a practical deployment path for $(2N-2):2N$ sparse LLMs.

## 2. Motivation

### 2.1. 2:4 Sparsity: Fast but Too Aggressive

NVIDIA's Sparse Tensor Cores *double* matrix throughput for 2:4 structured sparsity (Mishra et al., 2021), but impose a strict constraint: at least 2 of every 4 consecutive weights must be zero. Deviate from this pattern, and the hardware falls back to dense execution—*no acceleration*.

We test whether rigid 50% pruning exceeds LLM compression tolerance at practical model scales. Figure 2 summarizes average accuracy over 9 benchmarks for Qwen2.5-7B (FP16) and Qwen2.5-14B (BF16) under two one-shot pruning methods (Wanda (Sun et al., 2024) and SparseGPT (Frantar & Alistarh, 2023)), across Dense, 8:10, 6:8, 4:6, and 2:4 sparsity.

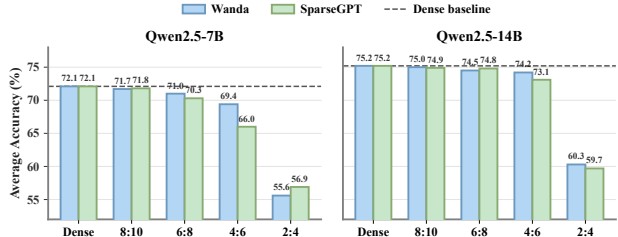

*Figure 2.* **Average accuracy across 9 benchmarks under structured sparsity.** On Qwen2.5-7B and Qwen2.5-14B, 6:8 stays within 0.4–1.8 points of dense accuracy across Wanda and SparseGPT, while 2:4 drops by 14.9–16.5 points.

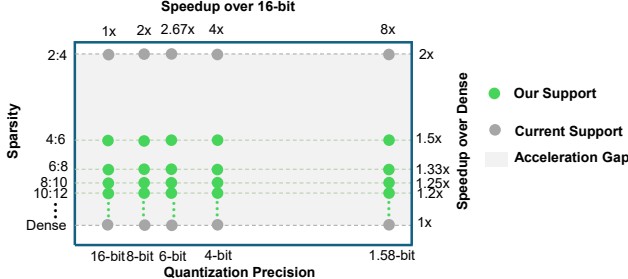

*Figure 3.* **Two-dimensional compression space for LLM acceleration.** X-axis: quantization precision (16-bit to 1.58-bit BitNet, up to $8\times$ speedup). Y-axis: sparsity (dense to 2:4, up to $2\times$ speedup). Gray dots mark existing hardware support—limited to dense or 2:4 extremes. Green dots show $(2N-2):2N$ patterns that SlideSparse enables, filling the *Acceleration Gap* and unlocking fine-grained sparsity–precision trade-offs.

Figure 2 shows a consistent pattern across models and pruning methods. Moderate sparsity such as 8:10 and 6:8 preserves accuracy far better than rigid 2:4. At 6:8, the average accuracy gap is only 0.4–1.8 points; at 2:4, it widens to 14.9–16.5 points (full per-benchmark breakdown in Appendix D.4). This motivates studying the $(2N-2):2N$ sparsity family, which retains $(2N-2)$ non-zeros per $2N$ elements and offers superior accuracy–efficiency trade-offs. Our finding suggests that *moderate sparsity with hardware support* is more valuable than aggressive sparsity.

### 2.2. The Deployment Gap

$(2N-2):2N$ patterns preserve accuracy but lack hardware support. Sparse Tensor Cores accelerate only 2:4 (Mishra et al., 2021; Pool & Yu, 2021); inference engines (Kwon et al., 2023; NVIDIA, 2023) cannot recognize 6:8 or 14:16 and must expand sparse weights to dense form.

The consequence: 6:8 models contain only 75% non-zero weights yet receive *zero latency reduction*. The GPU wastes compute and bandwidth resources on useless zeros. The algorithmic benefit of $(2N-2):2N$ with superior accuracy retention *does not translate to deployment speedup*.

## 2.3. Our Approach: Computational Arbitrage

We bridge this gap through *computational arbitrage*: trading data expansion for hardware compatibility. Our insight: any $(2N-2){:}2N$ block *decomposes losslessly* into $(N-1)$ overlapping 2:4-compliant windows. Each window satisfies the hardware constraint; collectively they preserve the original computation exactly.

This decomposition expands GEMM by factor $\gamma$. As long as the $2\times$ Sparse Tensor Core speedup exceeds $\gamma$, we achieve net acceleration. For 6:8 ($\gamma = 1.5$), theory predicts $\mathbf{1.33\times}$ over dense—unlocking real speedup from moderate sparsity for the first time. Section 3 formalizes *Sliding Window Decomposition* and proves optimality.

**A New Acceleration Dimension.** LLM inference acceleration relies almost exclusively on quantization: 16-, 8-, 4-, or 1.58-bit (Wang et al., 2023) precision yields up to $8\times$ speedup (Figure 3, X-axis). Sparsity, by contrast, remains binary—dense or 2:4, leaving a significant *acceleration gap*. SlideSparse fills this gap with hardware-accelerated execution at intermediate densities such as 83.3% (10:12), 75% (6:8), and 66.7% (4:6). This enables **Sparsity** as a *second optimization dimension* orthogonal to **Quantization** for future model compression.

## 3. SlideSparse Method

This section presents our theoretical foundation. We formulate the sparsity mismatch as a constraint decomposition problem, propose a sliding window solution with provable mathematical guarantees, and analyze the computational trade-off.

### 3.1. Problem: The Sparsity Mismatch

Consider the linear layer $\mathbf{Y} = \mathbf{W}\mathbf{X}$ with $\mathbf{W} \in \mathbb{R}^{M \times K}$, a fundamental building block in Transformer architectures (Vaswani et al., 2017). We define two constraint sets:

**Hardware Constraint $\mathcal{C}_{HW}$ (2:4 Sparsity).** NVIDIA Sparse Tensor Cores (Mishra et al., 2021) require the strict pattern: at most 2 non-zeros per 4 consecutive elements,

$$\mathcal{C}_{HW} = \{\mathbf{w} \in \mathbb{R}^K \mid \|\mathbf{w}_{4i:4i+4}\|_0 \leq 2, \forall i\} \quad (1)$$

**Algorithm Constraint $\mathcal{C}_{Alg}$ ($(\mathbf{2N-2}):\mathbf{2N}$ Sparsity).** Accuracy-preserving pruning often yields a *relaxed* pattern: at most $(2N-2)$ non-zeros per $2N$ elements (e.g., 6:8),

$$\mathcal{C}_{Alg} = \{\mathbf{w} \in \mathbb{R}^K \mid \|\mathbf{w}_{2Ni:2Ni+2N}\|_0 \leq 2N-2, \forall i\} \quad (2)$$

**The Incompatible Gap.** The $(2N-2) : 2N$ budget is *global* (over $2N$ positions), whereas 2:4 is *local* (every 4 consecutive elements). Non-zeros may cluster to satisfy

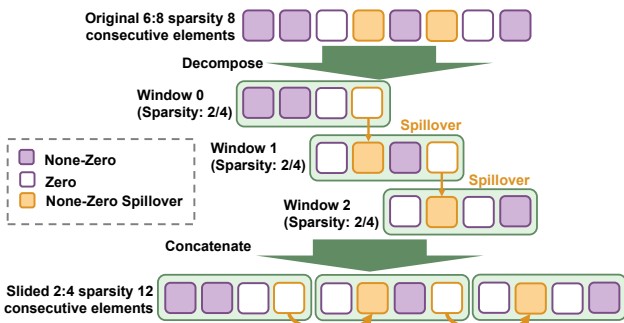

*Figure 4.* **Sliding window decomposition for 6:8 sparsity.** Three stride-2 windows (each size 4) cover all 8 positions. Overlap regions allow non-zeros to spill into the next window when one reaches capacity, converting any $(2N-2) : 2N$ pattern into concatenated 2:4 blocks for Sparse Tensor Core acceleration.

the global budget yet violate local 2:4 windows, but Sparse Tensor Cores *cannot process* such vectors.

**Our Goal.** Find operators $\Phi$ (for weights) and $\Psi$ (for inputs) such that the computation is *mathematically equivalent* while *physically* using only 2:4 operations:

$$\mathbf{w}^\top \mathbf{x} = \Phi(\mathbf{w})^\top \Psi(\mathbf{x}), \quad \text{where } \Phi(\mathbf{w}) \in \mathcal{C}_{HW} \quad (3)$$

The key challenge is that $\Phi$ and $\Psi$ must *jointly* preserve the inner product while transforming the sparsity pattern.

### 3.2. Solution: Sliding Window Decomposition

A $(2N-2) : 2N$ block contains up to $(2N-2)$ non-zeros, but each 2:4 window holds only 2. Therefore, multiple windows are needed. With non-overlapping windows (e.g., $[0-3], [4-7]$ for 6:8), total capacity is $2 \times 2 = 4 < 6$, which is insufficient. We propose *overlapping* windows to solve this: stride-2 placement yields $K = N-1$ windows with total capacity $2K = 2N-2$, exactly matching the non-zero count.

**Theorem 1 (Coverage).** $K = N-1$ *overlapping windows of size 4 and stride 2 are necessary and sufficient to cover any* $(2N-2) : 2N$ *sparse block.*

*Proof sketch.* (Full proof in Appendix C.) Each window holds up to 2 non-zeros; $K$ windows provide total capacity $2K$. To cover $2N-2$ non-zeros, we need $K \geq N-1$. Conversely, the overlapping structure ensures that any non-zero rejected by window $j$ (due to capacity) lies in the overlap region and is covered by window $j+1$.

**Example (6:8).** With $N = 4$, we use $K = 3$ windows to cover three indices $\{0-3\}, \{2-5\}, \{4-7\}$. Total capacity is $3 \times 2 = 6$, exactly matching the 6 non-zeros. Concatenating these windows yields an *expanded* representation of $4K = 12$ elements—a $1.5\times$ expansion from the original 8 elements. The cost analysis in §3.4 quantifies this trade-off.

**Corollary (Optimality).** $K = N-1$ is the *minimum* window count: fewer windows cannot provide sufficient capacity ($2K < 2N-2$). Thus SlideSparse achieves the theoretically optimal expansion.

This decomposition-and-concatenation procedure defines the weight transformation $\Phi$ from §3.1: each $(2N-2) : 2N$ block maps to $K$ concatenated 2:4 windows, yielding $\Phi(\mathbf{w}) \in \mathcal{C}_{HW}$.

### 3.3. Activation Lifting

Weight transformation $\Phi$ decomposes each $(2N-2) : 2N$ block into $K$ overlapping 2:4 windows. To preserve math correctness, inputs require a corresponding transformation. The *lifting operator* $\Psi : \mathbb{R}^{2N} \to \mathbb{R}^{K \times 4}$ replicates input elements according to window coverage:

$$\Psi(\mathbf{x}) = \begin{bmatrix} x_0 & x_1 & x_2 & x_3 \\ x_2 & x_3 & x_4 & x_5 \\ x_4 & x_5 & x_6 & x_7 \end{bmatrix} \quad \text{(6:8 example)} \quad (4)$$

Row $j$ contains $(x_{2j}, x_{2j+1}, x_{2j+2}, x_{2j+3})$, the four elements visible to window $j$. After reconstruction, $\mathbf{w}^\top \mathbf{x} = \sum_{j=0}^{K-1} \mathbf{w}_j^\top [\Psi(\mathbf{x})]_j$, preserving mathematical equivalence.

Crucially, $\Psi$ involves no arithmetic—it is pure index remapping. This enables fusion with quantization kernels: since LLM inference already requires per-token quantization (INT8/FP8/FP4), lifting piggybacks on the store phase at low marginal cost (§4).

### 3.4. Cost Analysis: When Does SlideSparse Pay Off?

The *Expansion Factor* $\gamma$ quantifies the computational overhead:

$$\gamma = \frac{K \cdot 4}{2N} = \frac{(N-1) \cdot 4}{2N} = 2 - \frac{2}{N} \quad (5)$$

For 6:8 ($N = 4$): $\gamma = 1.5$; for 14:16 ($N = 8$): $\gamma = 1.75$.

**Speedup Condition.** SlideSparse accelerates when $\gamma < \alpha$, where $\alpha \approx 2.0\times$ is the hardware speedup from 2:4 sparsity. Since $\gamma < 2$ for all $N > 2$, the condition always holds. Theoretical speedup bound: $S_{\text{eff}} = \alpha/\gamma = N/(N-1)$. More generally, we prove that for *any* $Z{:}L$ sparsity pattern mapped to $M{:}N$ hardware, the effective speedup is bounded by $S_{\text{eff}} \leq L/Z = 1/\text{density}$—the theoretical maximum is determined solely by density (Appendix C.1).

This analysis establishes theoretical soundness: SlideSparse preserves correctness while guaranteeing speedup. The next section addresses the *practical* challenge of implementing $\Phi$ and $\Psi$ efficiently on modern GPUs.

## 4. SlideSparse System Implementation

This section describes how to deploy SlideSparse in production LLM serving. The system comprises three phases (Figure 5): (1) *offline* weight preprocessing via pruning and sliding, (2) *initialization* that compresses weights into 2:4 format via cuSPARSELt (NVIDIA, 2021), and (3) *per-request* execution of fused kernels. We demonstrate integration with vLLM (Kwon et al., 2023) as a representative serving framework.

**Synergy with Quantization.** Our fused kernel performs activation lifting $\Psi$ *within* the per-token quantization pass, achieving dimensional expansion at low marginal cost.

### 4.1. Offline Weight Packer

The weight packer implements the constructive proof of Theorem 3.2: given a $(2N-2) : 2N$ sparse weight matrix, it produces an equivalent 2:4 sparse matrix with expansion factor $\gamma$. The key insight is that the 2-position overlap between adjacent windows acts as a "spillover buffer"—when one window reaches its capacity of 2 non-zeros, excess elements are guaranteed to fall within the next window's coverage. The algorithm iterates over stride-2 windows and greedily assigns up to 2 non-zeros per window; rejected elements remain candidates for the next window via this overlap region. This $O(K)$ procedure runs offline before deployment, adding no runtime cost. Pseudocode and correctness analysis are provided in Appendix B.

### 4.2. Fused Quantization-Slide Kernel

The kernel realizes activation lifting $\Psi$ (§3.3) while *hiding* the $\gamma\times$ expansion within the quantization pass. We implement this kernel in Triton (Tillet et al., 2019) for portability across GPU architectures. A naive two-step approach (first quantize, then slide) requires four memory operations: read $\mathbf{X}$, write quantized $\mathbf{X}'$, read $\mathbf{X}'$, and write expanded $\mathbf{Y}$. Our fused kernel reduces this to two: read $\mathbf{X}$ and write $\mathbf{Y}$ directly. Compared to standard quantization (read $\mathbf{X}$, write $\mathbf{X}'$), the only additional cost is writing $\gamma K$ instead of $K$ elements per row: a $(\gamma - 1) \approx 0.5\times$ overhead for 6:8 sparsity, easily amortized by the $\sim 2\times$ sparse GEMM speedup. Algorithm 1 presents the pseudocode.

**Output-Oriented Design.** We flatten the nested group-window loop into a single iteration over global window index $j$ (line 10), recovering group $g$ and local offset $\ell$ via integer division (line 11). The index formula $b = 2Ng + 2\ell$ directly realizes the lifting operator $\Psi$: loading 4 elements from position $b$ produces the overlapping window structure.

**Two-Pass Fusion.** Each row is processed by one thread-block rather than using 2D-tiles for better L2 cache usage. *Pass 1* (lines 6–8) computes per-row absmax for dynamic quantization (FP8 or INT8). *Pass 2* (lines 9–19) fuses quantization with sliding—the entire "read $\rightarrow$ quantize $\rightarrow$ slide $\rightarrow$ pack $\rightarrow$ write" pipeline executes in registers, avoiding intermediate buffers and I/O latency.

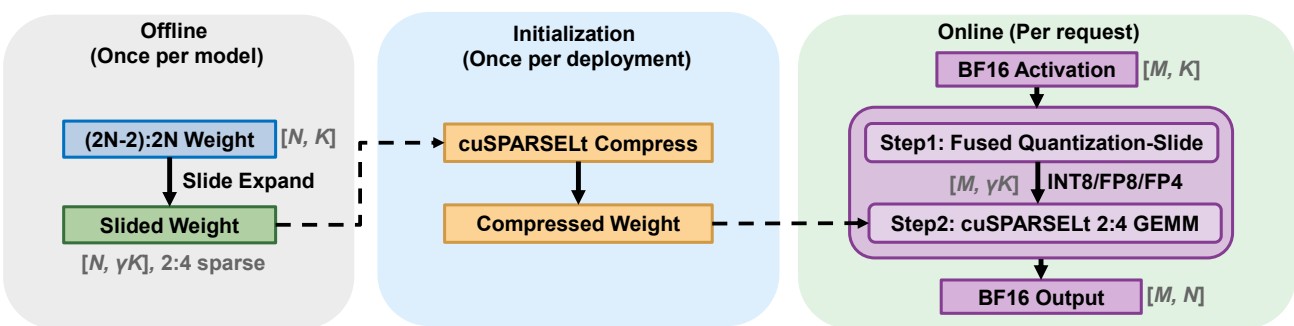

*Figure 5.* **SlideSparse system overview. Offline**: Weight preprocessing transforms $(2N-2) : 2N$ sparse weights into slide-expanded format with $\gamma\times$ expansion. **Initialization**: cuSPARSELt compresses weights into 2:4 format at model load time. **Online**: Per-request inference executes fused quantization-slide kernel followed by sparse GEMM.

---

**Algorithm 1** Fused Quantization-Slide Kernel

---

1: **Input:** $\mathbf{X} \in \mathbb{R}^{M \times K}$, block size $2N$
2: **Output:** $\mathbf{Y}$ (INT8/FP8), scales $\mathbf{s} \in \mathbb{R}^M$
3: $n_g \leftarrow \lceil K/2N \rceil; n_w \leftarrow n_g \cdot (N-1)$
4: Initialize $\mathbf{Y} \in \mathbb{R}^{M \times 4n_w}$
5: **for** row $i = 1$ **to** $M$ **in parallel do**
6:     */* Pass 1: dynamic quantization scale */*
7:     $a \leftarrow \max_k |X_{i,k}|; r \leftarrow Q_{\max}/a$
8:     $s_i \leftarrow a/Q_{\max}$
9:     */* Pass 2: output-oriented fused loop (§4.2) */*
10:     **for** $j = 0$ **to** $n_w - 1$ **do**
11:         $g \leftarrow \lfloor j/(N-1) \rfloor; \ell \leftarrow j \bmod (N-1)$
12:         $b \leftarrow 2Ng + 2\ell$
13:         */* realize activation lifting $\Psi$ (§3.3) */*
14:         $\mathbf{x} \leftarrow (X_{i,b}, X_{i,b+1}, X_{i,b+2}, X_{i,b+3})$
15:         $\mathbf{q} \leftarrow \text{Clamp}(\mathbf{x} \cdot r, -Q_{\max}, Q_{\max})$
16:         */* vectorized byte packing: 4 bytes → 1 word */*
17:         $p \leftarrow q_0 | (q_1 \ll 8) | (q_2 \ll 16) | (q_3 \ll 24)$
18:         $\mathbf{Y}_{i,j} \leftarrow p$
19:     **end for**
20: **end for**
21: **return** $\mathbf{Y}, \mathbf{s}$

---

**Vectorized Byte Packing.** Line 17 packs 4 quantized bytes into one 32-bit word via bit-shifting, achieving $4\times$ store efficiency. The packed format aligns with cuSPARSELt's 2:4 layout, enabling zero-copy handoff to sparse GEMM.

We empirically verify in Appendix D.2 that the fused kernel achieves near memory-bandwidth-bound throughput, confirming that the slide expansion adds minimal overhead beyond the unavoidable I/O cost.

### 4.3. System Integration

We use cuSPARSELt (NVIDIA, 2021) as the sparse GEMM backend for its broad hardware coverage (Ampere through Blackwell) and consistent performance. cuSPARSELt com-

presses 2:4 sparse weights into a hardware-optimized format storing only non-zeros plus compact metadata; the slide expansion thus incurs no storage overhead. This compression can be performed entirely offline; we apply it at model load time for simpler vLLM integration.

**Minimal-Invasive Design.** vLLM abstracts linear layer computation through a quantization interface. We implement a custom backend that intercepts these calls, redirecting them to our fused kernel followed by cuSPARSELt sparse GEMM. All other vLLM components including attention, KV cache, scheduling, and tensor parallelism remain unchanged. Users enable SlideSparse via a single configuration flag.

**Generality.** Our core algorithm—overlapping window decomposition plus 2:4 sparse GEMM—is framework-agnostic and can be adapted to TensorRT-LLM (NVIDIA, 2023), SGLang (Zheng et al., 2024a), or other serving systems.

## 5. Experiments

We evaluate SLIDESPARSE through kernel benchmarks and end-to-end inference on vLLM (Kwon et al., 2023). Our evaluation spans five precisions (FP4, INT8, FP8, BF16, FP16), six GPUs across three architecture generations, and workloads ranging from decode ($M{=}64$) to prefill ($M{=}65536$). We highlight INT8, FP8, and BF16 in the main text; FP4 and FP16 results appear in Appendix D.

### 5.1. Experimental Setup

**Hardware.** We test on six NVIDIA GPUs across three architecture generations: **Datacenter:** A100 (80GB, Ampere), H100 (80GB, Hopper), B200 (180GB, Blackwell); **Consumer:** RTX 4090 (24GB, Ada Lovelace), RTX 5080 (16GB, Blackwell); **Embedded:** DGX Spark GB10 (128GB, Blackwell). All GPUs support 2:4 structured sparsity via Sparse Tensor Cores (Mishra et al., 2021).

**Models and Workloads.** We evaluate Llama3.2-1B, Llama3.2-3B (Dubey et al., 2024), Qwen2.5-7B, Qwen2.5-14B (Qwen et al., 2025), and BitNet b1.58 (2B). Workloads span both **decode** ($M$=64–512, representing concurrent batch size) and **prefill** ($M$=512–65536, where $M$ = batch_size × seq_len).

**Baselines and Metrics.** We report speedup ratio over **cuBLASLt** (NVIDIA, 2024) (dense GEMM baseline), and SlideSparse with three sparsity patterns: **4:6** ($N$=3, theoretical $S_{\text{eff}} \leq 1.5\times$), **6:8** ($N$=4, $S_{\text{eff}} \leq 1.33\times$), and **8:10** ($N$=5, $S_{\text{eff}} \leq 1.25\times$). The Native 2:4 via cuSPARSELt (NVIDIA, 2021) serves as an upper bound.

**Accuracy Evaluation.** To complement throughput results, we evaluate accuracy on Qwen2.5-7B and Qwen2.5-14B across 9 benchmarks (full breakdown in Appendix D.4). We test Dense, 8:10, 6:8, 4:6, and 2:4 using two one-shot pruning methods: Wanda (Sun et al., 2024) and SparseGPT (Frantar & Alistarh, 2023). SparseGPT uses 128 WikiText-2 calibration samples with sequence length 2048, Wanda is calibration-free, and neither method uses sparse-aware fine-tuning. We average 7 commonsense reasoning tasks together with MMLU (Hendrycks et al., 2021) and GSM8K (Cobbe et al., 2021).

### 5.2. Kernel Performance

We first isolate kernel-level performance from system overhead. Figure 6 reports sparse GEMM speedup at $M$=16384 across three precisions (INT8, FP8, BF16) and five GPUs.

**INT8.** On A100, 6:8 sparsity achieves **1.41–1.42×** speedup, exceeding the $1.33\times$ prediction based on $\alpha$=2. This is consistent with native 2:4 on A100 achieving 2.03–2.08× over cuBLASLt at this matrix size (i.e., $\alpha > 2$), which propagates the excess to SlideSparse. On B200, 6:8 reaches **4.06–4.32×**; native 2:4 reaches 6.25–6.38×. These anomalously high gains arise because cuBLASLt's INT8 GEMM is not yet fully optimized on Blackwell—the dense baseline is slower than expected, inflating all speedup ratios.

**FP8 and BF16.** SlideSparse generalizes across precisions. On FP8, RTX 4090 achieves **1.35–1.37×** at 6:8—approaching the $1.33\times$ theoretical bound. On BF16, B200 and RTX 5080 reach 1.10–1.23× at 6:8 sparsity. These results confirm that SlideSparse extends beyond INT8 to FP8 and full-precision workloads.

**Kernel Scaling with M.** Figure 7 shows kernel speedup vs. $M$ on A100 and B200 (Qwen-7B, INT8). On A100, speedup increases with $M$. Notably, 6:8 improves from $1.34\times$ ($M$=2048) to $1.42\times$ ($M$=16384), consistently exceeding the $1.33\times$ theoretical bound. On B200, speedup

is consistently high across all $M$ (6:8: 4.0–4.3×), as the suboptimal dense baseline amplifies gains at every scale.

### 5.3. End-to-End Inference Performance

We now evaluate whether kernel gains translate to end-to-end inference. Figure 8 shows vLLM speedup on A100, B200 (INT8), and RTX 4090 (FP8).

**Key Observations.** SlideSparse accelerates the entire $(2N-2):2N$ family across different platforms. On A100 (INT8 prefill), 6:8 achieves **1.29–1.34×** across model sizes—approaching the $1.33\times$ theoretical bound. On B200 (INT8 prefill), 6:8 achieves **1.06–1.11×**; the smaller margin reflects B200's already highly optimized dense cuBLASLt baseline on Blackwell. On RTX 4090 (FP8 prefill), 6:8 reaches **1.18–1.19×**, demonstrating that SlideSparse benefits both datacenter and consumer GPUs. These E2E gains confirm that kernel-level speedups translate to real inference workloads.

**Decode Regime.** In decode scenarios ($M$=64–512), SlideSparse achieves **1.07–1.21×** speedup from two sources: sparse-Tensor-Core compute acceleration and 25% weight memory reduction under cuSPARSELt compression. SlideSparse targets the linear projection layers (QKV, output, gate/up/down), which are orthogonal to attention optimizers such as FlashAttention (Dao et al., 2022) and FlexAttention (Dong et al., 2025)—the two can be combined for compounding gains across the full inference pipeline.

**Serving Metrics.** On Qwen2.5-14B INT8 (A100) with a 4096-token prompt, 6:8 reduces time-to-first-token from 390 ms to 291 ms (**1.34×**). Total request latency improves by **1.13×** for short generation (128 output tokens) and **1.11×** for longer generation (512 output tokens), confirming that the prefill gains survive in realistic request mixes (details in Appendix D.5).

**Unlocking 2:4 Potential.** Does the sliding window transformation introduce hidden overhead, or does SlideSparse fully exploit the underlying 2:4 hardware? To answer this, we define *efficiency* as the ratio of measured speedup to the theoretical expectation derived from 2:4 baseline performance (Figure 9). If 2:4 achieves speedup $S_{2:4}$ over dense, then $(2N-2):2N$ should theoretically reach $S_{2:4} \times \gamma^{-1}$, where $\gamma$ is the expansion factor (§3). Efficiency of 100% means SlideSparse perfectly transmits the 2:4 benefit; values exceeding 100% reveal *additional* gains.

Remarkably, SlideSparse *exceeds* 100% efficiency on all datacenter GPUs in both INT8 and FP8. For INT8, 6:8 achieves **115%** (A100), **119%** (H100), and **134%** (B200). For FP8, 6:8 reaches **117%** (H100) and **122%** (B200)—confirming that the gains generalize across precisions. Values consis-

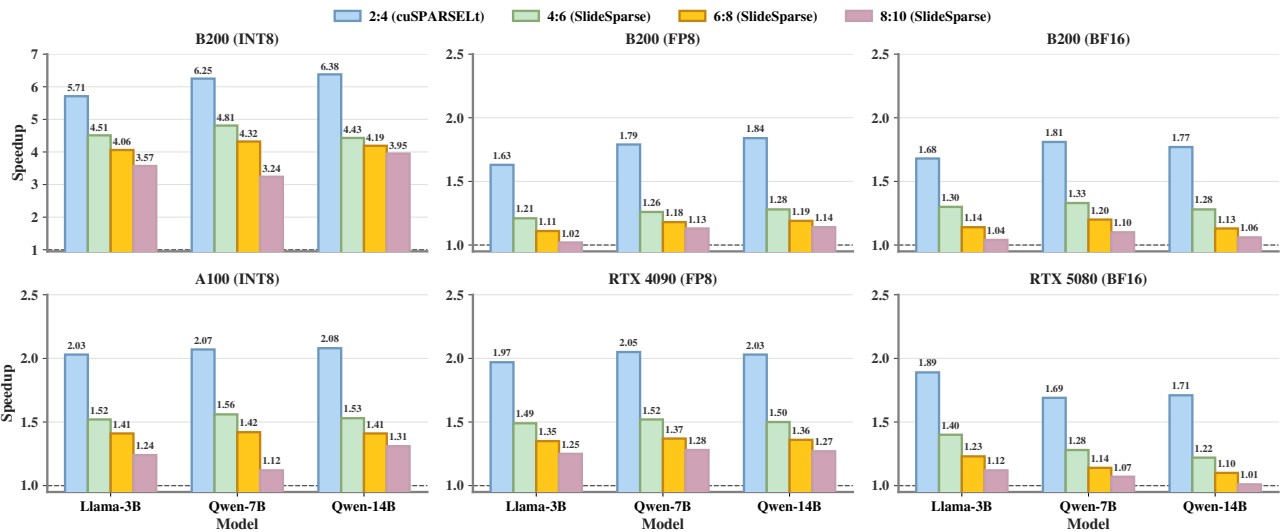

*Figure 6.* **Kernel-level speedup at** M=**16384 across varied precisions and GPUs.** Top row: B200 (INT8, FP8, BF16); Bottom row: A100 INT8, RTX 4090 FP8, RTX 5080 BF16. B200 INT8 achieves 4–6× due to suboptimal cuBLASLt INT8 performance on Blackwell; other configurations approach theoretical $S_{\text{eff}}$ bounds. Comprehensive kernel results are shown in Appendix D.3.

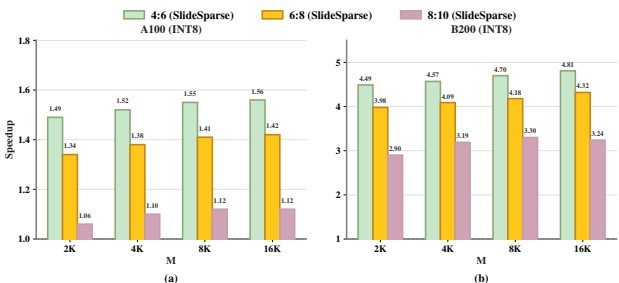

*Figure 7.* **Kernel speedup vs. M on A100 and B200 (Qwen-7B, INT8).** (a) A100: speedup increases with $M$, approaching the theoretical limit for 6:8. (b) B200: consistently high speedup across all $M$ (note different Y-axis scale).

tently above 100% indicate that, in compute-bound settings, SlideSparse introduces little practical overhead and can even unlock additional Sparse Tensor Core utilization beyond native 2:4 workflows. Our fused quantization-slide kernel (§4) ensures that activation lifting incurs low marginal cost, preserving these gains in end-to-end inference.

**Scaling with M.** Figure 10 shows E2E speedup vs. $M$ on B200 (Qwen-7B, INT8). Across both decode ($M\leq512$) and prefill ($M$ up to 32K), 6:8 consistently achieves 1.05–1.21× speedup. This confirms that SlideSparse benefits workloads across the full $M$ range.

**Summary.** SlideSparse delivers consistent speedups across the $(2N-2):2N$ family. At the kernel level, 6:8 sparsity reaches **1.42×** on A100 INT8—exceeding the 1.33× theoretical bound due to native 2:4 achieving more than 2×

throughput. At the end-to-end level, Qwen2.5-7B with 6:8 achieves **1.33×** on A100 (INT8, $M$=8192)—matching the theoretical $N/(N-1)$ bound exactly—and **1.19×** on RTX 4090 (FP8 prefill). Furthermore, efficiency consistently exceeds 100% across datacenter GPUs, demonstrating that SlideSparse not only avoids overhead but further unlocks the potential of Sparse Tensor Cores. These results validate SlideSparse's core claim: sliding window decomposition enables practical acceleration for the $(2N-2):2N$ sparsity family on existing 2:4 hardware, spanning datacenter and consumer GPUs, multiple precisions, and workloads from decode to prefill. Full results appear in Appendix D.

## 6. Related Work

Our work intersects three research areas: structured sparsity for neural networks, hardware-aware model optimization, and efficient LLM inference systems.

**Structured Sparsity.** Neural network pruning has evolved from early unstructured approaches (Han et al., 2015; 2016) to structured methods that remove entire neurons, channels, or attention heads (Li et al., 2017). The lottery ticket hypothesis (Frankle & Carbin, 2019) provided theoretical grounding for sparse network trainability. NVIDIA's 2:4 sparsity pattern (Mishra et al., 2021) represents a hardware-friendly middle ground: it constrains exactly 2 zeros per 4 consecutive elements, enabling 2× Tensor Core throughput. Follow-up work explored training recipes for N:M sparsity (Zhou et al., 2021; Pool & Yu, 2021; Hubara et al., 2021), demonstrating that 2:4 sparse networks can be trained from scratch without accuracy loss on vision tasks. How-

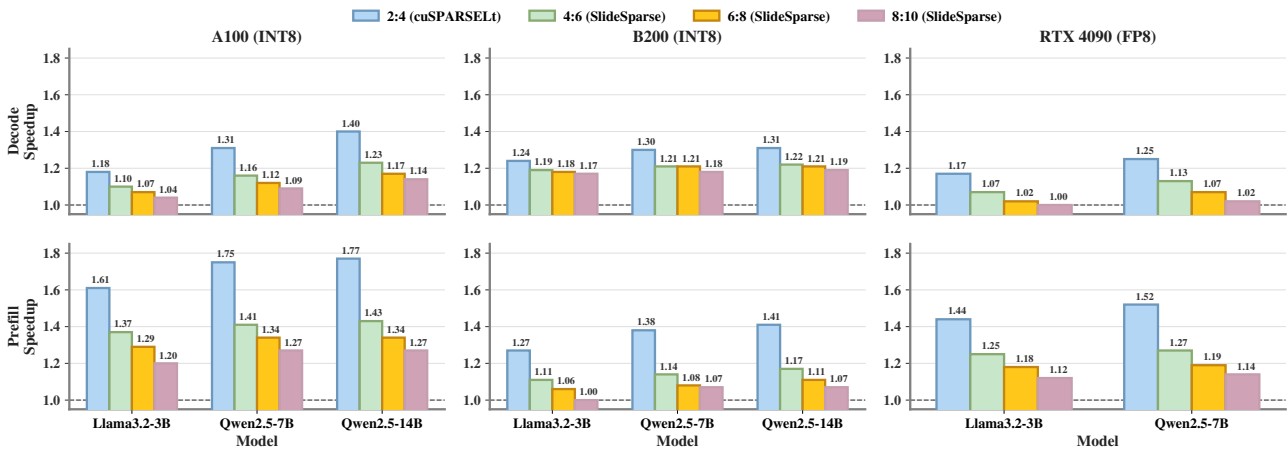

*Figure 8.* **End-to-end inference speedup.** Top: Decode; Bottom: Prefill. A100/B200: INT8 ($M$=512 decode, $M$=16384 prefill); RTX 4090: FP8 ($M$=512 decode, $M$=8192 prefill; 24GB memory limits exclude Qwen-14B). For all E2E results see Appendix D.5.

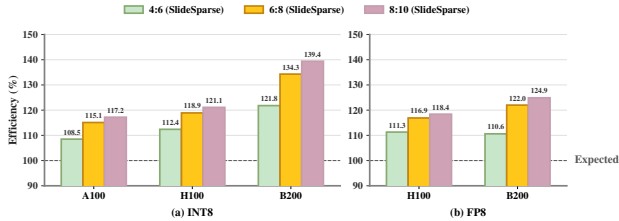

*Figure 9.* **Efficiency: actual speedup vs. expected speedup** (Qwen2.5-7B, Prefill, $M$=8192). (a) INT8; (b) FP8 (A100 lacks FP8 support). Expected speedup = (2:4 baseline) $\times \gamma^{-1}$. Values >100% indicate SlideSparse's fused kernel unlocks additional performance beyond 2:4 predictions.

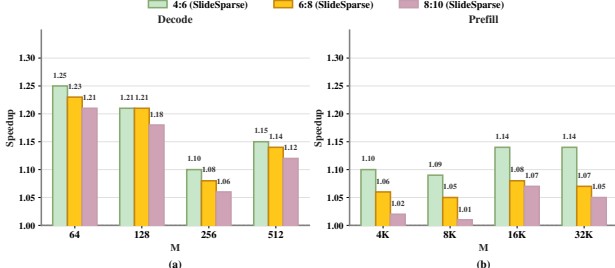

*Figure 10.* **Speedup vs. M** on B200 (Qwen-7B, INT8). (a) Decode ($M \in \{128, 256, 512\}$); (b) Prefill ($M \in \{4K, 8K, 16K, 32K\}$). All $(2N-2){:}2N$ patterns achieve consistent speedups across $M$.

ever, these approaches target the 50% sparsity enforced by 2:4—our work enables hardware acceleration for *milder* sparsity levels that better preserve LLM accuracy.

**Sparsity in Large Language Models.** Pruning LLMs presents unique challenges due to their scale and the distributed nature of knowledge. SparseGPT (Frantar & Alistarh, 2023) achieves one-shot pruning to 50% unstructured sparsity with minimal accuracy loss, while Wanda (Sun et al., 2024) further simplifies the process using activation-weighted importance scores. For structured pruning, Sheared LLaMA (Xia et al., 2024) and ZipLM (Kurtic et al., 2023) demonstrate effective layer and head removal with continued pre-training. For structured sparsity, however, the 50% constraint of 2:4 often exceeds LLM compression tolerance. Our expanded evaluation confirms this on Qwen2.5-7B and Qwen2.5-14B: across both Wanda and SparseGPT, 6:8 preserves accuracy far better than 2:4, which suffers severe degradation (§2), motivating our focus on the milder $(2N-2):2N$ family.

**Quantization for LLM Inference.** Quantization reduces memory footprint and enables faster Tensor Core operations (Dettmers et al., 2022; Xiao et al., 2023; Frantar et al., 2023). SmoothQuant (Xiao et al., 2023) addresses activation outliers by migrating quantization difficulty from activations to weights, enabling INT8 inference. More aggressive schemes like GPTQ (Frantar et al., 2023), AWQ (Lin et al., 2024), and SpQR (Dettmers et al., 2024) achieve 4-bit or lower weight quantization with near-lossless accuracy. At the extreme, BitNet (Wang et al., 2023) explores binary $\{-1, +1\}$ weights, while BitNet b1.58 (Ma et al., 2024) extends this to ternary $\{-1, 0, +1\}$ weights that match full-precision accuracy. Concurrent work Sherry (Huang et al., 2026) also targets 75% density (3:4 sparsity) for ternary quantization, providing independent evidence that this sparsity level is a favorable operating point—the same density as our 6:8 pattern. Recent mixed-precision methods such as LLM-MQ (Li et al., 2023), SliM-LLM (Huang et al., 2025), and MixLLM (Zheng et al., 2024b) explore the quantization axis through non-uniform bit allocation. These methods are complementary to SlideSparse: they reduce bits per value, whereas SlideSparse reduces the number of stored

and computed values, and the two axes compose naturally.

**Efficient Inference Systems.** Production LLM serving systems like vLLM (Kwon et al., 2023), SGLang (Zheng et al., 2024a), TensorRT-LLM (NVIDIA, 2023), and Flex-Gen (Sheng et al., 2023) optimize memory management, batching, and structured generation, but rely on standard dense or 2:4 sparse GEMM kernels. Triton (Tillet et al., 2019) has become the de facto standard for custom GPU kernel development, enabling rapid prototyping of fused operations. FlashAttention (Dao et al., 2022; Dao, 2023) demonstrates that algorithm-hardware co-design can yield substantial speedups by exploiting memory hierarchy. For sparse operations, NVIDIA's cuSPARSELt (NVIDIA, 2021) provides optimized 2:4 kernels. SlideSparse extends this ecosystem by enabling acceleration for sparsity patterns *beyond* the rigid 2:4 constraint, bridging the gap between algorithmic flexibility and hardware support.

**Sparse Attention and Sparse Compilation.** Sparse attention systems such as FlexAttention (Dong et al., 2025) and SPLAT (Gupta et al., 2025) optimize which token pairs are computed in the attention operator. They are orthogonal to SlideSparse, which targets weight-sparse GEMM in linear layers; the two can be combined in a single serving stack. More general sparse systems such as SparseTIR (Ye et al., 2023) provide compiler abstractions across many sparse formats and workloads, whereas SlideSparse gives a closed-form mapping specialized to NVIDIA's 2:4 Sparse Tensor Cores. On the hardware side, VEGETA (Jeong et al., 2023) and HighLight (Wu et al., 2023) require new CPU or accelerator support, while SlideSparse deploys on existing commodity GPUs. Jeong et al. (Jeong et al., 2025) also map broader sparsity to structured accelerators, but their decomposition is approximate for unstructured sparsity, whereas SlideSparse preserves the original dot product exactly for the $(2N-2) : 2N$ family.

**Summary.** Prior work on structured sparsity focuses primarily on 2:4, which demands 50% pruning that often exceeds LLM compression tolerance. SlideSparse is, to our knowledge, the first system to accelerate the $(2N-2) : 2N$ sparsity family on commodity GPUs, enabling practitioners to trade off between accuracy preservation and hardware speedup along a continuous spectrum rather than facing a binary choice.

## 7. Limitations

**Upstream Sparsification.** We evaluate throughput on post-hoc pruned checkpoints and report accuracy with one-shot Wanda and SparseGPT baselines. We do not perform large-scale sparse-aware training ourselves, and such training may further improve accuracy at higher sparsity levels (e.g., 4:6).

## 8. Conclusion

We presented SlideSparse, the first system to accelerate $(2N-2) : 2N$ structured sparsity on commodity GPUs by decomposing sparse blocks into overlapping 2:4-compliant windows. Across Qwen2.5-7B and Qwen2.5-14B, 6:8 preserves accuracy far better than 2:4 under both Wanda and SparseGPT, while achieving $1.33\times$ end-to-end speedup. SlideSparse opens a practical middle ground between aggressive 2:4 pruning and unaccelerated dense inference.

**Future Directions.** Now that $(2N-2) : 2N$ patterns can be hardware-accelerated, we encourage training LLMs with explicit $(2N-2) : 2N$ constraints from initialization. SlideSparse can also be integrated into frameworks such as TensorRT-LLM (NVIDIA, 2023) and SGLang (Zheng et al., 2024a), inspiring further exploration of the accuracy–efficiency Pareto frontier.

## Acknowledgements

This research is supported by the NSFC Fund No. U24B20151, and the SBSTR Fund No. KZD2024090310270801.

## Impact Statement

SlideSparse enables efficient deployment of structured sparse LLMs on commodity GPUs, reducing inference latency and energy consumption. The primary societal benefits include lower carbon footprint from LLM serving and democratized access to efficient AI infrastructure. We do not foresee direct negative societal impacts specific to our contribution; general concerns about LLM deployment apply broadly to the field.

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

# A. Implementation Details

### A.1. Kernel Implementation

We implement the sparse GEMM backend using cuSPARSELt (NVIDIA, 2021) rather than CUTLASS (NVIDIA, 2017). While CUTLASS offers greater flexibility for custom kernels, cuSPARSELt provides consistent performance across all target architectures (Ampere through Blackwell) without manual tuning. Additionally, CUTLASS's 2:4 sparse support has coverage gaps on certain compute capability and precision combinations.

### A.2. Weight Packing Pipeline

The offline weight packer is implemented in PyTorch with optional CUDA extensions for large models. On H100, packing throughput exceeds 10 GB/s, enabling full model conversion in under 30 seconds for Llama-3-70B (140GB weights). The packer outputs weights in cuSPARSELt's compressed sparse format, which can be serialized to disk for repeated loading.

### A.3. Integration with Inference Frameworks

We integrate SlideSparse with vLLM (Kwon et al., 2023) via its quantization interface. Our custom backend intercepts linear layer calls and redirects them to the fused kernel followed by cuSPARSELt sparse GEMM. This minimal-invasive design requires no changes to attention, KV cache, or scheduling components. Users enable SlideSparse by setting a single configuration flag at model load time.

# B. Offline Weight Packer Algorithm

This section provides the complete pseudocode and correctness analysis for the offline weight packer referenced in §4.1.

---

**Algorithm 2** Offline Weight Packer (Greedy Residual Allocation)

---

**Require:** Sparse row $\mathbf{w}[0 \ldots K-1]$ with $(2N-2):2N$ pattern
**Ensure:** Packed $\mathbf{w}'[0 \ldots \gamma K-1]$ with 2:4 pattern
1: $n_g \leftarrow K/2N$;   $\texttt{used}[0 \ldots K-1] \leftarrow \text{FALSE}$
2: **for** $g = 0$ **to** $n_g - 1$ **do**
3:   **for** $\ell = 0$ **to** $N - 2$ **do**
4:     $b \leftarrow 2Ng + 2\ell$ {stride-2 overlap}
5:     $\texttt{cnt} \leftarrow 0$
6:     **for** $\delta = 0, 1, 2, 3$ **do**
7:       **if** $\mathbf{w}[b+\delta] \neq 0$ **and** $\neg\texttt{used}[b+\delta]$ **and** $\texttt{cnt} < 2$ **then**
8:         $\mathbf{w}'[(N-1) \cdot 4g + 4\ell + \delta] \leftarrow \mathbf{w}[b+\delta]$
9:         $\texttt{used}[b+\delta] \leftarrow \text{TRUE}$;   $\texttt{cnt} \leftarrow \texttt{cnt} + 1$
10:      **end if**
11:    **end for**
12:  **end for**
13: **end for**
14: **return** $\mathbf{w}'$

---

### B.1. Correctness Analysis

**2:4 Compliance.**   Each window writes at most 2 non-zeros by construction: the condition $\texttt{cnt} < 2$ (line 7) enforces this invariant. Since each output window spans exactly 4 positions, the result satisfies the 2:4 constraint.

**Lossless Transformation.**   The $\texttt{used}$ array ensures each source non-zero is assigned exactly once. The key insight is the overlapping window design: adjacent windows share 2 positions (stride 2, window size 4). When window $\ell$ reaches capacity, any remaining non-zeros in the overlap region $[b+2, b+3]$ become candidates for window $\ell+1$. This *residual forwarding* guarantees that all $2N-2$ non-zeros are successfully distributed across the $N-1$ windows.

**Determinism.** The algorithm processes windows in fixed order (increasing $g$, then $\ell$, then $\delta$), and the greedy selection rule is deterministic. Identical inputs always produce identical outputs—a critical property for reproducible deployment.

# C. Mathematical Proofs

**Theorem 1 (Window Coverage).** For any $(2N-2) : 2N$ sparse vector $\mathbf{w} \in \mathbb{R}^{2N}$, there exists a decomposition into $N-1$ overlapping windows of size 4 with stride 2, such that each window satisfies the 2:4 constraint.

*Proof.* We construct the window index sets as:

$$
\begin{aligned}
I_0 &= \{0, 1, 2, 3\} \\
I_j &= \{2j, 2j{+}1, 2j{+}2, 2j{+}3\}, \quad j = 0, \ldots, N{-}2
\end{aligned}
$$

The total capacity across all windows is $2(N-1) = 2N-2$, exactly matching the number of non-zeros in $\mathbf{w}$. The greedy allocation in Algorithm 2 assigns each non-zero to the earliest window that (a) covers its index and (b) has remaining capacity. Since adjacent windows overlap on 2 positions, any non-zero rejected by window $j$ due to capacity is guaranteed to be covered by window $j{+}1$. By induction on $j$, all non-zeros are allocated. $\square$

**Corollary 1.1 (Minimal Window Count).** The minimum number of 2:4 windows required to cover a $(2N-2) : 2N$ group is exactly $N-1$.

*Proof.* Each window contributes at most 2 non-zeros. To accommodate $2N-2$ non-zeros, we need at least $\lceil (2N-2)/2 \rceil = N-1$ windows. Theorem 1 shows this bound is achievable. $\square$

**Corollary 1.2 (Theoretical Speedup Bound).** Let $\alpha$ denote the hardware speedup of 2:4 Sparse Tensor Cores over dense execution (nominally $\alpha = 2.0\times$). The effective speedup of SlideSparse for $(2N-2) : 2N$ sparsity is:

$$
S_{\text{eff}} = \frac{\alpha}{\gamma} = \frac{2.0}{(N-1) \cdot 4/2N} = \frac{4N}{4N-4} = \frac{N}{N-1} \tag{6}
$$

For 6:8 sparsity ($N = 4$), $\gamma = 1.5$ and $S_{\text{eff}} = 2.0/1.5 \approx 1.33\times$.

## C.1. Generalized Sliding Window Theory

This section generalizes sliding window decomposition from $(2N-2) : 2N \to 2{:}4$ to arbitrary $Z{:}L \to M{:}N$ transformations. The main results (Theorem 2 and Theorem 3) provide the theoretical foundation for the speedup bound discussed in §3.4.

### C.1.1. PROBLEM FORMULATION

**Definition.** Given a source sparsity pattern $Z{:}L$ (exactly $Z$ non-zeros in every $L$ consecutive elements) and hardware support for $M{:}N$ sparsity ($M$ non-zeros per $N$ elements), we seek a decomposition that transforms $Z{:}L$ blocks into a sequence of $M{:}N$-compliant blocks.

**Constraint.** The decomposition is valid only when the source is *at least as dense as* the hardware constraint:

$$
\frac{Z}{L} \geq \frac{M}{N} \tag{7}
$$

When $Z/L < M/N$, the source is already sparse enough for direct $M{:}N$ execution.

### C.1.2. SLIDING WINDOW CONSTRUCTION

We slide a window of size $N$ across the $L$-element source block with stride $s = N - M$:

• **Window count.** The number of windows is:

$$
w = \frac{L - N}{s} + 1 = \frac{L - N}{N - M} + 1 \tag{8}
$$

- **Total capacity.** Each window accepts at most $M$ non-zeros. The total capacity is $w \cdot M$.

- **Overlap region.** Adjacent windows overlap by $N - s = M$ positions, enabling *residual forwarding*—non-zeros rejected by window $j$ can be assigned to window $j + 1$.

### C.1.3. EXPANSION FACTOR DERIVATION

The expansion factor $\gamma$ is the ratio of output size to input size:

$$\gamma = \frac{w \cdot N}{L} = \frac{\left(\frac{L-N}{N-M} + 1\right) \cdot N}{L} \tag{9}$$

Simplifying:

$$\gamma = \frac{(L - N + N - M) \cdot N}{L(N - M)} = \frac{(L - M) \cdot N}{L(N - M)} \tag{10}$$

**Verification for** $(2N-2) : 2N \rightarrow 2{:}4$**:**

$$Z = 2N - 2, \quad L = 2N, \quad M = 2, \quad N_{\text{hw}} = 4 \tag{11}$$
$$s = 4 - 2 = 2 \tag{12}$$
$$w = \frac{2N - 4}{2} + 1 = N - 1 \tag{13}$$
$$\gamma = \frac{(N - 1) \cdot 4}{2N} = \frac{2(N - 1)}{N} = 2 - \frac{2}{N} \tag{14}$$

This matches the result derived in §3.

### C.1.4. SUFFICIENT CONDITION FOR VALID DECOMPOSITION

**Theorem 2 (Generalized Coverage).** The sliding window decomposition successfully transforms all $Z{:}L$ blocks into $M{:}N$-compliant blocks if and only if the total capacity equals or exceeds the number of non-zeros:

$$w \cdot M \geq Z \tag{15}$$

*Proof.* **Necessity:** If $wM < Z$, there are more non-zeros than capacity; decomposition fails.

**Sufficiency:** Suppose $wM \geq Z$. We prove by induction that greedy allocation succeeds.

*Base case:* Window 0 covers positions $\{0, 1, \ldots, N - 1\}$. It accepts up to $M$ non-zeros from these positions.

*Inductive step:* Assume windows $0, \ldots, j - 1$ have processed their covered positions. Window $j$ covers positions $\{js, \ldots, js + N - 1\}$. The overlap with window $(j - 1)$ is $\{js, \ldots, js + M - 1\}$, which has size $M$. Any non-zeros in this overlap region rejected by window $(j - 1)$ are candidates for window $j$. Since each window rejects at most $M$ non-zeros (when full), and the overlap is exactly $M$ positions, all rejected non-zeros are covered by the next window.

*Conclusion:* By induction, all $Z$ non-zeros are assigned to some window, completing the proof. $\square$

### C.1.5. CASE ANALYSIS: 2:4 HARDWARE (CURRENT)

NVIDIA's 2:4 Sparse Tensor Cores ($\alpha = 2\times$, stride 2) are the focus of SlideSparse. They efficiently support the $(2N-2) : 2N$ family:

| Pattern | N | Density | $\gamma$ | $S_{\text{eff}}$ | Achieves L/Z? |
|---------|---|---------|----------|------------------|---------------|
| 4:6 | 3 | 66.7% | 1.33 | 1.50× | Yes |
| 6:8 | 4 | 75.0% | 1.50 | 1.33× | Yes |
| 8:10 | 5 | 80.0% | 1.60 | 1.25× | Yes |
| 10:12 | 6 | 83.3% | 1.67 | 1.20× | Yes |
| 14:16 | 8 | 87.5% | 1.75 | 1.14× | Yes |

**Key observation**: For the $(2N-2) : 2N$ family, 2:4 hardware achieves the theoretical speedup limit $S_{\text{eff}} = L/Z$. This is because $\gamma = 2 - 2/N$ yields $\alpha/\gamma = 2/(2 - 2/N) = N/(N - 1) = L/Z$ exactly. This result generalizes Corollary 1.2 (§C) from the specific $(2N-2) : 2N$ case to the density-determined bound.

C.1.6. THEORETICAL SPEEDUP LIMIT (UPPER-BOUND)

**Theorem 3 (Density-Determined Speedup Limit).** For any $Z$:$L$ sparsity pattern accelerated via sliding window decomposition on $M$:$N$ hardware, the effective speedup is bounded by:

$$S_{\text{eff}} \leq \frac{L}{Z} = \frac{1}{\text{density}} \tag{16}$$

*Proof.* Let $\alpha = N/M$ be the hardware speedup. The minimum number of windows is $w_{\min} = \lceil Z/M \rceil$, giving $\gamma_{\min} = w_{\min} \cdot N/L \geq ZN/(ML)$. Thus $S_{\text{eff}} = \alpha/\gamma \leq (N/M) \cdot (ML)/(ZN) = L/Z$. $\square$

**Corollary 3.1.** The speedup upper bound depends *only* on density $Z/L$, not on the hardware's $M$:$N$ ratio.

**Practical Implication.** This theorem enables developers to evaluate new sparsity patterns against any $M$:$N$ hardware *without running experiments*: simply compute $S_{\text{eff}} \leq L/Z$ and check if the target hardware can achieve it. For example, a 70% sparse pattern ($Z$:$L$ = 7:10) can achieve at most $1.43\times$ speedup on *any* hardware—if 2:4 cores reach this bound, more advanced hardware offers no additional benefit for this pattern.

C.1.7. ACHIEVING THE BOUND: 1:4 HARDWARE

Interestingly, we find that hypothetical 1:4 hardware achieves the density-determined bound for *any* $Z$:$L$ pattern.

**1:4 Hardware Properties:**

- Hardware speedup: $\alpha = 4/1 = 4\times$
- Stride: $s = N - M = 4 - 1 = 3$ (overlap by 1 element)
- Each window accepts at most 1 non-zero

**Analysis.** For any $Z$:$L$ pattern, we need exactly $Z$ windows (one per non-zero), so $\gamma = 4Z/L$ and:

$$S_{\text{eff}} = \frac{\alpha}{\gamma} = \frac{4}{4Z/L} = \frac{L}{Z} \tag{17}$$

This exactly matches the theoretical limit from Theorem C.1.6.

**Conclusion.** 1:4 hardware is *optimal* in the sense of achieving the density-determined bound universally—it can accelerate *any* sparsity pattern to its theoretical maximum. This makes 1:4 a compelling target for future Sparse Tensor Core designs.

# D. Comprehensive Experimental Evaluation

This section provides a thorough experimental evaluation of SlideSparse, extending the results presented in §5. We report kernel-level GEMM performance across square and model-specific matrix dimensions, end-to-end inference throughput for both prefill and decode stages, and a novel *algorithmic efficiency* analysis that isolates SlideSparse's implementation quality from baseline variations. All speedup values are measured relative to the dense cuBLASLt baseline unless otherwise noted.

## D.1. Experimental Setup Overview

We provide a brief summary of the experimental configuration; full hardware and software specifications are detailed in Appendix E.

**Hardware Platforms.** Our evaluation spans **six NVIDIA GPU platforms** across four architecture generations:

- **Datacenter:** A100 80GB (Ampere, sm80), H100 80GB (Hopper, sm90), B200 180GB (Blackwell, sm100)

- **Consumer:** RTX 4090 24GB (Ada Lovelace, sm89), RTX 5080 16GB (Blackwell, sm120)

- **Embedded:** DGX Spark GB10 128GB (Blackwell, sm121, aarch64)

This selection covers both x86_64 and aarch64 host architectures, HBM and GDDR memory types, and platforms ranging from consumer workstations to datacenter servers.

**Data Precisions.** Kernel-level benchmarks cover **five precision types**: FP16, BF16, FP8 (E4M3), INT8, and FP4 (E2M1). End-to-end inference focuses on **INT8** and **FP8**, which represent the most practical quantized inference scenarios for production LLM deployment.

**Sparsity Configurations.** We evaluate the $(2N-2) : 2N$ sparsity family enabled by SlideSparse:

| Pattern | Density | Expansion $\gamma$ | Theoretical $S_{eff}$ | Note |
|---------|---------|--------------------|----------------------|------|
| 2:4 (native) | 50.0% | $1.00\times$ | $2.00\times$ | cuSPARSELt baseline |
| 4:6 | 66.7% | $1.33\times$ | $1.50\times$ | SlideSparse |
| 6:8 | 75.0% | $1.50\times$ | $1.33\times$ | SlideSparse |
| 8:10 | 80.0% | $1.60\times$ | $1.25\times$ | SlideSparse |

**Models and Workloads.** We evaluate **five model architectures**: Llama-3.2-1B, Llama-3.2-3B, Qwen-2.5-7B, Qwen-2.5-14B, and BitNet-2B (ternary 1.58-bit). Workloads span both **prefill** (compute-bound, $M=512-65536$) and **decode** (memory-bound, $M=64-512$) inference stages.

**Benchmark Methodology.**

- **Kernel benchmarks:** 25 warmup iterations followed by 100 measurement runs; we report mean latency.

- **End-to-end benchmarks:** Prefill uses $N=128$ iterations with `output_len`=1 to minimize decoding; Decode uses $N=256$ iterations per request with 16-token prompts for minial prefilling.

- **Algorithm optimization:** Both cuBLASLt and cuSPARSELt undergo exhaustive algorithm search to ensure fair comparison.

### D.2. Fused Kernel Efficiency

This section verifies that the fused quantization-slide kernel introduces negligible overhead compared to sparse GEMM.

**Latency Breakdown.** Table 1 compares the latency of standard per-token quantization (baseline) versus the fused quant+slide kernel for 6:8 sparsity ($\gamma = 1.5$) across representative $M$ values used in our experiments (§5).

*Table 1.* Fused kernel latency ($\mu$s) for 6:8 sparsity. Overhead is relative to quant-only baseline.

| GPU | M | Quant-only | Quant+Slide | Overhead | $\Delta$ ($\mu$s) |
|-----|------|-----------|-------------|----------|------------------|
| A100 | 2048 | 18.4 | 27.7 | +50% | 9.3 |
| A100 | 4096 | 30.7 | 44.0 | +43% | 13.3 |
| A100 | 8192 | 60.4 | 77.8 | +29% | 17.4 |
| A100 | 16384 | 109.6 | 141.3 | +29% | 31.7 |
| H100 | 4096 | 26.6 | 34.7 | +30% | 8.1 |
| H100 | 8192 | 47.3 | 64.2 | +36% | 16.9 |
| H100 | 16384 | 95.7 | 119.3 | +25% | 23.6 |
| B200 | 4096 | 18.4 | 25.5 | +39% | 7.1 |
| B200 | 8192 | 27.4 | 40.8 | +49% | 13.4 |
| B200 | 16384 | 46.2 | 70.6 | +53% | 24.4 |

**Comparison with GEMM.** The key observation is that sparse GEMM dominates end-to-end latency. The absolute slide overhead ($\Delta$) ranges from 7–32 $\mu$s, which is two orders of magnitude smaller than typical GEMM latencies at these matrix sizes. Even under the most conservative estimate where GEMM takes only 1 ms, the slide overhead remains below 3%. This confirms that the slide expansion adds minimal cost relative to the compute-bound GEMM, and the end-to-end speedup is dominated by sparse GEMM acceleration.

### D.3. Kernel-Level Performance Evaluation

Kernel-level benchmarks isolate raw GEMM performance from end-to-end inference overhead, providing insights into the theoretical speedup achievable at the computational level. We employ two complementary benchmark modes:

- **Square Mode** ($M=N=K$): Tests standardized matrix dimensions for systematic hardware characterization.
- **Model Mode**: Tests actual $(N, K)$ dimensions extracted from target model linear layers (Wqkv, Wo, W13, W2), directly reflecting real-world GEMM shapes.

**K Dimension Adjustment.** For SlideSparse configurations, the $K$ dimension is expanded according to the sliding window transformation: 2:4 uses the original $K$; 4:6 expands to $K' = 1.33K$; 6:8 expands to $K' = 1.50K$; 8:10 expands to $K' = 1.67K$. This expansion is the computational cost paid to enable hardware acceleration.

### D.3.1. SQUARE MATRIX KERNEL RESULTS

The following tables present kernel speedup across all tested GPUs and precisions for square matrices ($M=N=K$).

### Square Kernel (FP4)

| GPU | M | cuBLASLt Latency(μs) | Speedup Ratio under Different Sparsity | | | | | | | |
|---|---|---|---|---|---|---|---|---|---|---|
| | | | 2:4 | 4:6 | 6:8 | 8:10 | 10:12 | 12:14 | 14:16 | ∞:∞ |
| B200 | 64 | 8.42e+00 | 1.37 | 1.39 | 1.36 | 1.51 | 1.39 | 1.36 | 1.38 | 1.38 |
| | 128 | 8.47e+00 | 1.37 | 1.36 | 1.36 | 1.51 | 1.37 | 1.36 | 1.36 | 1.37 |
| | 256 | 8.48e+00 | 1.37 | 1.36 | 1.36 | 1.37 | 1.36 | 1.36 | 1.36 | 1.36 |
| | 512 | 8.47e+00 | 1.36 | 1.35 | 1.36 | 1.35 | 1.35 | 1.36 | 1.35 | 1.35 |
| | 1024 | 8.46e+00 | 1.35 | 1.20 | 1.22 | 1.09 | 1.07 | 1.11 | 1.10 | 1.03 |
| | 2048 | 8.70e+00 | 0.84 | 0.82 | 0.84 | 0.84 | 0.83 | 0.70 | 0.84 | 0.84 |
| | 4096 | 1.86e+01 | 0.81 | 0.65 | 0.65 | 0.53 | 0.53 | 0.59 | 0.60 | 0.53 |
| | 8192 | 9.31e+01 | 0.81 | 0.54 | 0.57 | 0.46 | 0.48 | 0.50 | 0.49 | 0.42 |
| | 16384 | 6.83e+02 | 0.75 | 0.54 | 0.50 | 0.43 | 0.39 | 0.37 | 0.40 | 0.35 |
| RTX5080 | 64 | 4.20e+00 | 1.03 | 1.02 | 1.02 | 1.02 | 1.02 | 1.02 | 1.02 | 1.02 |
| | 128 | 4.17e+00 | 1.01 | 1.02 | 1.01 | 1.02 | 1.02 | 1.01 | 1.02 | 1.01 |
| | 256 | 4.19e+00 | 1.02 | 1.02 | 1.01 | 1.01 | 1.01 | 1.01 | 1.01 | 1.01 |
| | 512 | 4.21e+00 | 0.69 | 0.69 | 0.69 | 0.68 | 0.68 | 0.68 | 0.69 | 0.68 |
| | 1024 | 6.20e+00 | 1.01 | 0.76 | 0.76 | 0.76 | 0.76 | 0.76 | 0.76 | 0.76 |
| GB10 | 64 | 6.17e+00 | 1.05 | 1.08 | 1.05 | 1.06 | 1.07 | 1.10 | 1.09 | 1.12 |
| | 128 | 6.20e+00 | 1.07 | 1.00 | 1.02 | 1.01 | 1.06 | 1.01 | 1.07 | 1.07 |
| | 256 | 6.82e+00 | 1.11 | 1.11 | 1.10 | 1.11 | 1.10 | 1.11 | 1.11 | 1.10 |
| | 512 | 6.21e+00 | 1.01 | 1.00 | 0.98 | 0.77 | 0.77 | 0.77 | 0.79 | 0.77 |
| | 1024 | 1.03e+01 | 0.84 | 0.64 | 0.71 | 0.63 | 0.56 | 0.63 | 0.63 | 0.63 |
| | 2048 | 3.29e+01 | 0.76 | 0.48 | 0.55 | 0.50 | 0.46 | 0.40 | 0.48 | 0.42 |
| | 4096 | 2.41e+02 | 0.78 | 0.54 | 0.55 | 0.42 | 0.41 | 0.44 | 0.49 | 0.43 |
| | 8192 | 1.70e+03 | 0.73 | 0.48 | 0.50 | 0.40 | 0.44 | 0.44 | 0.43 | 0.38 |

### Square Kernel (INT8)

| GPU | M | cuBLASLt Latency(μs) | Speedup Ratio under Different Sparsity | | | | | | | |
|---|---|---|---|---|---|---|---|---|---|---|
| | | | 2:4 | 4:6 | 6:8 | 8:10 | 10:12 | 12:14 | 14:16 | ∞:∞ |
| A100 | 64 | 5.57e+00 | 1.04 | 1.01 | 1.03 | 1.02 | 1.02 | 0.97 | 1.03 | 1.02 |
| | 128 | 5.86e+00 | 1.04 | 0.98 | 1.00 | 0.95 | 0.97 | 0.97 | 0.98 | 0.99 |
| | 256 | 6.14e+00 | 1.05 | 0.93 | 0.98 | 0.88 | 0.91 | 0.92 | 0.92 | 0.92 |
| | 512 | 7.11e+00 | 1.05 | 0.94 | 0.97 | 0.89 | 0.84 | 0.93 | 0.93 | 0.89 |
| | 1024 | 1.36e+01 | 1.18 | 0.95 | 1.00 | 0.95 | 0.92 | 0.86 | 0.93 | 0.88 |
| | 2048 | 5.54e+01 | 1.42 | 1.25 | 1.06 | 0.96 | 0.93 | 1.04 | 0.94 | 0.85 |
| | 4096 | 3.47e+02 | 2.06 | 1.37 | 1.44 | 1.13 | 1.25 | 1.23 | 1.19 | 1.08 |
| | 8192 | 3.05e+03 | 2.19 | 1.58 | 1.44 | 1.29 | 1.11 | 1.07 | 1.23 | 1.08 |
| | 16384 | 2.51e+04 | 2.18 | 1.37 | 1.46 | 1.36 | 1.22 | 1.23 | 1.22 | 0.91 |
| RTX4090 | 64 | 9.52e+00 | 1.05 | 1.09 | 1.06 | 0.99 | 0.38 | 0.42 | 0.35 | 0.35 |
| | 128 | 9.91e+00 | 0.48 | 1.13 | 1.00 | 1.17 | 0.31 | 0.31 | 0.31 | 0.32 |
| | 256 | 1.01e+01 | 1.04 | 1.11 | 1.05 | 1.08 | 0.27 | 0.27 | 0.27 | 0.26 |
| | 512 | 1.06e+01 | 1.21 | 1.14 | 1.08 | 1.12 | 0.20 | 0.21 | 0.21 | 0.20 |
| | 1024 | 1.16e+01 | 1.02 | 0.92 | 0.99 | 0.91 | 0.13 | 0.11 | 0.12 | 0.11 |
| | 2048 | 5.35e+01 | 2.06 | 1.74 | 1.49 | 1.24 | 0.14 | 0.16 | 0.15 | 0.14 |
| | 4096 | 2.36e+02 | 1.49 | 0.96 | 1.06 | 0.80 | 0.10 | 0.51 | 0.67 | 0.59 |
| | 8192 | 1.94e+03 | 1.60 | 1.21 | 1.07 | 1.01 | 0.79 | 0.77 | 0.94 | 0.82 |
| | 16384 | 1.53e+04 | 1.59 | 0.95 | 1.04 | 0.98 | 0.91 | 0.88 | 0.89 | 0.76 |
| H100 | 64 | 4.41e+00 | 0.87 | 0.88 | 0.87 | 0.88 | 0.85 | 0.84 | 0.85 | 0.84 |
| | 128 | 4.44e+00 | 0.86 | 0.85 | 0.84 | 0.84 | 0.81 | 0.81 | 0.81 | 0.83 |
| | 256 | 4.86e+00 | 0.93 | 0.87 | 0.89 | 0.87 | 0.86 | 0.85 | 0.86 | 0.64 |
| | 512 | 5.77e+00 | 0.96 | 0.97 | 0.97 | 0.95 | 0.90 | 0.93 | 0.93 | 0.91 |
| | 1024 | 1.12e+01 | 1.30 | 1.05 | 1.16 | 1.12 | 1.05 | 0.98 | 1.09 | 1.04 |
| | 2048 | 3.13e+01 | 1.58 | 1.25 | 1.17 | 1.08 | 1.05 | 1.08 | 1.10 | 1.00 |
| | 4096 | 1.36e+02 | 1.28 | 0.98 | 0.94 | 0.85 | 0.87 | 0.83 | 0.79 | 0.73 |
| | 8192 | 1.26e+03 | 1.71 | 1.24 | 0.98 | 1.00 | 0.90 | 0.80 | 0.91 | 0.75 |
| | 16384 | 1.25e+04 | 1.79 | 1.18 | 1.33 | 1.03 | 1.12 | 1.11 | 1.07 | 0.94 |
| B200 | 64 | 4.79e+00 | 0.77 | 0.78 | 0.77 | 0.77 | 0.77 | 0.77 | 0.77 | 0.77 |
| | 128 | 4.85e+00 | 0.78 | 0.78 | 0.78 | 0.78 | 0.78 | 0.78 | 0.78 | 0.78 |
| | 256 | 4.84e+00 | 0.78 | 0.81 | 0.78 | 0.78 | 0.78 | 0.78 | 0.78 | 0.79 |
| | 512 | 6.25e+00 | 1.01 | 1.01 | 1.01 | 1.01 | 1.00 | 1.01 | 1.00 | 1.01 |
| | 1024 | 8.31e+00 | 1.34 | 1.23 | 1.33 | 1.33 | 1.19 | 1.01 | 1.29 | 1.15 |
| | 2048 | 2.74e+01 | 2.65 | 2.21 | 2.22 | 1.97 | 1.90 | 2.21 | 2.21 | 2.18 |
| | 4096 | 1.54e+02 | 5.34 | 3.75 | 3.98 | 3.26 | 3.41 | 3.50 | 3.43 | 3.01 |
| | 8192 | 1.18e+03 | 6.47 | 4.46 | 4.31 | 3.62 | 3.21 | 3.11 | 3.56 | 3.09 |
| | 16384 | 9.67e+03 | 6.11 | 3.83 | 3.82 | 3.57 | 3.13 | 3.12 | 3.21 | 2.73 |
| RTX5080 | 64 | 4.16e+00 | 1.02 | 1.01 | 1.02 | 1.03 | 1.01 | 1.02 | 1.01 | 1.02 |
| | 128 | 4.15e+00 | 1.03 | 0.76 | 1.01 | 1.01 | 1.01 | 1.01 | 1.01 | 1.18 |
| | 256 | 4.17e+00 | 1.01 | 1.02 | 1.02 | 1.01 | 1.01 | 0.89 | 1.01 | 1.01 |
| | 512 | 5.94e+00 | 1.46 | 1.09 | 1.44 | 1.43 | 0.96 | 1.02 | 1.01 | 0.97 |
| | 1024 | 1.23e+01 | 1.51 | 1.49 | 1.50 | 1.21 | 1.21 | 1.20 | 1.20 | 1.20 |
| | 2048 | 5.95e+01 | 1.61 | 1.25 | 1.16 | 0.98 | 0.94 | 1.00 | 1.00 | 0.91 |
| | 4096 | 3.55e+02 | 1.63 | 1.12 | 1.11 | 0.94 | 0.99 | 0.97 | 0.96 | 0.85 |
| | 8192 | 2.60e+03 | 1.61 | 1.18 | 1.04 | 0.98 | 0.82 | 0.80 | 0.90 | 0.80 |
| | 16384 | 2.07e+04 | 1.57 | 1.03 | 1.04 | 0.98 | 0.93 | 0.90 | 0.90 | 0.80 |
| GB10 | 64 | 4.18e+00 | 1.00 | 1.02 | 0.94 | 1.00 | 1.04 | 1.01 | 1.00 | 1.00 |
| | 128 | 4.21e+00 | 0.97 | 0.98 | 1.01 | 1.00 | 1.01 | 1.00 | 1.01 | 0.87 |
| | 256 | 4.95e+00 | 1.19 | 1.15 | 0.77 | 0.83 | 0.92 | 0.92 | 0.90 | 0.92 |
| | 512 | 6.26e+00 | 1.01 | 1.01 | 0.84 | 1.01 | 0.90 | 0.93 | 1.01 | 1.01 |
| | 1024 | 2.26e+01 | 1.37 | 1.09 | 1.05 | 1.06 | 1.01 | 0.93 | 1.00 | 0.92 |
| | 2048 | 9.97e+01 | 1.23 | 1.06 | 0.95 | 0.77 | 0.65 | 0.87 | 0.84 | 0.74 |

### Square Kernel (INT8) (cont.)

| GPU | M | cuBLASLt Latency(μs) | Speedup Ratio under Different Sparsity | | | | | | | |
|---|---|---|---|---|---|---|---|---|---|---|
| | | | 2:4 | 4:6 | 6:8 | 8:10 | 10:12 | 12:14 | 14:16 | ∞:∞ |
| | 4096 | 7.70e+02 | 1.52 | 1.01 | 1.00 | 0.83 | 0.90 | 0.86 | 0.88 | 0.75 |
| | 8192 | 6.04e+03 | 1.46 | 1.09 | 0.99 | 0.88 | 0.74 | 0.72 | 0.84 | 0.70 |
| | 16384 | 5.18e+04 | 1.55 | 0.75 | 0.78 | 0.72 | 0.63 | 0.61 | 0.63 | 0.49 |

## Square Kernel (FP8)

| GPU | M | cuBLASLt Latency(μs) | Speedup Ratio under Different Sparsity | | | | | | | |
|---|---|---|---|---|---|---|---|---|---|---|
| | | | 2:4 | 4:6 | 6:8 | 8:10 | 10:12 | 12:14 | 14:16 | ∞:∞ |
| RTX4090 | 64 | 1.13e+01 | 1.12 | 1.18 | 1.06 | 1.22 | 0.40 | 0.40 | 0.65 | 0.40 |
| | 128 | 1.17e+01 | 1.30 | 1.36 | 1.11 | 1.31 | 0.33 | 0.33 | 1.23 | 0.34 |
| | 256 | 1.10e+01 | 1.20 | 1.10 | 1.22 | 1.07 | 0.25 | 0.25 | 1.14 | 0.25 |
| | 512 | 1.22e+01 | 1.31 | 1.16 | 1.21 | 1.27 | 0.20 | 0.21 | 1.08 | 0.19 |
| | 1024 | 1.24e+01 | 1.03 | 0.90 | 1.01 | 0.97 | 0.12 | 0.11 | 0.24 | 0.11 |
| | 2048 | 5.78e+01 | 1.75 | 1.36 | 1.26 | 1.14 | 0.10 | 0.10 | 0.10 | 0.12 |
| | 4096 | 4.20e+02 | 1.87 | 1.40 | 1.27 | 1.18 | 0.71 | 1.10 | 1.08 | 0.95 |
| | 8192 | 3.42e+03 | 1.99 | 1.50 | 1.36 | 1.26 | 1.20 | 1.13 | 1.15 | 1.03 |
| | 16384 | 2.84e+04 | 2.08 | 1.51 | 1.37 | 1.28 | 1.22 | 1.20 | 1.18 | 1.04 |
| H100 | 64 | 4.61e+00 | 0.95 | 0.89 | 0.91 | 0.93 | 0.90 | 0.90 | 0.89 | 0.90 |
| | 128 | 4.70e+00 | 0.93 | 0.90 | 0.90 | 0.89 | 0.47 | 0.85 | 0.85 | 0.87 |
| | 256 | 4.61e+00 | 0.91 | 0.83 | 0.85 | 0.82 | 0.82 | 0.82 | 0.81 | 0.83 |
| | 512 | 4.62e+00 | 0.82 | 0.77 | 0.76 | 0.75 | 0.72 | 0.75 | 0.75 | 0.74 |
| | 1024 | 6.36e+00 | 0.73 | 0.61 | 0.65 | 0.62 | 0.59 | 0.57 | 0.62 | 0.60 |
| | 2048 | 2.12e+01 | 1.07 | 0.84 | 0.80 | 0.72 | 0.71 | 0.75 | 0.74 | 0.68 |
| | 4096 | 1.56e+02 | 1.53 | 1.10 | 0.94 | 0.86 | 0.84 | 0.82 | 0.83 | 0.73 |
| | 8192 | 1.41e+03 | 1.54 | 1.15 | 1.05 | 0.85 | 0.93 | 0.86 | 0.92 | 0.79 |
| | 16384 | 1.28e+04 | 1.73 | 1.12 | 1.08 | 1.02 | 1.08 | 1.02 | 1.03 | 0.91 |
| B200 | 64 | 5.97e+00 | 0.96 | 0.96 | 0.97 | 0.97 | 0.96 | 0.97 | 0.96 | 0.96 |
| | 128 | 5.58e+00 | 0.90 | 0.90 | 0.90 | 0.90 | 0.90 | 0.90 | 0.92 | 0.90 |
| | 256 | 5.67e+00 | 0.91 | 0.91 | 0.91 | 0.91 | 0.91 | 0.91 | 0.93 | 0.91 |
| | 512 | 5.64e+00 | 0.91 | 0.91 | 0.91 | 0.91 | 0.91 | 0.91 | 0.91 | 0.91 |
| | 1024 | 5.64e+00 | 0.91 | 0.79 | 0.90 | 0.87 | 0.73 | 0.68 | 0.83 | 0.74 |
| | 2048 | 1.04e+01 | 1.00 | 0.84 | 0.84 | 0.84 | 0.75 | 0.84 | 0.84 | 0.84 |
| | 4096 | 4.55e+01 | 1.51 | 1.06 | 1.10 | 0.90 | 0.95 | 0.96 | 0.96 | 0.83 |
| | 8192 | 3.40e+02 | 1.72 | 1.14 | 1.16 | 0.97 | 0.88 | 0.82 | 1.00 | 0.86 |
| | 16384 | 3.03e+03 | 1.85 | 1.07 | 1.07 | 1.00 | 0.91 | 0.88 | 0.91 | 0.79 |
| RTX5080 | 64 | 3.34e+00 | 0.81 | 0.81 | 0.80 | 0.81 | 0.81 | 0.81 | 0.81 | 0.80 |
| | 128 | 3.32e+00 | 0.80 | 0.80 | 0.81 | 0.80 | 0.80 | 0.80 | 0.80 | 0.80 |
| | 256 | 3.37e+00 | 0.82 | 0.55 | 0.55 | 0.55 | 0.55 | 0.55 | 0.55 | 0.54 |
| | 512 | 4.19e+00 | 0.68 | 0.68 | 0.68 | 0.68 | 0.66 | 0.67 | 0.68 | 0.65 |
| | 1024 | 1.44e+01 | 1.40 | 1.00 | 1.00 | 0.88 | 0.89 | 0.88 | 0.88 | 0.81 |
| | 2048 | 7.99e+01 | 1.56 | 1.17 | 1.08 | 0.93 | 0.87 | 0.93 | 0.95 | 0.85 |
| | 4096 | 5.92e+02 | 1.83 | 1.31 | 1.20 | 1.11 | 1.07 | 1.03 | 1.03 | 0.91 |
| | 8192 | 4.56e+03 | 1.76 | 1.33 | 1.17 | 1.10 | 1.05 | 1.03 | 1.01 | 0.88 |
| | 16384 | 3.64e+04 | 1.74 | 1.31 | 1.17 | 1.10 | 1.07 | 1.03 | 1.00 | 0.88 |
| GB10 | 64 | 5.16e+00 | 0.96 | 0.95 | 0.93 | 0.96 | 0.96 | 0.96 | 0.95 | 0.97 |
| | 128 | 5.03e+00 | 1.00 | 1.03 | 1.03 | 0.98 | 0.96 | 0.97 | 0.97 | 1.04 |
| | 256 | 3.86e+00 | 0.75 | 0.63 | 0.63 | 0.63 | 0.62 | 0.63 | 0.62 | 0.63 |
| | 512 | 6.23e+00 | 0.98 | 0.76 | 0.77 | 0.76 | 0.75 | 0.75 | 0.76 | 0.75 |
| | 1024 | 1.98e+01 | 1.07 | 0.88 | 0.80 | 0.75 | 0.76 | 0.75 | 0.74 | 0.64 |
| | 2048 | 1.02e+02 | 1.19 | 0.95 | 0.84 | 0.81 | 0.74 | 0.76 | 0.73 | 0.65 |
| | 4096 | 7.32e+02 | 1.21 | 0.92 | 0.84 | 0.79 | 0.77 | 0.73 | 0.71 | 0.63 |
| | 8192 | 6.01e+03 | 1.26 | 0.97 | 0.85 | 0.80 | 0.75 | 0.75 | 0.73 | 0.64 |
| | 16384 | 5.37e+04 | 1.41 | 0.89 | 0.90 | 0.83 | 0.73 | 0.72 | 0.76 | 0.66 |

## Square Kernel (FP16)

| GPU | M | cuBLASLt Latency(μs) | Speedup Ratio under Different Sparsity | | | | | | | |
|---|---|---|---|---|---|---|---|---|---|---|
| | | | 2:4 | 4:6 | 6:8 | 8:10 | 10:12 | 12:14 | 14:16 | ∞:∞ |
| A100 | 64 | 4.01e+00 | 0.70 | 0.64 | 0.65 | 0.66 | 0.65 | 0.65 | 0.66 | 0.65 |
| | 128 | 4.28e+00 | 0.69 | 0.64 | 0.64 | 0.61 | 0.61 | 0.61 | 0.61 | 0.61 |
| | 256 | 5.20e+00 | 0.76 | 0.67 | 0.68 | 0.65 | 0.65 | 0.66 | 0.66 | 0.63 |
| | 512 | 7.17e+00 | 0.87 | 0.77 | 0.74 | 0.72 | 0.67 | 0.70 | 0.70 | 0.66 |
| | 1024 | 1.98e+01 | 1.24 | 1.04 | 1.01 | 0.97 | 0.95 | 0.91 | 0.93 | 0.86 |
| | 2048 | 7.45e+01 | 1.08 | 0.85 | 0.83 | 0.78 | 0.76 | 0.73 | 0.73 | 0.56 |
| | 4096 | 5.90e+02 | 1.81 | 1.24 | 1.17 | 1.16 | 1.07 | 1.00 | 0.95 | 0.83 |
| | 8192 | 4.68e+03 | 1.52 | 1.00 | 1.03 | 0.97 | 0.82 | 0.81 | 0.81 | 0.68 |
| | 16384 | 3.74e+04 | 1.22 | 0.91 | 0.84 | 0.79 | 0.77 | 0.72 | 0.71 | 0.62 |
| RTX4090 | 64 | 9.44e+00 | 1.00 | 1.01 | 0.99 | 1.08 | 0.90 | 1.01 | 1.01 | 0.24 |
| | 128 | 9.32e+00 | 1.01 | 1.00 | 0.90 | 0.92 | 0.24 | 0.74 | 0.87 | 0.19 |
| | 256 | 9.20e+00 | 0.88 | 0.90 | 0.89 | 0.80 | 0.14 | 0.36 | 0.84 | 0.15 |
| | 512 | 1.03e+01 | 0.97 | 0.89 | 0.84 | 0.84 | 0.10 | 0.10 | 0.10 | 0.17 |
| | 1024 | 1.96e+01 | 1.28 | 1.18 | 0.98 | 0.88 | 0.11 | 0.11 | 0.11 | 0.84 |
| | 2048 | 1.15e+02 | 1.86 | 1.44 | 1.30 | 1.21 | 0.10 | 0.10 | 0.11 | 1.06 |
| | 4096 | 8.18e+02 | 1.83 | 1.39 | 1.25 | 1.19 | 1.12 | 0.85 | 1.13 | 0.99 |
| | 8192 | 6.64e+03 | 1.95 | 1.47 | 1.31 | 1.23 | 1.17 | 1.15 | 1.14 | 0.99 |
| | 16384 | 5.52e+04 | 1.90 | 1.43 | 1.27 | 1.19 | 1.17 | 1.13 | 1.10 | 0.97 |
| H100 | 64 | 4.47e+00 | | | | | | | | |
| | 128 | 4.48e+00 | | | | | | | | |

### Square Kernel (FP16) (cont.)

| GPU | M | cuBLASLt Latency(μs) | 2:4 | 4:6 | 6:8 | 8:10 | 10:12 | 12:14 | 14:16 | ∞:∞ |
|---|---|---|---|---|---|---|---|---|---|---|
| | 256 | 4.49e+00 | | | | | | | | |
| | 512 | 4.75e+00 | | | | | | | | |
| | 1024 | 7.77e+00 | | | | | | | | |
| | 2048 | 3.59e+01 | | | | | | | | |
| | 4096 | 2.81e+02 | | | | | | | | |
| | 8192 | 2.48e+03 | | | | | | | | |
| | 16384 | 2.21e+04 | | | | | | | | |
| B200 | 64 | 5.61e+00 | 0.90 | 0.91 | 0.91 | 0.91 | 0.90 | 0.91 | 0.91 | 0.91 |
| | 128 | 5.25e+00 | 0.84 | 0.85 | 0.84 | 0.84 | 0.85 | 0.85 | 0.84 | 0.84 |
| | 256 | 5.29e+00 | 0.85 | 0.85 | 0.86 | 0.85 | 0.85 | 0.85 | 0.85 | 0.85 |
| | 512 | 5.25e+00 | 0.84 | 0.84 | 0.84 | 0.84 | 0.84 | 0.84 | 0.84 | 0.84 |
| | 1024 | 6.28e+00 | 0.86 | 0.76 | 0.76 | 0.76 | 0.76 | 0.76 | 0.76 | 0.76 |
| | 2048 | 1.65e+01 | 1.27 | 1.14 | 1.00 | 1.00 | 1.00 | 1.00 | 1.00 | 0.89 |
| | 4096 | 8.98e+01 | 1.62 | 1.14 | 1.08 | 0.94 | 0.94 | 0.92 | 0.93 | 0.81 |
| | 8192 | 6.54e+02 | 1.60 | 1.18 | 1.06 | 0.98 | 0.92 | 0.85 | 0.86 | 0.82 |
| | 16384 | 5.95e+03 | 1.63 | 1.16 | 1.09 | 1.03 | 0.98 | 0.95 | 0.94 | 0.83 |
| RTX5080 | 64 | 2.12e+00 | 0.52 | 0.52 | 0.51 | 0.52 | 0.52 | 0.54 | 0.52 | 0.52 |
| | 128 | 2.30e+00 | 0.56 | 0.56 | 0.55 | 0.37 | 0.37 | 0.37 | 0.37 | 0.38 |
| | 256 | 4.06e+00 | 0.66 | 0.66 | 0.66 | 0.66 | 0.66 | 0.66 | 0.66 | 0.66 |
| | 512 | 6.13e+00 | 0.99 | 0.75 | 0.74 | 0.74 | 0.74 | 0.73 | 0.71 | 0.60 |
| | 1024 | 2.67e+01 | 1.63 | 1.30 | 1.18 | 1.18 | 1.08 | 1.08 | 1.08 | 0.98 |
| | 2048 | 1.54e+02 | 1.53 | 1.17 | 1.06 | 0.99 | 0.96 | 0.94 | 0.92 | 0.81 |
| | 4096 | 1.15e+03 | 1.81 | 1.37 | 1.23 | 1.15 | 1.10 | 1.08 | 1.06 | 0.93 |
| | 8192 | 9.05e+03 | 1.81 | 1.34 | 1.16 | 1.13 | 1.06 | 1.04 | 1.01 | 0.92 |
| | 16384 | 7.27e+04 | 1.53 | 1.08 | 0.98 | 0.93 | 0.89 | 0.88 | 0.85 | 0.75 |
| GB10 | 64 | 3.45e+00 | 0.82 | 0.79 | 0.77 | 0.77 | 0.78 | 0.79 | 0.79 | 0.79 |
| | 128 | 3.21e+00 | 0.69 | 0.52 | 0.47 | 0.48 | 0.50 | 0.52 | 0.50 | 0.52 |
| | 256 | 4.18e+00 | 0.61 | 0.65 | 0.67 | 0.61 | 0.65 | 0.67 | 0.62 | 0.63 |
| | 512 | 8.54e+00 | 1.13 | 0.97 | 1.02 | 0.98 | 0.88 | 0.94 | 0.91 | 0.83 |
| | 1024 | 3.35e+01 | 1.25 | 1.01 | 0.96 | 0.86 | 0.85 | 0.84 | 0.82 | 0.75 |
| | 2048 | 2.18e+02 | 1.57 | 1.20 | 1.10 | 0.99 | 0.98 | 0.98 | 0.88 | 0.79 |
| | 4096 | 1.85e+03 | 1.61 | 0.67 | 0.69 | 0.51 | 0.57 | 0.58 | 0.58 | 0.48 |
| | 8192 | 1.36e+04 | 0.54 | 0.40 | 0.35 | 0.33 | 0.25 | 0.24 | 0.31 | 0.27 |
| | 16384 | 1.07e+05 | 0.53 | 0.30 | 0.35 | 0.33 | 0.32 | 0.31 | 0.30 | 0.26 |

### Square Kernel (BF16)

| GPU | M | cuBLASLt Latency(μs) | 2:4 | 4:6 | 6:8 | 8:10 | 10:12 | 12:14 | 14:16 | ∞:∞ |
|---|---|---|---|---|---|---|---|---|---|---|
| A100 | 64 | 4.32e+00 | 0.76 | 0.70 | 0.69 | 0.71 | 0.71 | 0.72 | 0.71 | 0.70 |
| | 128 | 4.57e+00 | 0.71 | 0.69 | 0.65 | 0.65 | 0.65 | 0.66 | 0.65 | 0.66 |
| | 256 | 5.80e+00 | 0.83 | 0.75 | 0.76 | 0.72 | 0.73 | 0.74 | 0.74 | 0.70 |
| | 512 | 7.58e+00 | 0.90 | 0.81 | 0.78 | 0.76 | 0.71 | 0.74 | 0.74 | 0.69 |
| | 1024 | 1.91e+01 | 1.19 | 1.01 | 0.97 | 0.93 | 0.91 | 0.88 | 0.90 | 0.83 |
| | 2048 | 8.21e+01 | 1.18 | 0.93 | 0.91 | 0.86 | 0.83 | 0.83 | 0.80 | 0.62 |
| | 4096 | 5.86e+02 | 1.71 | 1.25 | 1.12 | 1.01 | 0.99 | 0.96 | 0.95 | 0.82 |
| | 8192 | 4.76e+03 | 1.52 | 1.02 | 0.98 | 0.93 | 0.80 | 0.80 | 0.84 | 0.70 |
| | 16384 | 3.80e+04 | 1.22 | 0.91 | 0.81 | 0.77 | 0.74 | 0.72 | 0.71 | 0.62 |
| RTX4090 | 64 | 9.54e+00 | 0.99 | 0.90 | 0.92 | 0.92 | 0.39 | 0.94 | 0.43 | 0.97 |
| | 128 | 9.93e+00 | 1.15 | 1.08 | 1.09 | 1.00 | 0.20 | 0.32 | 0.20 | 0.20 |
| | 256 | 1.00e+01 | 1.03 | 0.91 | 0.98 | 0.95 | 0.15 | 0.15 | 0.15 | 0.19 |
| | 512 | 1.18e+01 | 1.14 | 1.00 | 0.96 | 0.95 | 0.12 | 0.12 | 0.12 | 0.13 |
| | 1024 | 1.92e+01 | 1.46 | 1.16 | 1.11 | 1.01 | 0.11 | 0.11 | 0.11 | 0.90 |
| | 2048 | 1.06e+02 | 1.72 | 1.33 | 1.20 | 1.13 | 0.09 | 0.09 | 0.12 | 0.97 |
| | 4096 | 8.24e+02 | 1.84 | 1.40 | 1.25 | 1.18 | 0.67 | 1.16 | 0.93 | 0.99 |
| | 8192 | 6.90e+03 | 2.01 | 1.51 | 1.33 | 1.25 | 1.22 | 1.18 | 1.17 | 1.00 |
| | 16384 | 5.73e+04 | 1.97 | 1.46 | 1.31 | 1.22 | 1.19 | 1.16 | 1.13 | 0.99 |
| H100 | 64 | 4.66e+00 | 0.80 | 0.76 | 0.76 | 0.78 | 0.74 | 0.75 | 0.74 | 0.74 |
| | 128 | 4.59e+00 | 0.69 | 0.71 | 0.71 | 0.68 | 0.66 | 0.65 | 0.66 | 0.66 |
| | 256 | 4.57e+00 | 0.67 | 0.61 | 0.62 | 0.61 | 0.59 | 0.59 | 0.59 | 0.58 |
| | 512 | 4.71e+00 | 0.60 | 0.56 | 0.54 | 0.53 | 0.50 | 0.51 | 0.51 | 0.49 |
| | 1024 | 7.80e+00 | 0.75 | 0.67 | 0.64 | 0.60 | 0.59 | 0.59 | 0.58 | 0.55 |
| | 2048 | 3.50e+01 | 0.97 | 0.78 | 0.72 | 0.65 | 0.63 | 0.62 | 0.62 | 0.54 |
| | 4096 | 2.89e+02 | 1.53 | 1.13 | 0.98 | 0.95 | 0.95 | 0.94 | 0.86 | 0.79 |
| | 8192 | 2.59e+03 | 1.59 | 1.18 | 1.05 | 0.95 | 0.93 | 0.92 | 0.90 | 0.79 |
| | 16384 | 2.23e+04 | 1.45 | 1.05 | 0.93 | 0.88 | 0.92 | 0.89 | 0.88 | 0.77 |
| B200 | 64 | 5.89e+00 | 1.15 | 0.95 | 0.95 | 0.95 | 0.96 | 1.13 | 1.13 | 0.95 |
| | 128 | 5.40e+00 | 0.87 | 0.87 | 0.87 | 0.87 | 0.87 | 0.87 | 0.92 | 0.87 |
| | 256 | 5.51e+00 | 0.89 | 0.89 | 0.89 | 0.89 | 0.88 | 0.89 | 0.89 | 0.89 |
| | 512 | 5.55e+00 | 0.89 | 0.89 | 0.89 | 0.89 | 0.89 | 0.89 | 0.89 | 0.89 |
| | 1024 | 6.09e+00 | 0.83 | 0.74 | 0.74 | 0.74 | 0.74 | 0.73 | 0.73 | 0.74 |
| | 2048 | 1.65e+01 | 1.28 | 1.14 | 1.00 | 1.00 | 0.99 | 1.00 | 1.00 | 0.89 |
| | 4096 | 9.09e+01 | 1.61 | 1.13 | 1.07 | 0.96 | 0.95 | 0.95 | 0.92 | 0.81 |
| | 8192 | 6.69e+02 | 1.64 | 1.20 | 1.06 | 1.00 | 0.92 | 0.88 | 0.93 | 0.83 |
| | 16384 | 5.97e+03 | 1.61 | 1.14 | 1.08 | 1.01 | 0.96 | 0.94 | 0.93 | 0.81 |
| RTX5080 | 64 | 2.13e+00 | 0.52 | 0.52 | 0.52 | 0.52 | 0.51 | 0.52 | 0.35 | 0.52 |
| | 128 | 2.41e+00 | 0.58 | 0.58 | 0.58 | 0.39 | 0.39 | 0.39 | 0.38 | 0.39 |
| | 256 | 4.06e+00 | 0.66 | 0.66 | 0.66 | 0.65 | 0.65 | 0.65 | 0.65 | 0.66 |

### *Square Kernel (BF16) (cont.)*

| GPU | M | cuBLASLt Latency(μs) | Speedup Ratio under Different Sparsity | | | | | | | |
|-----|-----|------|------|------|------|------|-------|-------|-------|------|
| | | | **2:4** | **4:6** | **6:8** | **8:10** | **10:12** | **12:14** | **14:16** | **∞:∞** |
| | 512 | 8.19e+00 | 1.32 | 1.00 | 0.99 | 0.99 | 0.95 | 0.90 | 0.78 | 0.80 |
| | 1024 | 2.67e+01 | 1.63 | 1.30 | 1.18 | 1.18 | 1.08 | 1.08 | 1.08 | 1.00 |
| | 2048 | 1.90e+02 | 1.89 | 1.45 | 1.31 | 1.22 | 1.18 | 1.16 | 1.15 | 1.01 |
| | 4096 | 1.22e+03 | 1.93 | 1.45 | 1.30 | 1.22 | 1.17 | 1.14 | 1.12 | 0.98 |
| | 8192 | 9.10e+03 | 1.81 | 1.34 | 1.17 | 1.13 | 1.06 | 1.04 | 1.00 | 0.92 |
| | 16384 | 7.28e+04 | 1.53 | 1.13 | 0.98 | 0.94 | 0.90 | 0.88 | 0.85 | 0.75 |
| GB10 | 64 | 3.03e+00 | 0.73 | 0.66 | 0.73 | 0.66 | 0.69 | 0.70 | 0.68 | 0.70 |
| | 128 | 4.29e+00 | 0.90 | 0.67 | 0.69 | 0.53 | 0.69 | 0.70 | 0.57 | 0.54 |
| | 256 | 5.30e+00 | 0.85 | 0.82 | 0.86 | 0.81 | 0.84 | 0.85 | 0.83 | 0.81 |
| | 512 | 1.24e+01 | 1.58 | 1.41 | 1.38 | 1.41 | 1.18 | 1.00 | 1.32 | 1.09 |
| | 1024 | 3.72e+01 | 1.38 | 1.13 | 1.03 | 0.93 | 0.95 | 0.95 | 0.92 | 0.82 |
| | 2048 | 1.96e+02 | 1.38 | 1.08 | 1.00 | 0.92 | 0.90 | 0.88 | 0.79 | 0.71 |
| | 4096 | 1.67e+03 | 1.36 | 0.63 | 0.63 | 0.47 | 0.53 | 0.52 | 0.51 | 0.43 |
| | 8192 | 1.36e+04 | 0.54 | 0.40 | 0.36 | 0.33 | 0.25 | 0.24 | 0.31 | 0.27 |
| | 16384 | 1.03e+05 | 0.51 | 0.29 | 0.34 | 0.32 | 0.31 | 0.30 | 0.29 | 0.26 |

The results reveal several architecture-dependent patterns:

**INT8 Precision.** A100 exhibits the most *consistent and predictable* behavior: 2:4 achieves 2.18–2.19× at large $M$ ($\geq$8192), closely matching the theoretical 2×. Speedups scale progressively with $M$, from ~1.04× at $M$=64 to the peak at $M$=16384. B200 shows *exceptionally high* INT8 speedups—2:4 reaches 6.47× at $M$=8192. This anomaly arises because cuBLASLt's INT8 implementation is not yet fully optimized on Blackwell; the dense baseline is slower than expected, inflating all speedup ratios. H100 transitions from sparse overhead dominance ($< 1.0$× at $M$<512) to meaningful speedup at $M\geq$1024, reaching 1.79× for 2:4 at $M$=16384. RTX 4090 shows highly irregular behavior at higher-density configurations (8:10 and beyond), with speedups dropping to 0.10–0.27× at certain $M$ values—likely due to API implementation issues rather than fundamental performance limitations.

**FP8 Precision.** FP8 shows more uniform cross-platform behavior. At $M$=16384: RTX 4090 achieves 2.08× (2:4), H100 achieves 1.73×, B200 achieves 1.85×, and RTX 5080 achieves 1.74×. The consistency indicates mature optimization of both cuBLASLt and cuSPARSELt FP8 implementations across architectures.

**BF16/FP16 Precision.** Half-precision formats demonstrate stable optimization: A100 achieves 1.52–1.71× (2:4) at large $M$; RTX 5080 peaks at 1.93× at $M$=4096, the highest among consumer GPUs. H100 shows API limitations for FP16 sparse configurations (missing data).

**M Dimension Scaling.** A consistent pattern emerges across all precisions:

- $M < 256$: Sparse kernel overhead often exceeds computation savings (speedup $< 1.0$)
- $256 \leq M < 1024$: Transition zone with variable speedups
- $M \geq 1024$: Speedups stabilize and approach theoretical bounds
- $M \geq 4096$: Best alignment with theoretical expectations

D.3.2. MODEL-SPECIFIC KERNEL RESULTS

The following tables present kernel speedup using actual model dimensions. For model-specific benchmarks, we aggregate results by summing latencies across all four linear layer types (Wqkv, Wo, W13, W2) for each $M$ value, as these execute together during inference.

**Model Kernel (INT8)**

| GPU | Model | M | cuBLASLt Latency(μs) | Speedup Ratio under Different Sparsity | | | | | | | |
|---|---|---|---|---|---|---|---|---|---|---|---|
| | | | | 2:4 | 4:6 | 6:8 | 8:10 | 10:12 | 12:14 | 14:16 | ∞:∞ |
| A100 | Llama3.2-1B | 64 | 8.14e+01 | 1.16 | 0.84 | 0.81 | 0.70 | 0.63 | 0.65 | 0.71 | 0.65 |
| | | 128 | 1.00e+02 | 1.40 | 1.06 | 1.03 | 0.89 | 0.81 | 0.82 | 0.90 | 0.82 |
| | | 256 | 1.24e+02 | 1.37 | 1.08 | 0.98 | 0.88 | 0.81 | 0.85 | 0.87 | 0.79 |
| | | 512 | 1.99e+02 | 1.52 | 1.19 | 1.13 | 0.97 | 0.92 | 0.96 | 1.01 | 0.92 |
| | | 1024 | 3.83e+02 | 1.87 | 1.44 | 1.35 | 1.14 | 1.07 | 1.13 | 1.19 | 1.06 |
| | | 2048 | 7.13e+02 | 1.79 | 1.34 | 1.24 | 1.09 | 1.01 | 1.09 | 1.11 | 0.96 |
| | | 4096 | 1.38e+03 | 1.88 | 1.40 | 1.31 | 1.10 | 1.04 | 1.09 | 1.14 | 1.02 |
| | | 8192 | 2.84e+03 | 2.00 | 1.47 | 1.40 | 1.16 | 1.10 | 1.15 | 1.21 | 1.06 |
| | | 16384 | 5.70e+03 | 1.96 | 1.46 | 1.39 | 1.19 | 1.10 | 1.15 | 1.20 | 1.07 |
| | BitNet-2B | 64 | 8.50e+01 | 1.27 | 0.89 | 0.88 | 0.78 | 0.77 | 0.69 | 0.72 | 0.69 |
| | | 128 | 9.53e+01 | 1.34 | 0.99 | 0.96 | 0.90 | 0.87 | 0.79 | 0.84 | 0.77 |
| | | 256 | 1.29e+02 | 1.57 | 1.14 | 1.12 | 1.08 | 1.02 | 0.95 | 1.00 | 0.90 |
| | | 512 | 2.26e+02 | 1.74 | 1.28 | 1.28 | 1.22 | 1.15 | 1.09 | 1.14 | 1.02 |
| | | 1024 | 4.17e+02 | 1.90 | 1.39 | 1.39 | 1.27 | 1.24 | 1.17 | 1.19 | 1.08 |
| | | 2048 | 7.46e+02 | 1.82 | 1.26 | 1.27 | 1.20 | 1.14 | 1.05 | 1.11 | 0.97 |
| | | 4096 | 1.50e+03 | 1.98 | 1.38 | 1.38 | 1.33 | 1.25 | 1.14 | 1.20 | 1.06 |
| | | 8192 | 3.12e+03 | 2.00 | 1.41 | 1.44 | 1.33 | 1.23 | 1.16 | 1.21 | 1.07 |
| | | 16384 | 6.25e+03 | 1.99 | 1.41 | 1.38 | 1.30 | 1.21 | 1.14 | 1.19 | 1.05 |
| | Llama3.2-3B | 64 | 1.11e+02 | 1.27 | 0.95 | 0.89 | 0.81 | 0.76 | 0.68 | 0.80 | 0.69 |
| | | 128 | 1.35e+02 | 1.55 | 1.18 | 1.07 | 1.01 | 0.95 | 0.87 | 0.94 | 0.86 |
| | | 256 | 1.79e+02 | 1.45 | 1.15 | 1.05 | 1.02 | 0.91 | 0.83 | 0.92 | 0.82 |
| | | 512 | 3.02e+02 | 1.73 | 1.35 | 1.24 | 1.16 | 1.08 | 0.98 | 1.08 | 0.95 |
| | | 1024 | 5.79e+02 | 2.02 | 1.57 | 1.43 | 1.29 | 1.25 | 1.11 | 1.26 | 1.11 |
| | | 2048 | 1.13e+03 | 2.03 | 1.58 | 1.41 | 1.28 | 1.22 | 1.08 | 1.24 | 1.07 |
| | | 4096 | 2.28e+03 | 2.04 | 1.56 | 1.40 | 1.27 | 1.22 | 1.05 | 1.21 | 1.06 |
| | | 8192 | 4.70e+03 | 2.02 | 1.55 | 1.42 | 1.27 | 1.24 | 1.08 | 1.22 | 1.06 |
| | | 16384 | 9.39e+03 | 2.03 | 1.56 | 1.41 | 1.28 | 1.24 | 1.10 | 1.22 | 1.07 |
| | Qwen2.5-7B | 64 | 2.38e+02 | 1.43 | 1.07 | 1.00 | 0.95 | 0.83 | 0.84 | | |
| | | 128 | 2.51e+02 | 1.45 | 1.12 | 1.04 | 0.98 | 0.87 | 0.89 | | |
| | | 256 | 4.01e+02 | 1.82 | 1.37 | 1.29 | 1.22 | 1.02 | 1.08 | | |
| | | 512 | 7.49e+02 | 1.95 | 1.45 | 1.35 | 1.27 | 1.08 | 1.14 | | |
| | | 1024 | 1.36e+03 | 1.85 | 1.40 | 1.27 | 1.18 | 1.02 | 1.06 | | |
| | | 2048 | 2.69e+03 | 2.08 | 1.43 | 1.34 | 1.25 | 1.06 | 1.12 | | |
| | | 4096 | 5.39e+03 | 2.03 | 1.49 | 1.38 | 1.29 | 1.10 | 1.16 | | |
| | | 8192 | 1.07e+04 | 2.06 | 1.51 | 1.41 | 1.31 | 1.12 | 1.17 | | |
| | | 16384 | 2.14e+04 | 2.07 | 1.52 | 1.42 | 1.32 | 1.12 | 1.18 | | |
| | Qwen2.5-14B | 64 | 2.53e+02 | 1.53 | 1.15 | 1.05 | 0.99 | 0.91 | 0.88 | 0.90 | 0.79 |
| | | 128 | 2.74e+02 | 1.59 | 1.23 | 1.12 | 1.05 | 0.97 | 0.93 | 0.97 | 0.86 |
| | | 256 | 4.48e+02 | 1.91 | 1.46 | 1.33 | 1.28 | 1.11 | 1.05 | 1.17 | 1.03 |
| | | 512 | 8.59e+02 | 2.16 | 1.63 | 1.50 | 1.39 | 1.21 | 1.14 | 1.28 | 1.13 |
| | | 1024 | 1.54e+03 | 1.99 | 1.47 | 1.35 | 1.25 | 1.10 | 1.02 | 1.16 | 1.02 |
| | | 2048 | 3.13e+03 | 2.18 | 1.57 | 1.45 | 1.35 | 1.18 | 1.08 | 1.24 | 1.08 |
| | | 4096 | 6.31e+03 | 2.12 | 1.58 | 1.43 | 1.33 | 1.16 | 1.07 | 1.23 | 1.08 |
| | | 8192 | 1.26e+04 | 2.11 | 1.55 | 1.41 | 1.33 | 1.16 | 1.07 | 1.22 | 1.06 |
| | | 16384 | 2.51e+04 | 2.08 | 1.53 | 1.41 | 1.31 | 1.15 | 1.06 | 1.21 | 1.05 |
| RTX4090 | Llama3.2-1B | 64 | 7.71e+01 | 1.38 | 1.16 | 1.12 | 1.04 | 0.89 | 0.99 | 0.91 | 0.86 |
| | | 128 | 8.53e+01 | 1.41 | 1.15 | 1.12 | 1.05 | 0.94 | 0.94 | 0.83 | 0.78 |
| | | 256 | 1.08e+02 | 1.59 | 1.31 | 1.18 | 1.01 | 0.50 | 0.79 | 1.07 | 0.54 |
| | | 512 | 1.57e+02 | 1.07 | 1.14 | 1.05 | 0.89 | 0.41 | 0.50 | 0.49 | 0.41 |
| | | 1024 | 2.91e+02 | 1.28 | 1.30 | 1.19 | 0.98 | 0.44 | 0.41 | 0.67 | 0.43 |
| | | 2048 | 5.65e+02 | 1.67 | 1.43 | 1.28 | 0.91 | 0.74 | 0.41 | 0.77 | 0.37 |
| | | 4096 | 9.22e+02 | 1.41 | 1.21 | 1.08 | 0.82 | 0.75 | 0.69 | 0.46 | 0.66 |
| | | 8192 | 1.85e+03 | 1.60 | 1.23 | 1.08 | 0.91 | 0.81 | 0.77 | 0.80 | 0.70 |
| | | 16384 | 3.73e+03 | 1.60 | 1.22 | 1.08 | 0.90 | 0.83 | 0.90 | 0.92 | 0.81 |
| | BitNet-2B | 64 | 7.16e+01 | 1.30 | 0.87 | 1.11 | 0.81 | 1.02 | 0.84 | 1.00 | 0.87 |
| | | 128 | 8.89e+01 | 1.21 | 0.94 | 1.17 | 0.85 | 1.06 | 0.98 | 1.03 | 0.90 |
| | | 256 | 1.18e+02 | 1.45 | 1.11 | 1.17 | 1.07 | 1.06 | 0.93 | 1.00 | 0.93 |
| | | 512 | 2.01e+02 | 1.64 | 1.19 | 1.20 | 1.11 | 1.09 | 1.02 | 1.05 | 0.92 |
| | | 1024 | 3.70e+02 | 1.91 | 1.33 | 1.37 | 1.27 | 1.22 | 1.13 | 1.17 | 1.04 |
| | | 2048 | 6.13e+02 | 1.63 | 1.11 | 1.12 | 1.04 | 1.01 | 0.93 | 0.95 | 0.84 |
| | | 4096 | 1.10e+03 | 1.59 | 1.08 | 1.10 | 1.02 | 0.99 | 0.90 | 0.93 | 0.83 |
| | | 8192 | 2.06e+03 | 1.56 | 1.04 | 1.07 | 0.99 | 0.97 | 0.89 | 0.91 | 0.82 |
| | | 16384 | 4.20e+03 | 1.58 | 1.07 | 1.08 | 1.01 | 0.99 | 0.91 | 0.93 | 0.83 |
| | Llama3.2-3B | 64 | 9.24e+01 | 1.37 | 1.24 | 1.19 | 1.12 | 0.45 | 0.86 | 0.40 | 0.93 |
| | | 128 | 1.06e+02 | 1.35 | 1.24 | 1.12 | 1.06 | 0.36 | 0.85 | 0.30 | 0.59 |
| | | 256 | 1.70e+02 | 1.41 | 1.16 | 1.30 | 1.03 | 0.29 | 0.41 | 0.24 | 0.37 |
| | | 512 | 2.69e+02 | 1.38 | 1.07 | 1.27 | 0.96 | 0.28 | 0.32 | 0.27 | 0.23 |
| | | 1024 | 4.74e+02 | 1.41 | 1.14 | 1.06 | 1.21 | 0.75 | 0.44 | 0.35 | 0.36 |
| | | 2048 | 8.97e+02 | 1.72 | 1.16 | 1.06 | 1.11 | 0.72 | 0.65 | 0.45 | 0.35 |
| | | 4096 | 1.49e+03 | 1.58 | 1.19 | 1.06 | 1.02 | 0.62 | 0.57 | 0.86 | 0.66 |
| | | 8192 | 3.01e+03 | 1.62 | 1.20 | 1.08 | 1.02 | 0.78 | 0.73 | 0.94 | 0.79 |
| | | 16384 | 6.13e+03 | 1.62 | 1.22 | 1.09 | 1.03 | 0.92 | 0.78 | 0.94 | 0.86 |
| | Qwen2.5-7B | 64 | 2.56e+02 | 1.61 | 1.18 | 1.11 | 1.05 | 0.08 | 0.28 | 0.22 | 0.15 |
| | | 128 | 2.60e+02 | 1.50 | 0.92 | 1.00 | 0.99 | 0.32 | 0.61 | 0.49 | 0.39 |
| | | 256 | 3.54e+02 | 1.53 | 1.08 | 1.05 | 1.01 | 0.53 | 0.32 | 0.58 | 0.29 |
| | | 512 | 5.91e+02 | 1.73 | 1.24 | 1.16 | 1.09 | 0.39 | 0.54 | 0.36 | 0.36 |
| | | 1024 | 1.14e+03 | 1.91 | 1.45 | 1.30 | 1.22 | 0.45 | 0.52 | 0.39 | 0.54 |
| | | 2048 | 1.83e+03 | 1.60 | 1.19 | 1.08 | 1.01 | 0.47 | 0.59 | 0.66 | 0.55 |

## *Model Kernel (INT8) (cont.)*

| GPU | Model | M | cuBLASLt Latency(μs) | Speedup Ratio under Different Sparsity | | | | | | | |
| --- | --- | --- | --- | --- | --- | --- | --- | --- | --- | --- | --- |
| | | | | 2:4 | 4:6 | 6:8 | 8:10 | 10:12 | 12:14 | 14:16 | ∞:∞ |
| | | 4096 | 3.57e+03 | 1.59 | 1.18 | 1.09 | 1.01 | 0.70 | 0.82 | 0.73 | 0.60 |
| | | 8192 | 6.92e+03 | 1.58 | 1.19 | 1.07 | 1.00 | 0.77 | 0.90 | 0.92 | 0.80 |
| | | 16384 | 1.40e+04 | 1.60 | 1.21 | 1.07 | 1.02 | 0.80 | 0.91 | 0.92 | 0.83 |
| | Qwen2.5-14B | 64 | 2.59e+02 | 1.61 | 1.17 | 1.09 | 1.05 | 0.22 | 0.18 | 0.23 | 0.45 |
| | | 128 | 2.81e+02 | 1.50 | 0.98 | 0.88 | 0.97 | 0.28 | 0.26 | 0.35 | 0.12 |
| | | 256 | 4.62e+02 | 1.90 | 1.05 | 1.14 | 1.19 | 0.28 | 0.33 | 0.27 | 0.44 |
| | | 512 | 7.37e+02 | 1.99 | 1.11 | 1.34 | 1.26 | 0.24 | 0.30 | 0.22 | 0.25 |
| | | 1024 | 1.27e+03 | 1.74 | 1.15 | 1.17 | 0.98 | 0.56 | 0.70 | 0.81 | 0.71 |
| | | 2048 | 2.18e+03 | 1.66 | 1.12 | 1.08 | 1.00 | 0.64 | 0.55 | 0.59 | 0.58 |
| | | 4096 | 4.01e+03 | 1.52 | 1.08 | 1.07 | 0.98 | 0.69 | 0.62 | 0.81 | 0.72 |
| | | 8192 | 8.03e+03 | 1.57 | 1.20 | 1.08 | 1.02 | 0.81 | 0.74 | 0.94 | 0.80 |
| | | 16384 | 1.63e+04 | 1.65 | 1.22 | 1.10 | 1.03 | 0.81 | 0.77 | 0.95 | 0.83 |
| H100 | Llama3.2-1B | 64 | 6.67e+01 | 1.67 | 1.44 | 1.37 | 1.31 | 1.25 | 1.25 | 1.30 | 1.07 |
| | | 128 | 7.43e+01 | 1.69 | 1.40 | 1.37 | 1.22 | 1.16 | 1.18 | 1.25 | 1.03 |
| | | 256 | 8.15e+01 | 1.42 | 1.18 | 1.13 | 1.01 | 0.94 | 0.97 | 1.01 | 0.90 |
| | | 512 | 1.25e+02 | 1.55 | 1.20 | 1.16 | 1.02 | 1.01 | 0.99 | 1.06 | 0.95 |
| | | 1024 | 1.60e+02 | 1.26 | 1.02 | 1.01 | 0.88 | 0.81 | 0.83 | 0.84 | 0.79 |
| | | 2048 | 3.12e+02 | 1.48 | 1.07 | 1.01 | 0.91 | 0.90 | 0.90 | 0.89 | 0.77 |
| | | 4096 | 5.47e+02 | 1.30 | 1.03 | 0.96 | 0.88 | 0.84 | 0.83 | 0.82 | 0.75 |
| | | 8192 | 1.02e+03 | 1.36 | 1.00 | 0.95 | 0.84 | 0.76 | 0.80 | 0.80 | 0.70 |
| | | 16384 | 2.09e+03 | 1.34 | 0.99 | 0.92 | 0.82 | 0.76 | 0.78 | 0.81 | 0.71 |
| | BitNet-2B | 64 | 6.28e+01 | 1.63 | 1.35 | 1.37 | 1.33 | 1.26 | 1.20 | 1.24 | 0.98 |
| | | 128 | 7.15e+01 | 1.70 | 1.34 | 1.42 | 1.35 | 1.27 | 1.19 | 1.22 | 0.99 |
| | | 256 | 7.90e+01 | 1.47 | 1.19 | 1.18 | 1.09 | 1.04 | 0.94 | 0.97 | 0.86 |
| | | 512 | 1.25e+02 | 1.59 | 1.26 | 1.20 | 1.13 | 1.10 | 1.06 | 1.05 | 0.94 |
| | | 1024 | 1.78e+02 | 1.32 | 1.05 | 1.01 | 0.93 | 0.88 | 0.85 | 0.84 | 0.79 |
| | | 2048 | 3.27e+02 | 1.40 | 1.04 | 0.96 | 0.94 | 0.91 | 0.83 | 0.85 | 0.78 |
| | | 4096 | 5.76e+02 | 1.31 | 1.02 | 0.92 | 0.88 | 0.84 | 0.78 | 0.80 | 0.72 |
| | | 8192 | 1.13e+03 | 1.33 | 0.96 | 0.93 | 0.86 | 0.81 | 0.78 | 0.81 | 0.71 |
| | | 16384 | 2.31e+03 | 1.29 | 0.94 | 0.90 | 0.83 | 0.79 | 0.75 | 0.77 | 0.67 |
| | Llama3.2-3B | 64 | 9.00e+01 | 1.96 | 1.54 | 1.29 | 1.22 | 1.19 | 1.15 | 1.18 | 1.09 |
| | | 128 | 9.61e+01 | 1.85 | 1.42 | 1.27 | 1.18 | 1.17 | 1.08 | 1.15 | 1.03 |
| | | 256 | 1.25e+02 | 1.76 | 1.37 | 1.24 | 1.15 | 1.13 | 1.04 | 1.11 | 1.02 |
| | | 512 | 1.70e+02 | 1.62 | 1.24 | 1.18 | 1.06 | 1.03 | 0.97 | 1.04 | 0.91 |
| | | 1024 | 2.43e+02 | 1.39 | 1.16 | 0.99 | 0.95 | 0.93 | 0.88 | 0.88 | 0.80 |
| | | 2048 | 5.00e+02 | 1.59 | 1.25 | 1.16 | 1.05 | 1.03 | 0.94 | 0.99 | 0.85 |
| | | 4096 | 8.84e+02 | 1.43 | 1.11 | 1.00 | 0.93 | 0.89 | 0.83 | 0.86 | 0.76 |
| | | 8192 | 1.75e+03 | 1.50 | 1.14 | 1.01 | 0.90 | 0.87 | 0.77 | 0.86 | 0.74 |
| | | 16384 | 3.67e+03 | 1.46 | 1.14 | 1.00 | 0.93 | 0.87 | 0.80 | 0.82 | 0.75 |
| | Qwen2.5-7B | 64 | 2.98e+02 | 2.94 | 2.26 | 2.10 | 1.96 | 1.90 | 1.90 | 1.84 | 1.66 |
| | | 128 | 3.06e+02 | 2.70 | 2.08 | 1.98 | 1.85 | 1.65 | 1.72 | 1.74 | 1.55 |
| | | 256 | 2.51e+02 | 1.80 | 1.34 | 1.31 | 1.23 | 1.12 | 1.14 | 1.11 | 1.02 |
| | | 512 | 3.33e+02 | 1.52 | 1.18 | 1.14 | 1.06 | 0.96 | 0.98 | 0.94 | 0.84 |
| | | 1024 | 5.43e+02 | 1.55 | 1.15 | 1.06 | 1.01 | 0.92 | 0.93 | 0.94 | 0.83 |
| | | 2048 | 1.06e+03 | 1.53 | 1.16 | 1.07 | 1.00 | 0.89 | 0.92 | 0.91 | 0.84 |
| | | 4096 | 2.38e+03 | 1.70 | 1.28 | 1.15 | 1.06 | 0.93 | 0.95 | 0.94 | 0.82 |
| | | 8192 | 4.59e+03 | 1.59 | 1.15 | 1.08 | 0.99 | 0.87 | 0.87 | 0.90 | 0.81 |
| | | 16384 | 9.16e+03 | 1.54 | 1.13 | 1.06 | 1.02 | 0.88 | 0.88 | 0.91 | 0.81 |
| | Qwen2.5-14B | 64 | 1.96e+02 | 1.85 | 1.41 | 1.26 | 1.26 | 1.14 | 1.10 | 1.14 | 0.97 |
| | | 128 | 2.65e+02 | 2.28 | 1.69 | 1.62 | 1.54 | 1.36 | 1.28 | 1.39 | 1.21 |
| | | 256 | 2.91e+02 | 2.03 | 1.50 | 1.44 | 1.36 | 1.24 | 1.18 | 1.24 | 1.11 |
| | | 512 | 3.44e+02 | 1.55 | 1.18 | 1.09 | 1.02 | 0.92 | 0.89 | 0.93 | 0.82 |
| | | 1024 | 6.80e+02 | 1.62 | 1.25 | 1.14 | 1.07 | 0.99 | 0.94 | 1.00 | 0.89 |
| | | 2048 | 1.25e+03 | 1.65 | 1.24 | 1.03 | 0.92 | 0.92 | 0.89 | 0.97 | 0.81 |
| | | 4096 | 2.62e+03 | 1.66 | 1.20 | 1.05 | 1.04 | 0.92 | 0.88 | 0.90 | 0.79 |
| | | 8192 | 5.41e+03 | 1.66 | 1.23 | 1.07 | 1.03 | 0.95 | 0.87 | 0.94 | 0.80 |
| | | 16384 | 1.35e+04 | 2.01 | 1.51 | 1.36 | 1.26 | 1.13 | 1.05 | 1.13 | 1.01 |
| B200 | Llama3.2-1B | 64 | 5.78e+01 | 1.86 | 1.64 | 1.66 | 1.51 | 1.40 | 1.47 | 1.55 | 1.47 |
| | | 128 | 6.39e+01 | 2.05 | 1.71 | 1.77 | 1.63 | 1.55 | 1.55 | 1.62 | 1.55 |
| | | 256 | 7.51e+01 | 2.26 | 1.83 | 1.82 | 1.74 | 1.66 | 1.73 | 1.74 | 1.69 |
| | | 512 | 9.99e+01 | 2.50 | 2.20 | 2.17 | 1.90 | 1.75 | 1.92 | 2.01 | 1.89 |
| | | 1024 | 1.71e+02 | 3.43 | 2.77 | 2.75 | 2.44 | 2.38 | 2.42 | 2.57 | 2.44 |
| | | 2048 | 3.18e+02 | 4.54 | 3.56 | 3.56 | 3.06 | 2.88 | 2.98 | 3.18 | 2.91 |
| | | 4096 | 5.83e+02 | 5.08 | 3.83 | 3.75 | 3.16 | 2.91 | 3.00 | 3.29 | 2.99 |
| | | 8192 | 1.14e+03 | 5.57 | 4.15 | 3.96 | 3.39 | 3.18 | 3.26 | 3.50 | 3.12 |
| | | 16384 | 2.25e+03 | 5.97 | 4.23 | 4.22 | 3.43 | 3.27 | 3.35 | 3.57 | 3.25 |
| | BitNet-2B | 64 | 6.09e+01 | 2.04 | 1.67 | 1.69 | 1.64 | 1.64 | 1.55 | 1.64 | 1.62 |
| | | 128 | 6.80e+01 | 2.21 | 1.83 | 1.80 | 1.74 | 1.73 | 1.65 | 1.73 | 1.73 |
| | | 256 | 7.57e+01 | 2.07 | 1.91 | 1.84 | 1.83 | 1.75 | 1.67 | 1.76 | 1.69 |
| | | 512 | 1.25e+02 | 2.99 | 2.37 | 2.51 | 2.39 | 2.25 | 2.17 | 2.30 | 2.17 |
| | | 1024 | 2.07e+02 | 3.64 | 2.88 | 2.99 | 2.95 | 2.74 | 2.64 | 2.73 | 2.57 |
| | | 2048 | 3.57e+02 | 4.48 | 3.31 | 3.42 | 3.27 | 3.02 | 2.73 | 3.03 | 2.80 |
| | | 4096 | 6.56e+02 | 4.99 | 3.78 | 3.74 | 3.56 | 3.27 | 3.06 | 3.27 | 2.91 |
| | | 8192 | 1.29e+03 | 5.79 | 4.06 | 4.08 | 3.86 | 3.51 | 3.31 | 3.51 | 3.17 |
| | | 16384 | 2.53e+03 | 5.87 | 4.15 | 4.20 | 3.95 | 3.59 | 3.31 | 3.61 | 3.20 |
| | Llama3.2-3B | 64 | 6.60e+01 | 2.00 | 1.77 | 1.75 | 1.68 | 1.60 | 1.47 | 1.68 | 1.60 |
| | | 128 | 7.58e+01 | 2.18 | 1.93 | 1.93 | 1.83 | 1.83 | 1.63 | 1.88 | 1.70 |
| | | 256 | 1.01e+02 | 2.57 | 2.33 | 2.22 | 2.12 | 2.08 | 1.88 | 2.13 | 1.96 |

### Model Kernel (INT8) (cont.)

| GPU | Model | M | cuBLASLt Latency(µs) | Speedup Ratio under Different Sparsity | | | | | | | |
| --- | --- | --- | --- | --- | --- | --- | --- | --- | --- | --- | --- |
| | | | | 2:4 | 4:6 | 6:8 | 8:10 | 10:12 | 12:14 | 14:16 | ∞:∞ |
| | | 512 | 1.62e+02 | 3.33 | 2.92 | 2.78 | 2.54 | 2.54 | 2.25 | 2.55 | 2.39 |
| | | 1024 | 2.62e+02 | 3.78 | 3.34 | 3.01 | 2.83 | 2.83 | 2.59 | 2.78 | 2.54 |
| | | 2048 | 4.97e+02 | 4.98 | 4.02 | 3.66 | 3.32 | 3.27 | 2.92 | 3.25 | 2.92 |
| | | 4096 | 9.41e+02 | 5.72 | 4.47 | 4.06 | 3.67 | 3.66 | 3.16 | 3.54 | 3.16 |
| | | 8192 | 1.82e+03 | 6.02 | 4.53 | 4.08 | 3.56 | 3.55 | 3.15 | 3.59 | 3.16 |
| | | 16384 | 3.64e+03 | 6.15 | 4.75 | 4.28 | 3.72 | 3.75 | 3.27 | 3.62 | 3.22 |
| | Qwen2.5-7B | 64 | 1.17e+02 | 2.58 | 2.10 | 2.00 | 1.89 | 1.68 | 1.78 | 1.78 | 1.63 |
| | | 128 | 1.52e+02 | 3.21 | 2.53 | 2.39 | 2.35 | 2.04 | 2.15 | 2.15 | 1.95 |
| | | 256 | 2.01e+02 | 3.84 | 3.04 | 2.85 | 2.67 | 2.35 | 2.44 | 2.56 | 2.28 |
| | | 512 | 3.27e+02 | 4.52 | 3.52 | 3.45 | 3.25 | 2.70 | 2.93 | 2.99 | 2.66 |
| | | 1024 | 5.94e+02 | 5.33 | 4.09 | 3.90 | 3.69 | 3.14 | 3.43 | 3.41 | 3.09 |
| | | 2048 | 1.10e+03 | 5.55 | 4.04 | 3.98 | 3.71 | 2.90 | 3.25 | 3.46 | 3.03 |
| | | 4096 | 2.13e+03 | 5.81 | 4.31 | 4.09 | 3.87 | 3.19 | 3.32 | 3.57 | 3.18 |
| | | 8192 | 4.21e+03 | 6.17 | 4.38 | 4.18 | 3.99 | 3.30 | 3.49 | 3.51 | 3.18 |
| | | 16384 | 8.48e+03 | 6.25 | 4.35 | 4.32 | 4.07 | 3.24 | 3.47 | 3.74 | 3.18 |
| | Qwen2.5-14B | 64 | 1.17e+02 | 2.69 | 1.90 | 1.82 | 1.72 | 1.57 | 1.49 | 1.65 | 1.53 |
| | | 128 | 1.33e+02 | 2.83 | 2.08 | 2.03 | 1.94 | 1.72 | 1.61 | 1.83 | 1.62 |
| | | 256 | 2.32e+02 | 3.88 | 3.04 | 2.94 | 2.81 | 2.42 | 2.30 | 2.57 | 2.28 |
| | | 512 | 3.82e+02 | 4.53 | 3.57 | 3.46 | 3.27 | 2.87 | 2.70 | 3.00 | 2.69 |
| | | 1024 | 6.72e+02 | 5.10 | 3.73 | 3.62 | 3.43 | 2.91 | 2.67 | 3.17 | 2.83 |
| | | 2048 | 1.25e+03 | 5.49 | 4.14 | 3.90 | 3.71 | 3.20 | 2.99 | 3.40 | 3.00 |
| | | 4096 | 2.46e+03 | 6.19 | 4.41 | 4.06 | 3.89 | 3.35 | 3.15 | 3.55 | 3.14 |
| | | 8192 | 4.86e+03 | 6.35 | 4.47 | 4.21 | 3.92 | 3.46 | 3.19 | 3.51 | 3.12 |
| RTX5080 | | 16384 | 9.89e+03 | 6.38 | 4.43 | 4.19 | 3.95 | 3.46 | 3.22 | 3.59 | 3.16 |
| | Llama3.2-1B | 64 | 6.94e+01 | 1.41 | 1.17 | 1.02 | 0.91 | 0.90 | 0.94 | 0.94 | 0.85 |
| | | 128 | 8.52e+01 | 1.54 | 1.26 | 1.12 | 1.04 | 0.98 | 1.03 | 1.02 | 0.92 |
| | | 256 | 1.17e+02 | 1.55 | 1.19 | 1.08 | 0.97 | 0.86 | 0.91 | 0.97 | 0.89 |
| | | 512 | 2.13e+02 | 1.71 | 1.35 | 1.22 | 1.06 | 0.99 | 1.03 | 1.07 | 0.97 |
| | | 1024 | 3.88e+02 | 1.68 | 1.31 | 1.16 | 1.02 | 0.94 | 0.99 | 1.03 | 0.92 |
| | | 2048 | 7.07e+02 | 1.62 | 1.24 | 1.11 | 0.97 | 0.90 | 0.94 | 0.96 | 0.86 |
| | | 4096 | 1.34e+03 | 1.62 | 1.24 | 1.10 | 0.97 | 0.89 | 0.93 | 0.96 | 0.85 |
| | | 8192 | 2.58e+03 | 1.57 | 1.19 | 1.08 | 0.95 | 0.88 | 0.92 | 0.95 | 0.84 |
| | | 16384 | 5.10e+03 | 1.55 | 1.17 | 1.07 | 0.94 | 0.87 | 0.91 | 0.93 | 0.83 |
| | BitNet-2B | 64 | 7.19e+01 | 1.46 | 1.07 | 1.06 | 1.00 | 0.95 | 0.92 | 0.92 | 0.81 |
| | | 128 | 8.21e+01 | 1.43 | 1.00 | 1.03 | 0.98 | 0.93 | 0.89 | 0.90 | 0.80 |
| | | 256 | 1.27e+02 | 1.59 | 1.11 | 1.09 | 1.10 | 1.01 | 0.94 | 0.97 | 0.90 |
| | | 512 | 2.12e+02 | 1.54 | 1.11 | 1.05 | 1.04 | 0.97 | 0.94 | 0.95 | 0.85 |
| | | 1024 | 4.15e+02 | 1.70 | 1.21 | 1.17 | 1.13 | 1.07 | 1.02 | 1.03 | 0.91 |
| | | 2048 | 7.61e+02 | 1.62 | 1.14 | 1.10 | 1.06 | 1.00 | 0.95 | 0.97 | 0.86 |
| | | 4096 | 1.37e+03 | 1.49 | 1.05 | 1.01 | 0.96 | 0.90 | 0.87 | 0.87 | 0.78 |
| | | 8192 | 2.78e+03 | 1.51 | 1.06 | 1.03 | 0.97 | 0.92 | 0.88 | 0.88 | 0.78 |
| | | 16384 | 5.70e+03 | 1.54 | 1.09 | 1.04 | 1.00 | 0.94 | 0.89 | 0.91 | 0.80 |
| | Llama3.2-3B | 64 | 8.20e+01 | 1.38 | 1.14 | 1.05 | 1.00 | 0.92 | 0.81 | 0.90 | 0.80 |
| | | 128 | 1.07e+02 | 1.53 | 1.24 | 1.13 | 1.05 | 1.00 | 0.86 | 0.97 | 0.86 |
| | | 256 | 1.96e+02 | 1.71 | 1.40 | 1.24 | 1.15 | 1.12 | 0.99 | 1.07 | 0.95 |
| | | 512 | 3.33e+02 | 1.61 | 1.25 | 1.12 | 1.05 | 0.99 | 0.86 | 0.96 | 0.84 |
| | | 1024 | 5.88e+02 | 1.63 | 1.27 | 1.12 | 1.04 | 0.97 | 0.85 | 0.96 | 0.85 |
| | | 2048 | 1.10e+03 | 1.65 | 1.27 | 1.13 | 1.05 | 0.98 | 0.87 | 0.97 | 0.86 |
| | | 4096 | 2.01e+03 | 1.51 | 1.17 | 1.03 | 0.98 | 0.91 | 0.80 | 0.90 | 0.79 |
| | | 8192 | 4.07e+03 | 1.54 | 1.19 | 1.05 | 0.99 | 0.92 | 0.82 | 0.91 | 0.80 |
| | | 16384 | 8.20e+03 | 1.56 | 1.21 | 1.06 | 1.00 | 0.93 | 0.83 | 0.92 | 0.81 |
| | Qwen2.5-7B | 64 | 2.54e+02 | 1.58 | 1.21 | 1.07 | 0.97 | 0.95 | 0.85 | 0.84 | 0.73 |
| | | 128 | 2.74e+02 | 1.64 | 1.25 | 1.10 | 0.93 | 0.90 | 0.87 | 0.87 | 0.75 |
| | | 256 | 3.99e+02 | 1.71 | 1.33 | 1.18 | 1.08 | 0.99 | 1.02 | 1.01 | 0.88 |
| | | 512 | 6.82e+02 | 1.54 | 1.16 | 1.03 | 0.96 | 0.83 | 0.87 | 0.89 | 0.79 |
| | | 1024 | 1.29e+03 | 1.64 | 1.24 | 1.12 | 1.03 | 0.90 | 0.95 | 0.96 | 0.85 |
| | | 2048 | 2.38e+03 | 1.53 | 1.16 | 1.04 | 0.95 | 0.83 | 0.88 | 0.90 | 0.79 |
| | | 4096 | 4.70e+03 | 1.51 | 1.16 | 1.03 | 0.95 | 0.83 | 0.88 | 0.90 | 0.79 |
| | | 8192 | 9.27e+03 | 1.49 | 1.14 | 1.02 | 0.95 | 0.82 | 0.87 | 0.89 | 0.78 |
| | | 16384 | 1.88e+04 | 1.50 | 1.16 | 1.02 | 0.97 | 0.84 | 0.89 | 0.90 | 0.80 |
| | Qwen2.5-14B | 64 | 2.62e+02 | 1.62 | 1.24 | 1.11 | 0.91 | 0.84 | 0.82 | 0.81 | 0.71 |
| | | 128 | 2.98e+02 | 1.63 | 1.21 | 1.09 | 0.95 | 0.88 | 0.85 | 0.85 | 0.75 |
| | | 256 | 4.28e+02 | 1.63 | 1.24 | 1.09 | 1.05 | 0.90 | 0.87 | 0.94 | 0.83 |
| | | 512 | 7.95e+02 | 1.72 | 1.28 | 1.15 | 1.10 | 0.95 | 0.90 | 1.01 | 0.88 |
| | | 1024 | 1.49e+03 | 1.66 | 1.25 | 1.11 | 1.06 | 0.93 | 0.87 | 0.97 | 0.86 |
| | | 2048 | 2.69e+03 | 1.51 | 1.13 | 1.00 | 0.96 | 0.83 | 0.79 | 0.88 | 0.77 |
| | | 4096 | 5.36e+03 | 1.50 | 1.12 | 0.99 | 0.94 | 0.82 | 0.79 | 0.88 | 0.77 |
| | | 8192 | 1.09e+04 | 1.52 | 1.15 | 1.02 | 0.97 | 0.84 | 0.81 | 0.90 | 0.79 |
| | | 16384 | 2.18e+04 | 1.52 | 1.14 | 1.02 | 0.97 | 0.85 | 0.81 | 0.90 | 0.79 |
| GB10 | Llama3.2-1B | 64 | 2.14e+02 | 2.39 | 1.31 | 1.23 | 0.97 | 0.88 | 0.96 | 1.00 | 0.82 |
| | | 128 | 2.30e+02 | 2.22 | 1.37 | 1.29 | 0.83 | 0.82 | 0.90 | 1.00 | 0.83 |
| | | 256 | 3.38e+02 | 2.29 | 1.52 | 1.54 | 1.03 | 1.00 | 1.04 | 1.09 | 1.06 |
| | | 512 | 4.63e+02 | 1.67 | 1.18 | 1.17 | 0.95 | 0.91 | 0.86 | 0.93 | 0.85 |
| | | 1024 | 8.19e+02 | 1.54 | 1.23 | 1.08 | 0.96 | 0.89 | 0.95 | 0.99 | 0.85 |
| | | 2048 | 1.40e+03 | 1.39 | 1.06 | 0.97 | 0.81 | 0.76 | 0.80 | 0.76 | 0.67 |
| | | 4096 | 2.79e+03 | 1.44 | 1.03 | 0.93 | 0.80 | 0.76 | 0.79 | 0.82 | 0.73 |
| | | 8192 | 5.43e+03 | 1.37 | 1.04 | 0.92 | 0.80 | 0.73 | 0.76 | 0.80 | 0.69 |
| | | 16384 | 1.10e+04 | 1.36 | 1.04 | 0.96 | 0.81 | 0.76 | 0.78 | 0.81 | 0.69 |

### *Model Kernel (INT8) (cont.)*

| GPU | Model | M | cuBLASLt Latency(μs) | Speedup Ratio under Different Sparsity | | | | | | | |
|---|---|---|---|---|---|---|---|---|---|---|---|
| | | | | 2:4 | 4:6 | 6:8 | 8:10 | 10:12 | 12:14 | 14:16 | ∞:∞ |
| | BitNet-2B | 64 | 2.19e+02 | 2.81 | 1.13 | 1.20 | 1.06 | 0.75 | 0.71 | 0.92 | 0.84 |
| | | 128 | 2.47e+02 | 2.26 | 1.21 | 1.26 | 1.09 | 0.90 | 0.74 | 0.97 | 0.76 |
| | | 256 | 3.48e+02 | 2.13 | 1.28 | 1.21 | 1.14 | 0.99 | 0.94 | 0.95 | 0.83 |
| | | 512 | 5.17e+02 | 1.77 | 1.11 | 1.15 | 1.11 | 0.96 | 0.93 | 0.97 | 0.83 |
| | | 1024 | 8.53e+02 | 1.41 | 1.07 | 1.02 | 0.96 | 0.95 | 0.89 | 0.87 | 0.76 |
| | | 2048 | 1.64e+03 | 1.48 | 1.06 | 1.02 | 0.96 | 0.92 | 0.87 | 0.87 | 0.77 |
| | | 4096 | 3.22e+03 | 1.45 | 1.01 | 1.00 | 0.93 | 0.87 | 0.79 | 0.81 | 0.73 |
| | | 8192 | 6.24e+03 | 1.39 | 0.99 | 0.96 | 0.91 | 0.86 | 0.81 | 0.82 | 0.73 |
| | | 16384 | 1.24e+04 | 1.38 | 0.98 | 0.96 | 0.90 | 0.84 | 0.79 | 0.81 | 0.72 |
| | Llama3.2-3B | 64 | 3.62e+02 | 2.41 | 1.30 | 1.22 | 0.91 | 0.96 | 0.75 | 0.99 | 0.76 |
| | | 128 | 3.90e+02 | 2.04 | 1.22 | 1.09 | 0.88 | 0.92 | 0.78 | 0.94 | 0.72 |
| | | 256 | 5.11e+02 | 1.93 | 1.34 | 1.18 | 0.95 | 1.01 | 0.80 | 1.03 | 0.81 |
| | | 512 | 7.60e+02 | 1.81 | 1.36 | 1.20 | 1.02 | 0.97 | 0.92 | 1.06 | 0.84 |
| | | 1024 | 1.28e+03 | 1.63 | 1.23 | 1.06 | 1.03 | 0.97 | 0.87 | 0.94 | 0.81 |
| | | 2048 | 2.40e+03 | 1.57 | 1.20 | 1.07 | 0.98 | 0.91 | 0.79 | 0.90 | 0.74 |
| | | 4096 | 4.65e+03 | 1.50 | 1.13 | 1.02 | 0.91 | 0.85 | 0.78 | 0.86 | 0.75 |
| | | 8192 | 9.08e+03 | 1.46 | 1.11 | 1.00 | 0.92 | 0.87 | 0.76 | 0.84 | 0.73 |
| | | 16384 | 1.78e+04 | 1.41 | 1.09 | 0.98 | 0.88 | 0.84 | 0.75 | 0.83 | 0.72 |
| | Qwen2.5-7B | 64 | 1.03e+03 | 1.58 | 0.90 | 1.01 | 0.93 | 0.71 | 0.82 | 0.87 | 0.77 |
| | | 128 | 1.22e+03 | 1.76 | 1.04 | 1.15 | 1.03 | 0.80 | 0.90 | 0.99 | 0.86 |
| | | 256 | 1.46e+03 | 1.69 | 1.11 | 1.16 | 1.07 | 0.84 | 0.93 | 0.96 | 0.89 |
| | | 512 | 1.90e+03 | 1.74 | 1.13 | 1.13 | 1.04 | 0.88 | 0.95 | 0.99 | 0.86 |
| | | 1024 | 3.01e+03 | 1.56 | 1.07 | 1.00 | 0.91 | 0.79 | 0.82 | 0.84 | 0.76 |
| | | 2048 | 5.67e+03 | 1.43 | 0.98 | 0.91 | 0.88 | 0.74 | 0.74 | 0.79 | 0.70 |
| | | 4096 | 1.10e+04 | 1.42 | 1.00 | 0.93 | 0.85 | 0.72 | 0.76 | 0.78 | 0.70 |
| | | 8192 | 2.11e+04 | 1.39 | 0.94 | 0.86 | 0.82 | 0.70 | 0.73 | 0.76 | 0.66 |
| | | 16384 | 4.19e+04 | 1.33 | 0.94 | 0.86 | 0.78 | 0.69 | 0.72 | 0.72 | 0.60 |
| | Qwen2.5-14B | 64 | 1.31e+03 | 1.82 | 1.07 | 1.09 | 0.96 | 0.77 | 0.70 | 0.92 | 0.75 |
| | | 128 | 1.44e+03 | 1.86 | 1.09 | 1.14 | 1.03 | 0.82 | 0.74 | 0.97 | 0.82 |
| | | 256 | 1.68e+03 | 1.92 | 1.15 | 1.16 | 1.07 | 0.89 | 0.80 | 1.00 | 0.83 |
| | | 512 | 2.29e+03 | 1.94 | 1.29 | 1.21 | 1.12 | 0.98 | 0.90 | 1.01 | 0.89 |
| | | 1024 | 3.79e+03 | 1.82 | 1.29 | 1.10 | 1.04 | 0.91 | 0.83 | 0.94 | 0.83 |
| | | 2048 | 6.67e+03 | 1.58 | 1.12 | 0.98 | 0.92 | 0.81 | 0.72 | 0.84 | 0.72 |
| | | 4096 | 1.27e+04 | 1.50 | 1.09 | 0.93 | 0.87 | 0.76 | 0.70 | 0.79 | 0.70 |
| | | 8192 | 2.46e+04 | 1.45 | 1.03 | 0.91 | 0.85 | 0.75 | 0.69 | 0.77 | 0.67 |
| | | 16384 | 4.80e+04 | 1.44 | 1.01 | 0.89 | 0.83 | 0.74 | 0.67 | 0.75 | 0.66 |

## Model Kernel (FP8)

| GPU | Model | M | cuBLASLt Latency(μs) | Speedup Ratio under Different Sparsity | | | | | | | |
|---|---|---|---|---|---|---|---|---|---|---|---|
| | | | | 2:4 | 4:6 | 6:8 | 8:10 | 10:12 | 12:14 | 14:16 | ∞:∞ |
| RTX4090 | Llama3.2-1B | 64 | 6.64e+01 | 0.80 | 0.78 | 0.71 | 0.63 | 0.61 | 0.50 | 0.65 | 0.52 |
| | | 128 | 8.19e+01 | 1.03 | 0.91 | 0.84 | 0.74 | 0.69 | 0.58 | 0.70 | 0.64 |
| | | 256 | 1.27e+02 | 1.45 | 1.14 | 1.02 | 0.97 | 0.45 | 0.51 | 0.52 | 0.42 |
| | | 512 | 2.17e+02 | 1.35 | 1.27 | 1.19 | 1.07 | 0.46 | 0.49 | 0.50 | 0.44 |
| | | 1024 | 4.07e+02 | 1.69 | 1.41 | 1.30 | 1.19 | 0.56 | 0.59 | 0.48 | 0.45 |
| | | 2048 | 7.98e+02 | 1.77 | 1.46 | 1.35 | 1.20 | 0.55 | 0.52 | 0.83 | 0.41 |
| | | 4096 | 1.59e+03 | 1.97 | 1.45 | 1.35 | 1.23 | 0.54 | 0.95 | 1.01 | 0.81 |
| | | 8192 | 3.21e+03 | 1.98 | 1.49 | 1.36 | 1.24 | 0.93 | 0.95 | 1.04 | 0.91 |
| | | 16384 | 6.46e+03 | 2.00 | 1.49 | 1.36 | 1.24 | 1.19 | 1.18 | 1.17 | 1.03 |
| | BitNet-2B | 64 | 6.61e+01 | 0.84 | 0.66 | 0.68 | 0.67 | 0.49 | 0.61 | 0.59 | 0.54 |
| | | 128 | 9.20e+01 | 1.14 | 0.81 | 0.90 | 0.88 | 0.73 | 0.80 | 0.78 | 0.71 |
| | | 256 | 1.51e+02 | 1.55 | 1.14 | 1.12 | 1.12 | 0.95 | 1.00 | 1.04 | 0.90 |
| | | 512 | 2.62e+02 | 1.62 | 1.24 | 1.14 | 1.08 | 1.03 | 1.00 | 1.00 | 0.88 |
| | | 1024 | 4.90e+02 | 1.68 | 1.26 | 1.16 | 1.10 | 1.05 | 1.02 | 1.01 | 0.89 |
| | | 2048 | 8.93e+02 | 1.77 | 1.33 | 1.21 | 1.15 | 1.09 | 1.04 | 1.05 | 0.92 |
| | | 4096 | 1.80e+03 | 1.91 | 1.42 | 1.30 | 1.22 | 1.16 | 1.10 | 1.12 | 0.98 |
| | | 8192 | 3.58e+03 | 1.96 | 1.45 | 1.34 | 1.25 | 1.19 | 1.14 | 1.14 | 1.00 |
| | | 16384 | 7.26e+03 | 2.00 | 1.48 | 1.35 | 1.27 | 1.20 | 1.16 | 1.15 | 1.02 |
| | Llama3.2-3B | 64 | 7.80e+01 | 0.85 | 0.72 | 0.70 | 0.60 | 0.62 | 0.57 | 0.27 | 0.58 |
| | | 128 | 1.17e+02 | 1.34 | 0.94 | 0.88 | 0.94 | 0.62 | 0.48 | 0.49 | 0.40 |
| | | 256 | 2.03e+02 | 1.53 | 1.35 | 1.24 | 1.17 | 0.43 | 0.48 | 0.29 | 0.34 |
| | | 512 | 3.58e+02 | 1.29 | 1.33 | 1.15 | 0.85 | 0.34 | 0.31 | 0.51 | 0.40 |
| | | 1024 | 6.58e+02 | 1.46 | 1.34 | 1.20 | 1.10 | 0.28 | 0.41 | 0.46 | 0.63 |
| | | 2048 | 1.32e+03 | 2.01 | 1.52 | 1.36 | 1.26 | 0.60 | 1.07 | 0.63 | 0.58 |
| | | 4096 | 2.64e+03 | 2.03 | 1.54 | 1.39 | 1.27 | 0.83 | 1.17 | 0.86 | 0.59 |
| | | 8192 | 5.31e+03 | 2.04 | 1.54 | 1.38 | 1.28 | 1.23 | 1.15 | 1.18 | 1.03 |
| | | 16384 | 1.07e+04 | 2.05 | 1.56 | 1.39 | 1.29 | 1.24 | 1.19 | 1.20 | 1.06 |
| | Qwen2.5-7B | 64 | 2.33e+02 | 1.17 | 0.88 | 0.80 | 0.71 | 0.24 | 0.44 | 0.34 | 0.45 |
| | | 128 | 2.83e+02 | 1.37 | 0.93 | 0.89 | 0.88 | 0.29 | 0.28 | 0.19 | 0.34 |
| | | 256 | 4.69e+02 | 1.14 | 1.36 | 1.23 | 1.16 | 0.52 | 0.51 | 0.17 | 0.47 |
| | | 512 | 8.31e+02 | 1.80 | 1.43 | 1.29 | 1.20 | 0.46 | 0.43 | 0.38 | 0.48 |
| | | 1024 | 1.62e+03 | 1.93 | 1.46 | 1.32 | 1.24 | 0.55 | 0.63 | 0.42 | 0.43 |
| | | 2048 | 3.07e+03 | 1.91 | 1.43 | 1.29 | 1.20 | 0.81 | 1.11 | 0.77 | 0.98 |
| | | 4096 | 6.12e+03 | 2.00 | 1.49 | 1.36 | 1.27 | 1.16 | 1.07 | 1.08 | 1.00 |
| | | 8192 | 1.24e+04 | 2.04 | 1.52 | 1.38 | 1.29 | 1.22 | 1.22 | 1.20 | 1.06 |
| | | 16384 | 2.48e+04 | 2.05 | 1.52 | 1.37 | 1.28 | 1.21 | 1.21 | 1.20 | 1.04 |
| | Qwen2.5-14B | 64 | 2.59e+02 | 1.36 | 1.04 | 0.92 | 0.87 | 0.32 | 0.13 | 0.29 | 0.31 |

## Model Kernel (FP8) (cont.)

| GPU | Model | M | cuBLASLt Latency(μs) | Speedup Ratio under Different Sparsity | | | | | | | |
|---|---|---|---|---|---|---|---|---|---|---|---|
| | | | | 2:4 | 4:6 | 6:8 | 8:10 | 10:12 | 12:14 | 14:16 | ∞:∞ |
| | | 128 | 3.22e+02 | 1.59 | 0.80 | 1.07 | 0.99 | 0.21 | 0.31 | 0.18 | 0.38 |
| | | 256 | 5.36e+02 | 1.34 | 1.31 | 1.09 | 1.01 | 0.21 | 0.29 | 0.30 | 0.27 |
| | | 512 | 1.00e+03 | 1.76 | 1.22 | 1.10 | 1.06 | 0.30 | 0.34 | 0.32 | 0.30 |
| | | 1024 | 1.82e+03 | 1.87 | 1.39 | 1.25 | 1.13 | 0.70 | 0.71 | 0.66 | 0.79 |
| | | 2048 | 3.57e+03 | 1.99 | 1.48 | 1.32 | 1.19 | 1.13 | 1.10 | 0.79 | 1.02 |
| | | 4096 | 7.14e+03 | 2.03 | 1.52 | 1.36 | 1.19 | 1.18 | 1.11 | 1.16 | 1.02 |
| | | 8192 | 1.45e+04 | 2.06 | 1.54 | 1.38 | 1.30 | 1.17 | 1.19 | 1.19 | 1.05 |
| | | 16384 | 2.95e+04 | 2.10 | 1.55 | 1.40 | 1.31 | 1.25 | 1.21 | 1.22 | 1.07 |
| H100 | Llama3.2-1B | 64 | 3.80e+01 | 0.95 | 0.81 | 0.78 | 0.72 | 0.70 | 0.71 | 0.73 | 0.60 |
| | | 128 | 4.27e+01 | 0.97 | 0.81 | 0.80 | 0.72 | 0.69 | 0.70 | 0.71 | 0.60 |
| | | 256 | 5.12e+01 | 0.91 | 0.73 | 0.69 | 0.64 | 0.60 | 0.60 | 0.61 | 0.55 |
| | | 512 | 8.43e+01 | 1.04 | 0.79 | 0.79 | 0.67 | 0.65 | 0.68 | 0.69 | 0.64 |
| | | 1024 | 1.51e+02 | 1.17 | 0.95 | 0.86 | 0.83 | 0.75 | 0.77 | 0.78 | 0.71 |
| | | 2048 | 2.88e+02 | 1.24 | 0.98 | 0.95 | 0.83 | 0.80 | 0.78 | 0.80 | 0.72 |
| | | 4096 | 5.83e+02 | 1.37 | 1.06 | 0.98 | 0.86 | 0.87 | 0.87 | 0.84 | 0.78 |
| | | 8192 | 1.13e+03 | 1.42 | 1.07 | 0.94 | 0.85 | 0.83 | 0.85 | 0.82 | 0.72 |
| | | 16384 | 2.53e+03 | 1.54 | 1.14 | 1.02 | 0.96 | 0.91 | 0.87 | 0.90 | 0.81 |
| | BitNet-2B | 64 | 4.52e+01 | 1.18 | 0.95 | 0.99 | 0.95 | 0.91 | 0.87 | 0.89 | 0.69 |
| | | 128 | 4.62e+01 | 1.10 | 0.87 | 0.91 | 0.88 | 0.81 | 0.77 | 0.76 | 0.65 |
| | | 256 | 5.63e+01 | 1.04 | 0.85 | 0.81 | 0.76 | 0.74 | 0.67 | 0.69 | 0.62 |
| | | 512 | 8.86e+01 | 1.10 | 0.87 | 0.84 | 0.77 | 0.74 | 0.70 | 0.71 | 0.65 |
| | | 1024 | 1.67e+02 | 1.25 | 0.93 | 0.90 | 0.85 | 0.81 | 0.80 | 0.78 | 0.69 |
| | | 2048 | 3.19e+02 | 1.30 | 1.00 | 0.95 | 0.85 | 0.86 | 0.79 | 0.82 | 0.72 |
| | | 4096 | 6.87e+02 | 1.48 | 1.12 | 1.05 | 0.99 | 0.93 | 0.89 | 0.90 | 0.82 |
| | | 8192 | 1.36e+03 | 1.50 | 1.11 | 1.04 | 0.98 | 0.94 | 0.90 | 0.88 | 0.78 |
| | | 16384 | 2.88e+03 | 1.57 | 1.11 | 1.05 | 0.99 | 0.92 | 0.88 | 0.92 | 0.80 |
| | Llama3.2-3B | 64 | 6.69e+01 | 1.46 | 1.11 | 0.97 | 0.92 | 0.88 | 0.85 | 0.88 | 0.82 |
| | | 128 | 6.64e+01 | 1.27 | 0.96 | 0.88 | 0.81 | 0.80 | 0.75 | 0.79 | 0.71 |
| | | 256 | 7.85e+01 | 1.07 | 0.87 | 0.77 | 0.72 | 0.71 | 0.67 | 0.67 | 0.61 |
| | | 512 | 1.29e+02 | 1.20 | 0.93 | 0.86 | 0.81 | 0.77 | 0.70 | 0.75 | 0.66 |
| | | 1024 | 2.47e+02 | 1.39 | 1.07 | 0.97 | 0.91 | 0.90 | 0.85 | 0.84 | 0.75 |
| | | 2048 | 4.78e+02 | 1.49 | 1.16 | 1.04 | 0.96 | 0.92 | 0.85 | 0.89 | 0.77 |
| | | 4096 | 1.09e+03 | 1.67 | 1.34 | 1.19 | 1.08 | 1.03 | 0.97 | 1.02 | 0.87 |
| | | 8192 | 1.95e+03 | 1.53 | 1.22 | 1.02 | 0.95 | 0.91 | 0.82 | 0.89 | 0.76 |
| | | 16384 | 4.66e+03 | 1.75 | 1.35 | 1.20 | 1.12 | 1.02 | 0.94 | 1.03 | 0.89 |
| | Qwen2.5-7B | 64 | 1.53e+02 | 1.53 | 1.16 | 1.08 | 1.00 | 0.97 | 0.97 | 0.95 | 0.85 |
| | | 128 | 1.46e+02 | 1.29 | 1.00 | 0.94 | 0.89 | 0.81 | 0.83 | 0.81 | 0.74 |
| | | 256 | 1.80e+02 | 1.26 | 0.94 | 0.90 | 0.85 | 0.78 | 0.78 | 0.79 | 0.72 |
| | | 512 | 3.05e+02 | 1.32 | 1.08 | 0.96 | 0.90 | 0.84 | 0.84 | 0.84 | 0.77 |
| | | 1024 | 5.61e+02 | 1.54 | 1.14 | 1.08 | 0.97 | 0.87 | 0.90 | 0.92 | 0.81 |
| | | 2048 | 1.14e+03 | 1.60 | 1.23 | 1.08 | 1.03 | 0.84 | 0.90 | 0.93 | 0.82 |
| | | 4096 | 2.55e+03 | 1.74 | 1.25 | 1.13 | 0.99 | 0.96 | 0.93 | 0.94 | 0.85 |
| | | 8192 | 4.96e+03 | 1.60 | 1.19 | 1.08 | 0.93 | 0.89 | 0.92 | 0.92 | 0.82 |
| | | 16384 | 1.10e+04 | 1.74 | 1.30 | 1.19 | 1.02 | 1.00 | 1.01 | 1.01 | 0.90 |
| | Qwen2.5-14B | 64 | 1.48e+02 | 1.38 | 1.06 | 0.99 | 0.94 | 0.87 | 0.83 | 0.85 | 0.72 |
| | | 128 | 1.56e+02 | 1.32 | 1.00 | 0.94 | 0.89 | 0.80 | 0.77 | 0.81 | 0.70 |
| | | 256 | 1.92e+02 | 1.31 | 0.98 | 0.75 | 0.85 | 0.80 | 0.75 | 0.79 | 0.70 |
| | | 512 | 3.69e+02 | 1.58 | 1.18 | 1.05 | 1.00 | 0.94 | 0.90 | 0.94 | 0.82 |
| | | 1024 | 7.09e+02 | 1.50 | 1.22 | 1.12 | 1.05 | 0.98 | 0.90 | 0.99 | 0.84 |
| | | 2048 | 1.48e+03 | 1.84 | 1.35 | 1.24 | 1.16 | 1.02 | 0.94 | 1.06 | 0.89 |
| | | 4096 | 2.99e+03 | 1.72 | 1.33 | 1.14 | 1.11 | 1.01 | 0.94 | 0.99 | 0.88 |
| | | 8192 | 6.30e+03 | 1.73 | 1.34 | 1.08 | 1.01 | 1.01 | 0.95 | 1.03 | 0.88 |
| | | 16384 | 1.30e+04 | 1.67 | 1.32 | 1.07 | 1.00 | 1.02 | 1.00 | 1.02 | 0.90 |
| B200 | Llama3.2-1B | 64 | 3.64e+01 | 1.17 | 1.01 | 1.01 | 0.93 | 0.88 | 0.93 | 0.95 | 0.93 |
| | | 128 | 3.78e+01 | 1.22 | 0.99 | 1.02 | 0.96 | 0.92 | 0.92 | 0.92 | 0.92 |
| | | 256 | 4.32e+01 | 1.28 | 1.05 | 1.05 | 1.02 | 0.95 | 0.99 | 1.00 | 0.99 |
| | | 512 | 4.96e+01 | 1.23 | 1.08 | 1.05 | 0.94 | 0.86 | 0.95 | 1.01 | 0.93 |
| | | 1024 | 6.33e+01 | 1.25 | 1.05 | 1.03 | 0.90 | 0.88 | 0.88 | 0.93 | 0.89 |
| | | 2048 | 9.71e+01 | 1.35 | 1.05 | 1.04 | 0.92 | 0.85 | 0.88 | 0.93 | 0.86 |
| | | 4096 | 1.70e+02 | 1.43 | 1.05 | 1.03 | 0.87 | 0.80 | 0.84 | 0.92 | 0.82 |
| | | 8192 | 3.42e+02 | 1.58 | 1.16 | 1.12 | 0.97 | 0.90 | 0.93 | 0.99 | 0.88 |
| | | 16384 | 6.55e+02 | 1.61 | 1.15 | 1.14 | 0.95 | 0.89 | 0.91 | 0.99 | 0.86 |
| | BitNet-2B | 64 | 3.50e+01 | 1.10 | 0.94 | 0.94 | 0.94 | 0.94 | 0.87 | 0.94 | 0.91 |
| | | 128 | 3.39e+01 | 1.04 | 0.91 | 0.87 | 0.86 | 0.86 | 0.82 | 0.86 | 0.85 |
| | | 256 | 3.74e+01 | 1.03 | 0.94 | 0.91 | 0.90 | 0.89 | 0.83 | 0.87 | 0.81 |
| | | 512 | 4.79e+01 | 1.13 | 0.89 | 0.94 | 0.91 | 0.86 | 0.83 | 0.86 | 0.83 |
| | | 1024 | 6.52e+01 | 1.18 | 0.91 | 0.94 | 0.93 | 0.87 | 0.84 | 0.86 | 0.80 |
| | | 2048 | 1.05e+02 | 1.31 | 0.93 | 0.98 | 0.92 | 0.85 | 0.77 | 0.85 | 0.78 |
| | | 4096 | 1.90e+02 | 1.41 | 1.05 | 1.03 | 0.98 | 0.90 | 0.85 | 0.90 | 0.79 |
| | | 8192 | 3.65e+02 | 1.54 | 1.08 | 1.08 | 1.01 | 0.94 | 0.88 | 0.93 | 0.83 |
| | | 16384 | 7.37e+02 | 1.62 | 1.13 | 1.13 | 1.05 | 0.96 | 0.92 | 0.95 | 0.86 |
| | Llama3.2-3B | 64 | 4.16e+01 | 1.20 | 1.12 | 1.08 | 1.05 | 1.01 | 0.86 | 1.02 | 1.00 |
| | | 128 | 4.03e+01 | 1.14 | 1.03 | 1.03 | 0.95 | 0.97 | 0.83 | 0.97 | 0.88 |
| | | 256 | 4.45e+01 | 1.13 | 1.04 | 0.98 | 0.93 | 0.90 | 0.83 | 0.93 | 0.84 |
| | | 512 | 5.71e+01 | 1.20 | 1.03 | 0.96 | 0.88 | 0.89 | 0.79 | 0.89 | 0.84 |
| | | 1024 | 8.24e+01 | 1.22 | 1.04 | 0.95 | 0.87 | 0.88 | 0.79 | 0.85 | 0.79 |
| | | 2048 | 1.43e+02 | 1.37 | 1.12 | 1.02 | 0.89 | 0.89 | 0.79 | 0.89 | 0.79 |
| | | 4096 | 2.75e+02 | 1.56 | 1.24 | 1.12 | 0.98 | 0.99 | 0.87 | 0.97 | 0.86 |

### Model Kernel (FP8) (cont.)

| GPU | Model | M | cuBLASLt Latency(μs) | Speedup Ratio under Different Sparsity | | | | | | | |
|---|---|---|---|---|---|---|---|---|---|---|---|
| | | | | 2:4 | 4:6 | 6:8 | 8:10 | 10:12 | 12:14 | 14:16 | ∞:∞ |
| | | 8192 | 5.41e+02 | 1.64 | 1.26 | 1.13 | 0.99 | 1.00 | 0.88 | 0.99 | 0.87 |
| | | 16384 | 1.11e+03 | 1.72 | 1.32 | 1.20 | 1.06 | 1.04 | 0.92 | 1.03 | 0.92 |
| | Qwen2.5-7B | 64 | 6.08e+01 | 1.34 | 1.05 | 1.01 | 0.98 | 0.86 | 0.89 | 0.91 | 0.83 |
| | | 128 | 6.30e+01 | 1.33 | 1.02 | 0.99 | 0.96 | 0.84 | 0.87 | 0.87 | 0.81 |
| | | 256 | 7.33e+01 | 1.41 | 1.08 | 1.01 | 0.96 | 0.84 | 0.89 | 0.91 | 0.82 |
| | | 512 | 1.01e+02 | 1.37 | 1.03 | 1.01 | 0.98 | 0.80 | 0.87 | 0.89 | 0.80 |
| | | 1024 | 1.71e+02 | 1.49 | 1.12 | 1.07 | 1.01 | 0.86 | 0.94 | 0.93 | 0.84 |
| | | 2048 | 3.33e+02 | 1.58 | 1.13 | 1.12 | 1.05 | 0.84 | 0.93 | 0.98 | 0.86 |
| | | 4096 | 6.18e+02 | 1.60 | 1.13 | 1.09 | 1.04 | 0.86 | 0.93 | 0.95 | 0.86 |
| | | 8192 | 1.28e+03 | 1.75 | 1.22 | 1.15 | 1.09 | 0.93 | 1.01 | 0.99 | 0.89 |
| | | 16384 | 2.59e+03 | 1.79 | 1.26 | 1.18 | 1.13 | 0.93 | 1.02 | 1.07 | 0.90 |
| | Qwen2.5-14B | 64 | 6.84e+01 | 1.55 | 1.10 | 1.07 | 1.01 | 0.92 | 0.85 | 0.95 | 0.86 |
| | | 128 | 7.23e+01 | 1.51 | 1.13 | 1.09 | 1.06 | 0.90 | 0.86 | 0.96 | 0.87 |
| | | 256 | 7.80e+01 | 1.31 | 1.00 | 0.98 | 0.93 | 0.81 | 0.77 | 0.87 | 0.76 |
| | | 512 | 1.15e+02 | 1.34 | 1.06 | 1.02 | 0.96 | 0.84 | 0.79 | 0.89 | 0.79 |
| | | 1024 | 2.01e+02 | 1.47 | 1.07 | 1.03 | 0.97 | 0.82 | 0.77 | 0.89 | 0.80 |
| | | 2048 | 3.73e+02 | 1.59 | 1.16 | 1.09 | 1.03 | 0.90 | 0.85 | 0.94 | 0.83 |
| | | 4096 | 7.34e+02 | 1.72 | 1.23 | 1.16 | 1.07 | 0.94 | 0.86 | 0.98 | 0.86 |
| | | 8192 | 1.53e+03 | 1.85 | 1.32 | 1.25 | 1.16 | 1.01 | 0.94 | 1.06 | 0.92 |
| | | 16384 | 3.12e+03 | 1.84 | 1.28 | 1.19 | 1.14 | 1.01 | 0.94 | 1.05 | 0.92 |
| RTX5080 | Llama3.2-1B | 64 | 5.57e+01 | 0.89 | 0.76 | 0.75 | 0.67 | 0.66 | 0.67 | 0.67 | 0.62 |
| | | 128 | 9.48e+01 | 1.22 | 1.00 | 0.95 | 0.88 | 0.85 | 0.85 | 0.84 | 0.76 |
| | | 256 | 1.62e+02 | 1.50 | 1.17 | 1.06 | 0.95 | 0.90 | 0.93 | 0.92 | 0.85 |
| | | 512 | 2.97e+02 | 1.56 | 1.20 | 1.11 | 1.01 | 0.94 | 0.96 | 0.98 | 0.88 |
| | | 1024 | 5.64e+02 | 1.64 | 1.24 | 1.15 | 1.06 | 1.00 | 1.00 | 1.00 | 0.89 |
| | | 2048 | 1.09e+03 | 1.69 | 1.29 | 1.15 | 1.08 | 1.03 | 1.03 | 1.00 | 0.88 |
| | | 4096 | 2.14e+03 | 1.70 | 1.29 | 1.17 | 1.09 | 1.03 | 1.03 | 1.00 | 0.89 |
| | | 8192 | 4.22e+03 | 1.70 | 1.28 | 1.15 | 1.09 | 1.03 | 1.02 | 1.00 | 0.88 |
| | | 16384 | 8.40e+03 | 1.69 | 1.29 | 1.15 | 1.08 | 1.02 | 1.01 | 1.00 | 0.87 |
| | BitNet-2B | 64 | 6.17e+01 | 1.02 | 0.84 | 0.84 | 0.82 | 0.77 | 0.75 | 0.77 | 0.72 |
| | | 128 | 9.66e+01 | 1.28 | 0.98 | 0.96 | 0.94 | 0.87 | 0.84 | 0.86 | 0.79 |
| | | 256 | 1.72e+02 | 1.46 | 1.13 | 1.04 | 1.00 | 0.95 | 0.90 | 0.90 | 0.82 |
| | | 512 | 3.20e+02 | 1.60 | 1.15 | 1.12 | 1.07 | 1.00 | 0.96 | 0.96 | 0.86 |
| | | 1024 | 6.32e+02 | 1.70 | 1.27 | 1.17 | 1.11 | 1.05 | 1.01 | 1.00 | 0.89 |
| | | 2048 | 1.23e+03 | 1.71 | 1.29 | 1.17 | 1.10 | 1.06 | 1.02 | 1.00 | 0.89 |
| | | 4096 | 2.43e+03 | 1.73 | 1.28 | 1.17 | 1.11 | 1.06 | 1.01 | 1.00 | 0.89 |
| | | 8192 | 4.80e+03 | 1.72 | 1.27 | 1.15 | 1.09 | 1.05 | 1.00 | 0.99 | 0.87 |
| | | 16384 | 9.53e+03 | 1.70 | 1.25 | 1.15 | 1.08 | 1.03 | 0.98 | 0.98 | 0.87 |
| | Llama3.2-3B | 64 | 8.99e+01 | 1.20 | 0.99 | 0.94 | 0.86 | 0.85 | 0.76 | 0.86 | 0.74 |
| | | 128 | 1.43e+02 | 1.44 | 1.14 | 1.03 | 0.96 | 0.95 | 0.88 | 0.95 | 0.83 |
| | | 256 | 2.53e+02 | 1.50 | 1.19 | 1.08 | 1.00 | 0.97 | 0.92 | 0.95 | 0.85 |
| | | 512 | 4.68e+02 | 1.61 | 1.26 | 1.12 | 1.02 | 1.02 | 0.95 | 0.98 | 0.85 |
| | | 1024 | 9.10e+02 | 1.73 | 1.32 | 1.17 | 1.07 | 1.06 | 1.01 | 1.00 | 0.88 |
| | | 2048 | 1.77e+03 | 1.71 | 1.30 | 1.17 | 1.08 | 1.06 | 1.03 | 1.00 | 0.89 |
| | | 4096 | 3.50e+03 | 1.73 | 1.32 | 1.16 | 1.08 | 1.07 | 1.03 | 1.00 | 0.88 |
| | | 8192 | 6.93e+03 | 1.72 | 1.30 | 1.16 | 1.07 | 1.05 | 1.03 | 1.00 | 0.88 |
| | | 16384 | 1.38e+04 | 1.71 | 1.30 | 1.16 | 1.07 | 1.05 | 1.01 | 1.00 | 0.87 |
| | Qwen2.5-7B | 64 | 2.43e+02 | 1.47 | 1.12 | 0.99 | 0.90 | 0.81 | 0.76 | 0.76 | 0.67 |
| | | 128 | 3.02e+02 | 1.52 | 1.19 | 1.06 | 0.94 | 0.89 | 0.88 | 0.87 | 0.76 |
| | | 256 | 5.56e+02 | 1.64 | 1.28 | 1.14 | 1.05 | 1.03 | 1.01 | 0.97 | 0.87 |
| | | 512 | 1.06e+03 | 1.69 | 1.31 | 1.17 | 1.08 | 1.05 | 1.03 | 1.00 | 0.88 |
| | | 1024 | 2.05e+03 | 1.72 | 1.31 | 1.18 | 1.09 | 1.04 | 1.04 | 1.00 | 0.88 |
| | | 2048 | 4.06e+03 | 1.76 | 1.32 | 1.17 | 1.09 | 1.04 | 1.03 | 1.00 | 0.88 |
| | | 4096 | 8.02e+03 | 1.72 | 1.30 | 1.15 | 1.07 | 1.03 | 1.01 | 0.99 | 0.87 |
| | | 8192 | 1.60e+04 | 1.72 | 1.30 | 1.16 | 1.08 | 1.03 | 1.01 | 0.99 | 0.87 |
| | | 16384 | 3.17e+04 | 1.71 | 1.29 | 1.15 | 1.07 | 1.03 | 1.01 | 0.98 | 0.87 |
| | Qwen2.5-14B | 64 | 2.74e+02 | 1.53 | 1.17 | 1.04 | 0.88 | 0.82 | 0.79 | 0.79 | 0.69 |
| | | 128 | 3.47e+02 | 1.54 | 1.21 | 1.08 | 0.97 | 0.90 | 0.86 | 0.87 | 0.77 |
| | | 256 | 6.39e+02 | 1.69 | 1.30 | 1.16 | 1.10 | 1.05 | 1.02 | 1.02 | 0.89 |
| | | 512 | 1.23e+03 | 1.73 | 1.32 | 1.16 | 1.10 | 1.05 | 1.03 | 1.01 | 0.88 |
| | | 1024 | 2.43e+03 | 1.76 | 1.33 | 1.19 | 1.13 | 1.07 | 1.05 | 1.03 | 0.89 |
| | | 2048 | 4.80e+03 | 1.75 | 1.32 | 1.17 | 1.10 | 1.06 | 1.03 | 1.01 | 0.88 |
| | | 4096 | 9.50e+03 | 1.75 | 1.31 | 1.16 | 1.09 | 1.04 | 1.01 | 1.00 | 0.87 |
| | | 8192 | 1.88e+04 | 1.74 | 1.31 | 1.16 | 1.09 | 1.04 | 1.01 | 1.00 | 0.87 |
| | | 16384 | 3.74e+04 | 1.72 | 1.30 | 1.15 | 1.08 | 1.03 | 1.01 | 0.99 | 0.87 |
| GB10 | Llama3.2-1B | 64 | 2.06e+02 | 2.79 | 1.47 | 1.33 | 1.08 | 1.04 | 0.99 | 1.06 | 0.87 |
| | | 128 | 2.28e+02 | 2.18 | 1.37 | 1.28 | 1.08 | 1.05 | 0.98 | 1.05 | 0.91 |
| | | 256 | 2.96e+02 | 1.72 | 1.30 | 1.21 | 1.06 | 1.04 | 1.00 | 1.04 | 0.91 |
| | | 512 | 4.10e+02 | 1.37 | 1.08 | 0.96 | 0.88 | 0.85 | 0.86 | 0.83 | 0.72 |
| | | 1024 | 7.06e+02 | 1.22 | 0.96 | 0.85 | 0.80 | 0.75 | 0.75 | 0.74 | 0.64 |
| | | 2048 | 1.43e+03 | 1.27 | 0.96 | 0.84 | 0.80 | 0.75 | 0.74 | 0.73 | 0.63 |
| | | 4096 | 2.74e+03 | 1.23 | 0.93 | 0.82 | 0.77 | 0.72 | 0.72 | 0.70 | 0.62 |
| | | 8192 | 5.56e+03 | 1.24 | 0.93 | 0.83 | 0.78 | 0.73 | 0.73 | 0.72 | 0.63 |
| | | 16384 | 1.11e+04 | 1.22 | 0.93 | 0.83 | 0.77 | 0.74 | 0.73 | 0.71 | 0.63 |
| | BitNet-2B | 64 | 2.19e+02 | 2.38 | 1.35 | 1.23 | 1.14 | 0.99 | 0.95 | 1.02 | 0.84 |
| | | 128 | 2.49e+02 | 2.05 | 1.27 | 1.14 | 1.09 | 0.99 | 0.95 | 1.00 | 0.85 |
| | | 256 | 3.20e+02 | 1.69 | 1.19 | 1.14 | 1.07 | 0.98 | 0.96 | 0.98 | 0.88 |
| | | 512 | 4.52e+02 | 1.33 | 1.01 | 0.93 | 0.87 | 0.83 | 0.80 | 0.81 | 0.71 |

### Model Kernel (FP8) (cont.)

| GPU | Model | M | cuBLASLt Latency(μs) | Speedup Ratio under Different Sparsity | | | | | | | |
|---|---|---|---|---|---|---|---|---|---|---|---|
| | | | | 2:4 | 4:6 | 6:8 | 8:10 | 10:12 | 12:14 | 14:16 | ∞:∞ |
| | | 1024 | 7.92e+02 | 1.24 | 0.92 | 0.85 | 0.79 | 0.77 | 0.74 | 0.72 | 0.64 |
| | | 2048 | 1.52e+03 | 1.22 | 0.91 | 0.83 | 0.78 | 0.74 | 0.70 | 0.70 | 0.62 |
| | | 4096 | 3.07e+03 | 1.21 | 0.90 | 0.81 | 0.77 | 0.74 | 0.70 | 0.70 | 0.61 |
| | | 8192 | 6.15e+03 | 1.21 | 0.90 | 0.81 | 0.76 | 0.74 | 0.70 | 0.70 | 0.61 |
| | | 16384 | 1.24e+04 | 1.22 | 0.90 | 0.82 | 0.77 | 0.74 | 0.70 | 0.70 | 0.60 |
| | Llama3.2-3B | 64 | 3.71e+02 | 2.35 | 1.62 | 1.36 | 1.12 | 1.14 | 0.95 | 1.10 | 0.93 |
| | | 128 | 4.00e+02 | 2.12 | 1.51 | 1.33 | 1.11 | 1.13 | 0.97 | 1.08 | 0.90 |
| | | 256 | 4.70e+02 | 1.80 | 1.40 | 1.23 | 1.03 | 1.06 | 0.95 | 1.04 | 0.90 |
| | | 512 | 6.64e+02 | 1.41 | 1.11 | 0.99 | 0.90 | 0.88 | 0.88 | 0.83 | 0.77 |
| | | 1024 | 1.09e+03 | 1.23 | 0.94 | 0.82 | 0.76 | 0.75 | 0.73 | 0.71 | 0.61 |
| | | 2048 | 2.31e+03 | 1.29 | 0.97 | 0.85 | 0.79 | 0.76 | 0.74 | 0.73 | 0.64 |
| | | 4096 | 4.45e+03 | 1.23 | 0.92 | 0.81 | 0.75 | 0.74 | 0.71 | 0.70 | 0.61 |
| | | 8192 | 9.11e+03 | 1.24 | 0.94 | 0.83 | 0.77 | 0.76 | 0.73 | 0.71 | 0.63 |
| | | 16384 | 1.83e+04 | 1.24 | 0.94 | 0.83 | 0.78 | 0.76 | 0.73 | 0.71 | 0.63 |
| | Qwen2.5-7B | 64 | 1.08e+03 | 1.97 | 1.26 | 1.27 | 1.14 | 0.92 | 1.02 | 1.05 | 0.91 |
| | | 128 | 1.15e+03 | 1.94 | 1.25 | 1.25 | 1.16 | 0.94 | 1.03 | 1.05 | 0.94 |
| | | 256 | 1.29e+03 | 1.85 | 1.27 | 1.23 | 1.15 | 0.96 | 1.05 | 1.05 | 0.94 |
| | | 512 | 1.61e+03 | 1.55 | 1.12 | 1.01 | 0.92 | 0.85 | 0.86 | 0.84 | 0.74 |
| | | 1024 | 2.78e+03 | 1.30 | 0.95 | 0.85 | 0.79 | 0.73 | 0.73 | 0.73 | 0.65 |
| | | 2048 | 5.58e+03 | 1.29 | 0.94 | 0.85 | 0.78 | 0.72 | 0.70 | 0.72 | 0.64 |
| | | 4096 | 1.12e+04 | 1.23 | 0.90 | 0.81 | 0.76 | 0.68 | 0.67 | 0.70 | 0.61 |
| | | 8192 | 2.25e+04 | 1.30 | 0.95 | 0.86 | 0.80 | 0.72 | 0.72 | 0.74 | 0.65 |
| | | 16384 | 4.45e+04 | 1.26 | 0.91 | 0.83 | 0.78 | 0.70 | 0.69 | 0.72 | 0.63 |
| | Qwen2.5-14B | 64 | 1.39e+03 | 2.36 | 1.42 | 1.36 | 1.25 | 1.04 | 0.96 | 1.13 | 0.94 |
| | | 128 | 1.44e+03 | 2.20 | 1.42 | 1.36 | 1.22 | 1.01 | 0.93 | 1.12 | 0.97 |
| | | 256 | 1.55e+03 | 1.99 | 1.38 | 1.31 | 1.19 | 1.03 | 0.95 | 1.11 | 0.92 |
| | | 512 | 1.97e+03 | 1.59 | 1.21 | 1.07 | 1.00 | 0.97 | 0.90 | 0.93 | 0.81 |
| | | 1024 | 3.12e+03 | 1.28 | 0.95 | 0.83 | 0.78 | 0.75 | 0.70 | 0.71 | 0.63 |
| | | 2048 | 6.21e+03 | 1.24 | 0.94 | 0.83 | 0.78 | 0.75 | 0.69 | 0.72 | 0.63 |
| | | 4096 | 1.26e+04 | 1.26 | 0.96 | 0.84 | 0.79 | 0.76 | 0.71 | 0.72 | 0.64 |
| | | 8192 | 2.48e+04 | 1.25 | 0.95 | 0.83 | 0.78 | 0.75 | 0.70 | 0.72 | 0.63 |
| | | 16384 | 5.00e+04 | 1.26 | 0.96 | 0.84 | 0.79 | 0.76 | 0.71 | 0.72 | 0.63 |

**Qwen-2.5-14B on A100 (INT8).** This configuration achieves excellent theoretical alignment: 2:4 reaches $2.08$–$2.18\times$; 4:6 reaches $1.46$–$1.63\times$ (theoretical: $1.50\times$); 6:8 reaches $1.33$–$1.50\times$ (theoretical: $1.33\times$). The results validate that SlideSparse correctly realizes the expected compute reduction for realistic model dimensions.

**Cross-Model Consistency.** Larger models (Qwen-7B, Qwen-14B) consistently achieve higher speedups than smaller models (Llama-1B, Llama-3B). This is expected: larger $(N, K)$ dimensions yield better hardware utilization and amortize sparse format overhead.

### D.3.3. KERNEL PERFORMANCE ANALYSIS

**Why B200 INT8 Speedups Are Exceptionally High.** The extraordinary B200 INT8 speedups (up to $6.47\times$) are primarily due to *cuBLASLt's suboptimal INT8 baseline* on Blackwell, not superior sparse kernel efficiency. Evidence: even $\infty{:}\infty$ (dense weights in sliding format, theoretically $1.0\times$) achieves $3.09\times$ speedup—an impossible result if the baseline were optimal. We expect these gains to normalize as NVIDIA releases optimized INT8 drivers for Blackwell.

**The M Threshold Effect.** The consistent $M{\approx}1024$ crossover point across GPUs reflects the fundamental overhead structure of sparse operations: below this threshold, sparse metadata handling and tensor core scheduling overhead exceed computation savings. For practical LLM inference, prefill workloads ($M \geq 4096$) are ideal targets for SlideSparse acceleration.

### D.4. Full Accuracy Breakdown

Tables 2 and 3 report per-benchmark accuracy for all sparsity levels and pruning methods, corresponding to the averages plotted in Figure 2. Table 4 summarizes relative gaps from the dense baseline.

*Table 2.* **Qwen2.5-7B (FP16)** accuracy under $(2N{-}2) : 2N$ structured sparsity. Seven commonsense tasks are 0-shot; MMLU and GSM8K are 5-shot.

| Method | Sparsity | PIQA | ARC-E | ARC-C | HSwag | Wino | BoolQ | OBQA | MMLU | GSM8K | Avg |
|---|---|---|---|---|---|---|---|---|---|---|---|
| Dense | 0% | 79.8 | 77.3 | 51.1 | 78.9 | 72.9 | 84.6 | 47.2 | 74.3 | 82.9 | **72.1** |
| Wanda | 8:10 | 79.8 | 77.5 | 50.9 | 78.1 | 72.8 | 85.1 | 45.8 | 73.5 | 81.7 | 71.7 |
| Wanda | 6:8 | 79.5 | 75.9 | 48.8 | 77.7 | 72.2 | 84.5 | 45.0 | 72.9 | 82.3 | 71.0 |
| Wanda | 4:6 | 78.1 | 75.8 | 48.9 | 75.3 | 70.8 | 84.6 | 44.6 | 70.5 | 75.6 | 69.4 |
| Wanda | 2:4 | 72.1 | 70.1 | 42.2 | 58.7 | 65.4 | 72.2 | 36.4 | 52.8 | 30.3 | 55.6 |
| SparseGPT | 8:10 | 79.7 | 78.5 | 52.1 | 78.1 | 72.5 | 84.9 | 46.2 | 73.6 | 81.1 | 71.8 |
| SparseGPT | 6:8 | 79.4 | 72.0 | 49.7 | 77.8 | 72.1 | 83.9 | 44.8 | 72.6 | 80.7 | 70.3 |
| SparseGPT | 4:6 | 77.1 | 70.9 | 45.9 | 72.9 | 69.6 | 84.7 | 43.8 | 67.2 | 62.3 | 66.0 |
| SparseGPT | 2:4 | 72.1 | 71.2 | 43.4 | 62.0 | 65.5 | 74.3 | 37.8 | 55.8 | 30.3 | 56.9 |

*Table 3.* **Qwen2.5-14B (BF16)** accuracy under $(2N{-}2) : 2N$ structured sparsity. Same evaluation protocol as Table 2.

| Method | Sparsity | PIQA | ARC-E | ARC-C | HSwag | Wino | BoolQ | OBQA | MMLU | GSM8K | Avg |
|---|---|---|---|---|---|---|---|---|---|---|---|
| Dense | 0% | 82.1 | 79.3 | 59.4 | 83.0 | 74.7 | 85.4 | 45.4 | 79.7 | 88.2 | **75.2** |
| Wanda | 8:10 | 81.8 | 80.8 | 59.1 | 82.2 | 76.6 | 86.6 | 46.2 | 79.1 | 82.9 | 75.0 |
| Wanda | 6:8 | 81.6 | 80.8 | 56.4 | 81.4 | 74.2 | 85.3 | 45.6 | 78.4 | 86.9 | 74.5 |
| Wanda | 4:6 | 81.7 | 83.8 | 58.3 | 79.4 | 74.7 | 85.0 | 46.0 | 77.1 | 81.9 | 74.2 |
| Wanda | 2:4 | 74.1 | 68.1 | 40.2 | 64.8 | 70.4 | 79.0 | 41.0 | 57.9 | 47.3 | 60.3 |
| SparseGPT | 8:10 | 81.8 | 81.6 | 59.4 | 82.0 | 76.2 | 86.2 | 45.0 | 79.3 | 82.4 | 74.9 |
| SparseGPT | 6:8 | 81.6 | 81.4 | 57.3 | 81.3 | 75.8 | 85.9 | 45.6 | 78.4 | 85.5 | 74.8 |
| SparseGPT | 4:6 | 81.1 | 83.0 | 56.3 | 79.5 | 75.8 | 82.9 | 44.0 | 76.4 | 79.1 | 73.1 |
| SparseGPT | 2:4 | 74.8 | 72.1 | 43.1 | 66.0 | 70.7 | 66.1 | 40.0 | 60.9 | 43.7 | 59.7 |

### D.5. End-to-End Inference Performance

End-to-end benchmarks measure actual LLM inference throughput (tokens/s), capturing real-world overheads including memory management, kernel launch latency, attention computation, and KV cache access. We evaluate both prefill

*Table 4.* **Relative accuracy gap (%) from dense baseline.** Computed as $(\text{Avg}_{\text{pruned}} - \text{Avg}_{\text{dense}})/\text{Avg}_{\text{dense}} \times 100$.

| Model | Method | Sparsity | PIQA | ARC-E | ARC-C | HSwag | Wino | BoolQ | OBQA | MMLU | GSM8K | Avg gap |
|---|---|---|---|---|---|---|---|---|---|---|---|---|
| 7B | Wanda | 8:10 | +0.0 | +0.2 | −0.5 | −1.0 | −0.2 | +0.6 | −3.0 | −1.1 | −1.4 | −0.6 |
| 7B | Wanda | 6:8 | −0.3 | −1.8 | −4.5 | −1.5 | −0.9 | −0.1 | −4.7 | −1.8 | −0.8 | −1.5 |
| 7B | Wanda | 4:6 | −2.1 | −1.9 | −4.3 | −4.6 | −2.9 | +0.0 | −5.5 | −5.1 | −8.8 | −3.7 |
| 7B | Wanda | 2:4 | −9.6 | −9.3 | −17.4 | −25.6 | −10.3 | −14.7 | −22.9 | −28.9 | −63.4 | −22.9 |
| 7B | SparseGPT | 8:10 | −0.1 | +1.6 | +2.0 | −1.0 | −0.5 | +0.4 | −2.1 | −0.9 | −2.2 | −0.4 |
| 7B | SparseGPT | 6:8 | −0.5 | −6.9 | −2.7 | −1.4 | −1.1 | −0.8 | −5.1 | −2.3 | −2.6 | −2.5 |
| 7B | SparseGPT | 4:6 | −3.4 | −8.3 | −10.2 | −7.6 | −4.5 | +0.1 | −7.2 | −9.6 | −24.8 | −8.5 |
| 7B | SparseGPT | 2:4 | −9.6 | −7.9 | −15.1 | −21.4 | −10.2 | −12.2 | −19.9 | −24.9 | −63.4 | −21.1 |
| 14B | Wanda | 8:10 | −0.4 | +1.9 | −0.5 | −1.0 | +2.5 | +1.4 | +1.8 | −0.8 | −6.0 | −0.3 |
| 14B | Wanda | 6:8 | −0.6 | +1.9 | −5.1 | −1.9 | −0.7 | −0.1 | +0.4 | −1.6 | −1.5 | −0.9 |
| 14B | Wanda | 4:6 | −0.5 | +5.7 | −1.9 | −4.3 | +0.0 | −0.5 | +1.3 | −3.3 | −7.1 | −1.3 |
| 14B | Wanda | 2:4 | −9.7 | −14.1 | −32.3 | −21.9 | −5.8 | −7.5 | −9.7 | −27.4 | −46.4 | −19.8 |
| 14B | SparseGPT | 8:10 | −0.4 | +2.9 | +0.0 | −1.2 | +2.0 | +0.9 | −0.9 | −0.5 | −6.6 | −0.4 |
| 14B | SparseGPT | 6:8 | −0.6 | +2.6 | −3.5 | −2.0 | +1.5 | +0.6 | +0.4 | −1.6 | −3.0 | −0.5 |
| 14B | SparseGPT | 4:6 | −1.2 | +4.7 | −5.2 | −4.2 | +1.5 | −2.9 | −3.1 | −4.1 | −10.3 | −2.8 |
| 14B | SparseGPT | 2:4 | −8.9 | −9.1 | −27.4 | −20.5 | −5.4 | −22.6 | −11.9 | −23.6 | −50.5 | −20.6 |

(compute-bound) and decode (memory-bound) stages.

### D.5.1. PREFILL STAGE RESULTS

Prefill processes the entire input prompt and is **compute-bound**, making it an ideal target for GEMM acceleration. We configure $M = \texttt{max\_num\_seqs} \times \texttt{prompt\_len}$ with $M \in \{512, 1024, 2048, 4096, 8192, 16384, 32768, 65536\}$. The following tables present complete prefill throughput speedup results across all tested configurations.

## Prefill (INT8)

| GPU | Model | Batchsize M | cuBLASLt Throughput | 2:4 | 4:6 | 6:8 | 8:10 |
|---|---|---|---|---|---|---|---|
| A100 | Llama3.2-1B | 512 | 1.19e+04 | 0.88 | 0.89 | 0.84 | 0.88 |
| | | 1024 | 2.46e+04 | 0.85 | 0.71 | 0.84 | 0.84 |
| | | 2048 | 4.79e+04 | 0.86 | 0.81 | 0.85 | 0.88 |
| | | 4096 | 9.03e+04 | 0.88 | 0.83 | 0.88 | 0.87 |
| | | 8192 | 1.03e+05 | 1.37 | 1.21 | 1.23 | 1.11 |
| | | 16384 | 1.06e+05 | 1.48 | 1.28 | 1.24 | 1.12 |
| | | 32768 | 1.07e+05 | 1.50 | 1.29 | 1.24 | 1.12 |
| | BitNet-2B | 512 | 9.24e+03 | 1.02 | 1.05 | 1.08 | 1.09 |
| | | 1024 | 1.83e+04 | 1.03 | 1.06 | 1.13 | 1.11 |
| | | 2048 | 3.73e+04 | 1.03 | 1.08 | 1.09 | 1.09 |
| | | 4096 | 5.09e+04 | 1.48 | 1.26 | 1.26 | 1.21 |
| | | 8192 | 5.33e+04 | 1.52 | 1.26 | 1.26 | 1.21 |
| | | 16384 | 5.43e+04 | 1.53 | 1.26 | 1.26 | 1.21 |
| | | 32768 | 5.46e+04 | 1.54 | 1.27 | 1.26 | 1.21 |
| | Llama3.2-3B | 512 | 9.71e+03 | 1.09 | 1.10 | 1.09 | 1.06 |
| | | 1024 | 2.03e+04 | 1.04 | 1.04 | 1.05 | 1.02 |
| | | 2048 | 3.77e+04 | 1.13 | 1.15 | 1.13 | 1.14 |
| | | 4096 | 4.09e+04 | 1.59 | 1.36 | 1.28 | 1.20 |
| | | 8192 | 4.21e+04 | 1.59 | 1.36 | 1.27 | 1.19 |
| | | 16384 | 4.27e+04 | 1.61 | 1.37 | 1.29 | 1.20 |
| | | 32768 | 4.28e+04 | 1.62 | 1.37 | 1.29 | 1.20 |
| | Qwen2.5-7B | 512 | 1.02e+04 | 1.05 | 1.10 | 1.12 | 1.09 |
| | | 1024 | 1.80e+04 | 1.24 | 1.25 | 1.18 | 1.14 |
| | | 2048 | 1.98e+04 | 1.63 | 1.34 | 1.26 | 1.20 |
| | | 4096 | 2.06e+04 | 1.69 | 1.37 | 1.30 | 1.24 |
| | | 8192 | 2.08e+04 | 1.72 | 1.40 | 1.32 | 1.26 |
| | | 16384 | 2.09e+04 | 1.75 | 1.41 | 1.34 | 1.27 |
| | | 32768 | 2.08e+04 | 1.75 | 1.41 | 1.34 | 1.26 |
| | Qwen2.5-14B | 512 | 6.42e+03 | 1.03 | 1.04 | 1.01 | 1.06 |
| | | 1024 | 9.68e+03 | 1.38 | 1.31 | 1.27 | 1.19 |
| | | 2048 | 1.03e+04 | 1.73 | 1.42 | 1.33 | 1.28 |
| | | 4096 | 1.05e+04 | 1.75 | 1.43 | 1.34 | 1.28 |
| | | 8192 | 1.06e+04 | 1.77 | 1.43 | 1.33 | 1.27 |
| | | 16384 | 1.06e+04 | 1.77 | 1.43 | 1.34 | 1.27 |
| | | 32768 | 1.07e+04 | 1.77 | 1.42 | 1.33 | 1.26 |
| RTX4090 | Llama3.2-1B | 512 | 9.21e+03 | 0.91 | 0.91 | 0.88 | 0.90 |
| | | 1024 | 1.87e+04 | 0.85 | 0.88 | 0.89 | 0.87 |
| | | 2048 | 3.68e+04 | 0.91 | 0.88 | 0.91 | 0.89 |
| | | 4096 | 7.02e+04 | 0.92 | 0.90 | 0.93 | 0.91 |
| | | 8192 | 9.14e+04 | 1.24 | 1.14 | 1.10 | 1.05 |
| | | 16384 | 9.09e+04 | 1.30 | 1.16 | 1.13 | 1.06 |
| | | 32768 | 9.24e+04 | 1.30 | 1.16 | 1.12 | 1.04 |
| | BitNet-2B | 512 | 7.79e+03 | 1.11 | 1.14 | 1.10 | 1.12 |
| | | 1024 | 1.60e+04 | 1.11 | 1.06 | 1.07 | 1.04 |
| | | 2048 | 3.15e+04 | 1.12 | 1.13 | 1.07 | 1.13 |
| | | 4096 | 4.72e+04 | 1.43 | 1.16 | 1.23 | 1.13 |
| | | 8192 | 5.36e+04 | 1.31 | 1.06 | 1.05 | 1.04 |
| | | 16384 | 5.32e+04 | 1.30 | 1.07 | 1.14 | 1.07 |
| | | 32768 | 5.45e+04 | 1.31 | 1.11 | 1.11 | 1.08 |
| | Llama3.2-3B | 512 | 6.64e+03 | 1.08 | 1.12 | 1.09 | 1.12 |
| | | 1024 | 1.38e+04 | 1.01 | 1.12 | 1.07 | 1.11 |
| | | 2048 | 2.78e+04 | 1.09 | 1.12 | 1.11 | 1.06 |
| | | 4096 | 4.04e+04 | 1.30 | 1.20 | 1.11 | 1.09 |
| | | 8192 | 3.94e+04 | 1.35 | 1.19 | 1.13 | 1.09 |
| | | 16384 | 3.96e+04 | 1.34 | 1.18 | 1.12 | 1.09 |
| | | 32768 | 3.98e+04 | 1.34 | 1.18 | 1.13 | 1.08 |
| | Qwen2.5-7B | 512 | 8.27e+03 | 1.06 | 1.06 | 1.07 | 1.06 |
| | | 1024 | 1.63e+04 | 1.07 | 1.09 | 1.05 | 1.04 |
| | | 2048 | 2.03e+04 | 1.36 | 1.18 | 1.10 | 1.06 |
| | | 4096 | 2.07e+04 | 1.42 | 1.21 | 1.13 | 1.08 |
| | | 8192 | 2.12e+04 | 1.39 | 1.18 | 1.11 | 1.08 |
| | | 16384 | 2.13e+04 | 1.39 | 1.18 | 0.58 | 0.40 |
| | | 32768 | 2.14e+04 | 0.41 | 0.31 | 0.27 | 0.25 |
| | Qwen2.5-14B | 512 | 4.49e+03 | 1.06 | 1.11 | 1.07 | |
| | | 1024 | 9.26e+03 | 1.02 | 1.07 | 1.07 | |
| | | 2048 | 1.11e+04 | 1.43 | 1.21 | 1.14 | |
| | | 4096 | 1.14e+04 | 1.40 | 1.19 | 1.12 | |
| | | 8192 | 1.14e+04 | 1.40 | 1.17 | 1.12 | |
| | | 16384 | 1.13e+04 | 1.40 | 1.19 | | |
| | | 32768 | | | | | |
| H100 | Llama3.2-1B | 512 | 1.82e+04 | 0.87 | 0.90 | 0.85 | 0.91 |
| | | 1024 | 3.56e+04 | 0.88 | 0.90 | 0.87 | 0.90 |
| | | 2048 | 6.84e+04 | 0.93 | 0.83 | 0.90 | 0.93 |
| | | 4096 | 1.31e+05 | 0.88 | 0.86 | 0.87 | 0.86 |
| | | 8192 | 1.59e+05 | 1.09 | 1.02 | 0.96 | 0.95 |
| | | 16384 | 1.68e+05 | 1.20 | 1.12 | 1.12 | 1.02 |
| | | 32768 | 1.75e+05 | 1.30 | 1.12 | 1.11 | 1.04 |
| | BitNet-2B | 512 | 1.49e+04 | 1.10 | 1.15 | 1.15 | 1.09 |

### *Prefill (INT8) (cont.)*

| GPU | Model | Batchsize M | cuBLASLt Throughput | Speedup Ratio under Different Sparsity | | | |
|---|---|---|---|---|---|---|---|
| | | | | 2:4 | 4:6 | 6:8 | 8:10 |
| | | 1024 | 3.06e+14 | 1.10 | 1.14 | 1.08 | 1.10 |
| | | 2048 | 5.67e+04 | 1.18 | 1.12 | 1.14 | 1.16 |
| | | 4096 | 7.93e+04 | 1.20 | 1.11 | 1.09 | 1.05 |
| | | 8192 | 8.87e+04 | 1.18 | 1.10 | 1.05 | 1.02 |
| | | 16384 | 9.22e+04 | 1.25 | 1.09 | 1.05 | 1.03 |
| | | 32768 | 9.41e+04 | 1.25 | 1.09 | 1.05 | 1.02 |
| | Llama3.2-3B | 512 | 1.42e+04 | 1.15 | 1.25 | 1.23 | 1.19 |
| | | 1024 | 3.10e+04 | 1.11 | 1.15 | 1.16 | 1.15 |
| | | 2048 | 5.72e+04 | 1.17 | 1.17 | 1.12 | 1.09 |
| | | 4096 | 6.53e+04 | 1.31 | 1.16 | 1.12 | 1.07 |
| | | 8192 | 7.19e+04 | 1.29 | 1.13 | 1.08 | 1.04 |
| | | 16384 | 7.43e+04 | 1.29 | 1.14 | 1.08 | 1.03 |
| | | 32768 | 7.58e+04 | 1.31 | 1.14 | 1.07 | 1.02 |
| | Qwen2.5-7B | 512 | 1.46e+04 | 1.15 | 1.28 | 1.24 | 1.25 |
| | | 1024 | 2.91e+04 | 1.21 | 1.16 | 1.15 | 1.11 |
| | | 2048 | 3.65e+04 | 1.30 | 1.09 | 1.03 | 1.00 |
| | | 4096 | 4.00e+04 | 1.26 | 1.05 | 1.00 | 0.96 |
| | | 8192 | 3.70e+04 | 1.40 | 1.18 | 1.11 | 1.06 |
| | | 16384 | 3.77e+04 | 1.41 | 1.18 | 1.12 | 1.08 |
| | | 32768 | 3.77e+04 | 1.42 | 1.19 | 1.12 | 1.06 |
| | Qwen2.5-14B | 512 | 1.01e+04 | 1.19 | 1.14 | 1.16 | 1.16 |
| | | 1024 | 1.68e+04 | 1.34 | 1.13 | 1.07 | 1.04 |
| | | 2048 | 1.82e+04 | 1.38 | 1.19 | 1.12 | 1.08 |
| | | 4096 | 1.92e+04 | 1.35 | 1.18 | 1.11 | 1.07 |
| | | 8192 | 1.95e+04 | 1.45 | 1.20 | 1.11 | 1.07 |
| | | 16384 | 1.99e+04 | 1.43 | 1.19 | 1.11 | 1.07 |
| | | 32768 | 2.00e+04 | 1.43 | 1.18 | 1.11 | 1.05 |
| B200 | Llama3.2-1B | 512 | 2.09e+04 | 1.00 | 1.06 | 0.97 | 1.05 |
| | | 1024 | 4.14e+04 | 0.89 | 1.01 | 0.93 | 1.02 |
| | | 2048 | 8.15e+04 | 1.01 | 1.04 | 0.95 | 0.97 |
| | | 4096 | 1.73e+05 | 0.96 | 0.95 | 0.95 | 0.96 |
| | | 8192 | 3.22e+05 | 0.93 | 0.95 | 0.96 | 0.96 |
| | | 16384 | 4.71e+05 | 1.00 | 1.00 | 1.00 | 0.95 |
| | | 32768 | 4.83e+05 | 0.99 | 0.99 | 0.99 | 0.98 |
| | BitNet-2B | 512 | 1.54e+04 | 1.08 | 1.10 | 1.10 | 1.12 |
| | | 1024 | 3.20e+04 | 1.02 | 1.06 | 1.06 | 1.05 |
| | | 2048 | 6.41e+04 | 1.05 | 1.06 | 1.12 | 1.06 |
| | | 4096 | 1.25e+05 | 1.09 | 1.09 | 1.09 | 1.08 |
| | | 8192 | 2.30e+05 | 1.13 | 1.07 | 1.07 | 1.05 |
| | | 16384 | 2.69e+05 | 1.20 | 1.04 | 1.04 | 1.01 |
| | | 32768 | 2.82e+05 | 1.22 | 1.05 | 1.04 | 1.00 |
| | Llama3.2-3B | 512 | 1.60e+04 | 1.05 | 1.10 | 1.09 | 1.04 |
| | | 1024 | 3.21e+04 | 1.07 | 1.12 | 1.11 | 1.10 |
| | | 2048 | 6.22e+04 | 1.12 | 1.13 | 1.11 | 1.12 |
| | | 4096 | 1.28e+05 | 1.04 | 1.08 | 1.12 | 1.11 |
| | | 8192 | 2.10e+05 | 1.21 | 1.08 | 1.03 | 0.99 |
| | | 16384 | 2.26e+05 | 1.27 | 1.11 | 1.06 | 1.00 |
| | | 32768 | 2.34e+05 | 1.29 | 1.12 | 1.07 | 1.00 |
| | Qwen2.5-7B | 512 | 1.70e+04 | 1.13 | 1.13 | 1.12 | 1.15 |
| | | 1024 | 3.58e+04 | 1.11 | 1.11 | 1.14 | 1.13 |
| | | 2048 | 7.22e+04 | 1.03 | 1.11 | 1.10 | 1.09 |
| | | 4096 | 1.11e+05 | 1.27 | 1.10 | 1.06 | 1.02 |
| | | 8192 | 1.21e+05 | 1.31 | 1.09 | 1.05 | 1.01 |
| | | 16384 | 1.21e+05 | 1.38 | 1.14 | 1.08 | 1.07 |
| | | 32768 | 1.26e+05 | 1.35 | 1.14 | 1.07 | 1.05 |
| | Qwen2.5-14B | 512 | 1.01e+04 | 1.10 | 1.15 | 1.21 | 1.14 |
| | | 1024 | 2.13e+04 | 1.08 | 1.10 | 1.13 | 1.13 |
| | | 2048 | 4.22e+04 | 1.14 | 1.15 | 1.10 | 1.08 |
| | | 4096 | 6.17e+04 | 1.31 | 1.11 | 1.04 | 1.02 |
| | | 8192 | 6.46e+04 | 1.33 | 1.10 | 1.08 | 1.03 |
| | | 16384 | 6.40e+04 | 1.41 | 1.17 | 1.11 | 1.07 |
| | | 32768 | 6.61e+04 | 1.41 | 1.15 | 1.11 | 1.06 |
| RTX5080 | Llama3.2-1B | 512 | 3.43e+04 | 0.96 | 1.06 | 0.96 | 1.08 |
| | | 1024 | 7.16e+04 | 0.92 | 0.97 | 0.95 | 0.96 |
| | | 2048 | 8.09e+04 | 1.30 | 1.20 | 1.15 | 1.06 |
| | | 4096 | 8.25e+04 | 1.40 | 1.22 | 1.16 | 1.07 |
| | | 8192 | 8.49e+04 | 1.33 | 1.17 | 1.11 | 1.04 |
| | | 16384 | 8.41e+04 | 1.31 | 1.16 | 1.10 | 1.03 |
| | | 32768 | 8.39e+04 | 1.31 | 1.16 | 1.10 | 1.03 |
| | BitNet-2B | 512 | 2.68e+04 | 1.06 | 1.05 | 1.06 | 1.03 |
| | | 1024 | 4.34e+04 | 1.26 | 1.08 | 1.05 | 1.04 |
| | | 2048 | 4.61e+04 | 1.35 | 1.10 | 1.10 | 1.08 |
| | | 4096 | 4.49e+04 | 1.37 | 1.14 | 1.12 | 1.09 |
| | | 8192 | 4.35e+04 | 1.33 | 1.12 | 1.10 | 1.07 |
| | | 16384 | 4.31e+04 | 1.31 | 1.11 | 1.09 | 1.06 |
| | | 32768 | 4.30e+04 | 1.31 | 1.11 | 1.09 | 1.06 |
| | Llama3.2-3B | 512 | 2.16e+03 | 11.74 | 13.37 | 13.13 | 13.33 |
| | | 1024 | 3.26e+04 | 1.29 | 1.21 | 1.13 | 1.10 |

### *Prefill (INT8) (cont.)*

| GPU | Model | Batchsize M | cuBLASLt Throughput | Speedup Ratio under Different Sparsity | | | |
|---|---|---|---|---|---|---|---|
| | | | | 2:4 | 4:6 | 6:8 | 8:10 |
| | | 2048 | 3.67e+04 | 1.38 | 1.23 | 1.14 | 1.09 |
| | | 4096 | 3.60e+04 | 1.37 | 1.20 | 1.11 | 1.06 |
| | | 8192 | 3.54e+04 | 1.35 | 1.18 | 1.10 | 1.05 |
| | | 16384 | 3.51e+04 | 1.34 | 1.17 | 1.10 | 1.05 |
| | | 32768 | 3.51e+04 | 1.34 | 1.17 | 1.10 | 1.05 |
| | Qwen2.5-7B | 512 | 1.52e+04 | 1.32 | 1.22 | 1.14 | 1.08 |
| | | 1024 | 1.76e+04 | 1.35 | 1.17 | 1.09 | 1.04 |
| | | 2048 | 1.84e+04 | 1.40 | 1.15 | 1.08 | 1.03 |
| | | 4096 | 1.84e+04 | 1.38 | 1.16 | 1.09 | 1.03 |
| | | 8192 | 1.81e+04 | 1.38 | 1.17 | 1.10 | |
| | | 16384 | 1.82e+04 | 1.38 | | | |
| | | 32768 | | | | | |
| | Qwen2.5-14B | 512 | | | | | |
| | | 1024 | | | | | |
| | | 2048 | | | | | |
| | | 4096 | | | | | |
| | | 8192 | | | | | |
| | | 16384 | | | | | |
| | | 32768 | | | | | |
| GB10 | Llama3.2-1B | 512 | 2.16e+04 | 1.35 | 1.18 | 1.20 | 1.10 |
| | | 1024 | 2.42e+04 | 1.32 | 1.19 | 1.17 | 1.10 |
| | | 2048 | 2.64e+04 | 1.31 | 1.17 | 1.13 | 1.05 |
| | | 4096 | 2.60e+04 | 1.31 | 1.17 | 1.14 | 1.05 |
| | | 8192 | 2.59e+04 | 1.28 | 1.16 | 1.12 | 1.04 |
| | | 16384 | 2.62e+04 | 1.27 | 1.14 | 1.11 | 1.04 |
| | | 32768 | 2.66e+04 | 1.23 | 1.14 | 1.10 | 1.04 |
| | BitNet-2B | 512 | 1.12e+04 | 1.39 | 0.96 | 1.26 | 1.18 |
| | | 1024 | 1.32e+04 | 1.32 | 1.18 | 1.16 | 1.12 |
| | | 2048 | 1.37e+04 | 1.38 | 1.19 | 1.16 | 1.13 |
| | | 4096 | 1.30e+04 | 1.36 | 1.17 | 1.15 | 1.12 |
| | | 8192 | 1.34e+04 | 1.29 | 1.11 | 1.10 | 1.09 |
| | | 16384 | 1.36e+04 | 1.28 | 1.11 | 1.09 | 1.07 |
| | | 32768 | 1.37e+04 | 1.28 | 1.12 | 1.09 | 1.08 |
| | Llama3.2-3B | 512 | 2.27e+03 | 6.12 | 5.26 | 5.19 | 4.71 |
| | | 1024 | 1.14e+04 | 1.34 | 1.18 | 1.14 | 1.07 |
| | | 2048 | 1.12e+04 | 1.41 | 1.25 | 1.19 | 1.12 |
| | | 4096 | 1.03e+04 | 1.45 | 1.31 | 1.24 | 1.17 |
| | | 8192 | 1.11e+04 | 1.32 | 1.21 | 1.13 | 1.07 |
| | | 16384 | 1.13e+04 | 1.31 | 1.18 | 1.12 | 1.07 |
| | | 32768 | 3.84e+03 | 2.98 | 3.43 | 3.26 | 3.14 |
| | Qwen2.5-7B | 512 | 4.98e+03 | 1.24 | 1.07 | 1.06 | 0.97 |
| | | 1024 | 5.48e+03 | 1.38 | 1.19 | 1.11 | 1.05 |
| | | 2048 | 5.57e+03 | 1.39 | 1.19 | 1.12 | 1.08 |
| | | 4096 | 5.66e+03 | 1.37 | 1.14 | 1.08 | 1.05 |
| | | 8192 | 2.28e+03 | 3.35 | 2.82 | 2.54 | 2.53 |
| | | 16384 | 5.80e+03 | 0.23 | 1.11 | 1.06 | 0.13 |
| | | 32768 | 5.84e+03 | 1.33 | 1.11 | 1.05 | 1.01 |
| | Qwen2.5-14B | 512 | 2.76e+03 | 1.39 | 1.19 | 1.10 | 1.04 |
| | | 1024 | 3.10e+03 | 1.43 | 1.21 | 1.08 | 1.08 |
| | | 2048 | 3.02e+03 | 1.45 | 1.18 | 1.10 | 1.05 |
| | | 4096 | 3.17e+03 | 1.39 | 1.12 | 1.06 | 1.01 |
| | | 8192 | 3.22e+03 | 1.35 | 1.11 | 1.05 | 1.00 |
| | | 16384 | 3.22e+03 | 1.37 | 1.10 | 1.05 | 1.00 |
| | | 32768 | 3.26e+03 | 0.34 | 1.09 | 1.03 | 0.99 |

### Prefill (FP8)

| GPU | Model | Batchsize M | cuBLASLt Throughput | Speedup Ratio under Different Sparsity | | | |
|---|---|---|---|---|---|---|---|
| | | | | 2:4 | 4:6 | 6:8 | 8:10 |
| RTX4090 | Llama3.2-1B | 512 | 9.59e+03 | 0.91 | 0.88 | 0.87 | 0.88 |
| | | 1024 | 1.93e+04 | 0.86 | 0.87 | 0.84 | 0.85 |
| | | 2048 | 3.81e+04 | 0.88 | 0.87 | 0.86 | 0.86 |
| | | 4096 | 7.14e+04 | 0.91 | 0.91 | 0.90 | 0.86 |
| | | 8192 | 8.13e+04 | 1.28 | 1.15 | 1.10 | 1.05 |
| | | 16384 | 7.99e+04 | 1.34 | 1.18 | 1.13 | 1.07 |
| | | 32768 | 8.02e+04 | 1.36 | 1.19 | 1.14 | 1.08 |
| | BitNet-2B | 512 | 8.06e+03 | 1.05 | 1.03 | 1.06 | 1.02 |
| | | 1024 | 1.60e+04 | 1.03 | 1.02 | 1.10 | 1.05 |
| | | 2048 | 3.12e+04 | 1.07 | 1.07 | 1.01 | 1.05 |
| | | 4096 | 4.61e+04 | 1.33 | 1.19 | 1.17 | 1.13 |
| | | 8192 | 4.83e+04 | 1.31 | 1.14 | 1.09 | 1.09 |
| | | 16384 | 4.65e+04 | 1.38 | 1.20 | 1.14 | 1.12 |
| | | 32768 | 4.62e+04 | 1.38 | 1.22 | 1.14 | 1.13 |
| | Llama3.2-3B | 512 | 7.24e+03 | 1.02 | 1.04 | 1.05 | 0.98 |
| | | 1024 | 1.47e+04 | 1.02 | 1.00 | 0.99 | 0.99 |
| | | 2048 | 2.83e+04 | 1.06 | 1.06 | 1.07 | 1.06 |
| | | 4096 | 3.32e+04 | 1.44 | 1.26 | 1.18 | 1.10 |
| | | 8192 | 3.24e+04 | 1.44 | 1.25 | 1.18 | 1.12 |

### Prefill (FP8) (cont.)

| GPU | Model | Batchsize M | cuBLASLt Throughput | Speedup Ratio under Different Sparsity | | | |
|-----|-------|-------------|---------------------|------|------|------|------|
| | | | | 2:4 | 4:6 | 6:8 | 8:10 |
| | | 16384 | 3.27e+04 | 1.43 | 1.24 | 1.16 | 1.11 |
| | | 32768 | 3.27e+04 | 1.43 | 1.24 | 1.16 | 1.11 |
| | Qwen2.5-7B | 512 | 8.30e+03 | 1.01 | 1.01 | 1.00 | 1.00 |
| | | 1024 | 1.51e+04 | 1.10 | 1.11 | 1.10 | 1.07 |
| | | 2048 | 1.59e+04 | 1.48 | 1.23 | 1.15 | 1.11 |
| | | 4096 | 1.65e+04 | 1.50 | 1.25 | 1.17 | 1.12 |
| | | 8192 | 1.63e+04 | 1.52 | 1.27 | 1.19 | 1.14 |
| | | 16384 | 1.60e+04 | 1.55 | 1.30 | 0.77 | 0.61 |
| | | 32768 | 1.62e+04 | 0.51 | 0.38 | 0.34 | 0.32 |
| | Qwen2.5-14B | 512 | 4.70e+03 | 1.00 | 1.03 | 1.04 | |
| | | 1024 | 8.44e+03 | 1.17 | 1.18 | 1.13 | |
| | | 2048 | 8.57e+03 | 1.55 | 1.29 | 1.20 | |
| | | 4096 | 8.48e+03 | 1.58 | 1.31 | 1.22 | |
| | | 8192 | 8.45e+03 | 1.57 | 1.29 | 1.22 | |
| | | 16384 | 8.42e+03 | 1.57 | 1.30 | | |
| | | 32768 | | | | | |
| H100 | Llama3.2-1B | 512 | 2.02e+04 | 0.81 | 0.83 | 0.77 | 0.81 |
| | | 1024 | 3.92e+04 | 0.83 | 0.83 | 0.80 | 0.82 |
| | | 2048 | 7.36e+04 | 0.84 | 0.88 | 0.84 | 0.84 |
| | | 4096 | 1.35e+05 | 0.84 | 0.87 | 0.90 | 0.84 |
| | | 8192 | 1.70e+05 | 1.00 | 0.96 | 0.90 | 0.89 |
| | | 16384 | 1.90e+05 | 1.13 | 0.97 | 0.95 | 0.90 |
| | | 32768 | 1.97e+05 | 1.14 | 0.98 | 0.97 | 0.90 |
| | BitNet-2B | 512 | 1.56e+04 | 1.03 | 1.06 | 1.04 | 1.07 |
| | | 1024 | 3.18e+04 | 1.03 | 1.04 | 1.03 | 1.02 |
| | | 2048 | 6.31e+04 | 0.99 | 1.03 | 1.01 | 1.08 |
| | | 4096 | 8.74e+04 | 1.10 | 0.96 | 0.94 | 0.92 |
| | | 8192 | 9.58e+04 | 1.12 | 0.99 | 0.96 | 0.92 |
| | | 16384 | 9.95e+04 | 1.14 | 0.98 | 0.96 | 0.94 |
| | | 32768 | 1.03e+05 | 1.15 | 0.99 | 0.96 | 0.93 |
| | Llama3.2-3B | 512 | 1.75e+04 | 0.99 | 0.97 | 0.95 | 0.99 |
| | | 1024 | 3.49e+04 | 1.02 | 1.02 | 1.00 | 1.01 |
| | | 2048 | 6.46e+04 | 1.04 | 1.03 | 0.97 | 0.92 |
| | | 4096 | 7.48e+04 | 1.13 | 1.01 | 0.95 | 0.87 |
| | | 8192 | 7.85e+04 | 1.17 | 1.01 | 0.96 | 0.91 |
| | | 16384 | 7.95e+04 | 1.20 | 1.04 | 0.98 | 0.92 |
| | | 32768 | 8.08e+04 | 1.20 | 1.04 | 0.97 | 0.92 |
| | Qwen2.5-7B | 512 | 1.84e+04 | 1.02 | 0.98 | 1.01 | 1.03 |
| | | 1024 | 3.38e+04 | 1.12 | 0.99 | 0.94 | 0.92 |
| | | 2048 | 3.64e+04 | 1.21 | 1.05 | 1.03 | 0.96 |
| | | 4096 | 3.91e+04 | 1.24 | 1.05 | 0.99 | 0.93 |
| | | 8192 | 3.96e+04 | 1.27 | 1.06 | 0.99 | 0.94 |
| | | 16384 | 3.98e+04 | 1.30 | 1.08 | 1.02 | 0.97 |
| | | 32768 | 4.07e+04 | 1.26 | 1.04 | 0.98 | 0.94 |
| | Qwen2.5-14B | 512 | 1.15e+04 | 1.05 | 0.99 | 1.06 | 0.99 |
| | | 1024 | 1.85e+04 | 1.18 | 1.00 | 0.94 | 0.92 |
| | | 2048 | 1.97e+04 | 1.24 | 1.06 | 0.98 | 0.96 |
| | | 4096 | 2.03e+04 | 1.30 | 1.07 | 1.00 | 0.97 |
| | | 8192 | 2.08e+04 | 1.30 | 1.08 | 1.01 | 0.96 |
| | | 16384 | 2.09e+04 | 1.31 | 1.07 | 1.00 | 0.96 |
| | | 32768 | 2.11e+04 | 1.30 | 1.07 | 1.00 | 0.95 |
| B200 | Llama3.2-1B | 512 | 2.47e+04 | 0.90 | 0.89 | 0.85 | 0.88 |
| | | 1024 | 4.83e+04 | 0.87 | 0.88 | 0.87 | 0.88 |
| | | 2048 | 8.66e+04 | 1.01 | 0.92 | 0.98 | 0.98 |
| | | 4096 | 1.83e+05 | 0.90 | 0.91 | 0.89 | 0.92 |
| | | 8192 | 3.42e+05 | 0.90 | 0.92 | 0.92 | 0.87 |
| | | 16384 | 4.68e+05 | 0.99 | 0.98 | 1.00 | 0.98 |
| | | 32768 | 4.72e+05 | 1.01 | 1.00 | 1.00 | 1.01 |
| | BitNet-2B | 512 | 1.67e+04 | 1.01 | 1.00 | 1.01 | 1.01 |
| | | 1024 | 3.40e+04 | 0.99 | 0.99 | 0.98 | 0.98 |
| | | 2048 | 6.68e+04 | 1.04 | 1.05 | 1.01 | 0.98 |
| | | 4096 | 1.31e+05 | 1.03 | 1.03 | 1.02 | 1.03 |
| | | 8192 | 2.42e+05 | 1.04 | 1.00 | 0.97 | 0.98 |
| | | 16384 | 2.92e+05 | 1.10 | 0.95 | 0.93 | 0.91 |
| | | 32768 | 2.97e+05 | 1.13 | 0.98 | 0.96 | 0.93 |
| | Llama3.2-3B | 512 | 1.78e+04 | 1.01 | 0.99 | 0.98 | 0.97 |
| | | 1024 | 3.62e+04 | 1.02 | 0.96 | 0.98 | 1.02 |
| | | 2048 | 6.82e+04 | 1.01 | 1.01 | 1.01 | 1.02 |
| | | 4096 | 1.37e+05 | 1.01 | 1.02 | 1.03 | 1.03 |
| | | 8192 | 2.25e+05 | 1.10 | 0.98 | 0.95 | 0.88 |
| | | 16384 | 2.38e+05 | 1.15 | 1.02 | 0.97 | 0.91 |
| | | 32768 | 2.44e+05 | 1.20 | 1.04 | 0.99 | 0.92 |
| | Qwen2.5-7B | 512 | 1.85e+04 | 1.06 | 1.08 | 1.05 | 1.05 |
| | | 1024 | 3.86e+04 | 1.06 | 1.06 | 1.06 | 1.05 |
| | | 2048 | 7.12e+04 | 1.05 | 1.08 | 1.07 | 1.10 |
| | | 4096 | 1.17e+05 | 1.19 | 1.01 | 0.98 | 0.93 |
| | | 8192 | 1.23e+05 | 1.23 | 1.02 | 1.00 | 0.96 |
| | | 16384 | 1.29e+05 | 1.25 | 1.03 | 0.99 | 0.94 |

### Prefill (FP8) (cont.)

| GPU | Model | Batchsize M | cuBLASLt Throughput | Speedup Ratio under Different Sparsity | | | |
|---|---|---|---|---|---|---|---|
| | | | | 2:4 | 4:6 | 6:8 | 8:10 |
| | | 32768 | 1.31e+05 | 1.26 | 1.04 | 0.99 | 0.95 |
| | Qwen2.5-14B | 512 | 1.12e+04 | 1.06 | 1.02 | 1.05 | 1.05 |
| | | 1024 | 2.31e+04 | 1.04 | 1.02 | 1.05 | 1.04 |
| | | 2048 | 4.63e+04 | 1.07 | 1.00 | 1.01 | 1.05 |
| | | 4096 | 6.26e+04 | 1.23 | 1.02 | 0.99 | 0.93 |
| | | 8192 | 6.59e+04 | 1.21 | 1.05 | 1.00 | 0.96 |
| | | 16384 | 6.78e+04 | 1.27 | 0.93 | 0.99 | 0.95 |
| | | 32768 | 6.89e+04 | 1.28 | 1.03 | 1.00 | 0.97 |
| RTX5080 | Llama3.2-1B | 512 | 3.77e+04 | 0.97 | 0.99 | 0.93 | 1.00 |
| | | 1024 | 6.82e+04 | 1.03 | 1.03 | 1.00 | 0.95 |
| | | 2048 | 7.10e+04 | 1.33 | 1.14 | 1.09 | 1.04 |
| | | 4096 | 7.26e+04 | 1.33 | 1.15 | 1.08 | 1.04 |
| | | 8192 | 7.23e+04 | 1.33 | 1.14 | 1.07 | 1.04 |
| | | 16384 | 7.14e+04 | 1.32 | 1.14 | 1.07 | 1.03 |
| | | 32768 | 7.10e+04 | 1.32 | 1.14 | 1.07 | 1.03 |
| | BitNet-2B | 512 | 2.62e+04 | 1.10 | 1.10 | 1.11 | 1.08 |
| | | 1024 | 3.62e+04 | 1.27 | 1.08 | 1.02 | 0.95 |
| | | 2048 | 3.67e+04 | 1.35 | 1.17 | 1.10 | 1.04 |
| | | 4096 | 3.67e+04 | 1.38 | 1.17 | 1.10 | 1.06 |
| | | 8192 | 3.53e+04 | 1.36 | 1.16 | 1.09 | 1.05 |
| | | 16384 | 3.51e+04 | 1.35 | 1.14 | 1.08 | 1.04 |
| | | 32768 | 3.50e+04 | 1.34 | 1.15 | 1.08 | 1.04 |
| | Llama3.2-3B | 512 | 2.47e+04 | 1.19 | 1.12 | 1.05 | 1.01 |
| | | 1024 | 2.86e+04 | 1.31 | 1.13 | 1.05 | 0.99 |
| | | 2048 | 2.84e+04 | 1.47 | 1.23 | 1.13 | 1.08 |
| | | 4096 | 2.85e+04 | 1.41 | 1.20 | 1.10 | 1.05 |
| | | 8192 | 2.80e+04 | 1.40 | 1.19 | 1.10 | 1.04 |
| | | 16384 | 2.79e+04 | 1.39 | 1.18 | 1.10 | 1.04 |
| | | 32768 | 2.78e+04 | 1.39 | 1.18 | 1.10 | 1.04 |
| | Qwen2.5-7B | 512 | 1.30e+04 | 1.42 | 1.18 | 1.10 | 1.04 |
| | | 1024 | 1.35e+04 | 1.48 | 1.19 | 1.12 | 1.06 |
| | | 2048 | 1.34e+04 | 1.52 | 1.24 | 1.14 | 1.08 |
| | | 4096 | 1.36e+04 | 1.47 | 1.23 | 1.12 | 1.06 |
| | | 8192 | 1.35e+04 | 1.48 | 1.22 | 1.12 | 1.06 |
| | | 16384 | 1.35e+04 | 1.47 | 1.22 | 1.12 | 1.06 |
| | | 32768 | | | | | |
| | Qwen2.5-14B | 512 | | | | | |
| | | 1024 | | | | | |
| | | 2048 | | | | | |
| | | 4096 | | | | | |
| | | 8192 | | | | | |
| | | 16384 | | | | | |
| | | 32768 | | | | | |
| GB10 | Llama3.2-1B | 512 | 2.70e+04 | 1.10 | 0.98 | 0.92 | 0.92 |
| | | 1024 | 3.07e+04 | 1.04 | 0.93 | 0.90 | 0.86 |
| | | 2048 | 3.23e+04 | 1.06 | 0.94 | 0.88 | 0.86 |
| | | 4096 | 3.19e+04 | 1.06 | 0.93 | 0.91 | 0.85 |
| | | 8192 | 3.13e+04 | 1.05 | 0.94 | 0.91 | 0.87 |
| | | 16384 | 3.13e+04 | 1.06 | 0.97 | 0.93 | 0.89 |
| | | 32768 | 3.19e+04 | 1.06 | 0.97 | 0.92 | 0.89 |
| | BitNet-2B | 512 | 1.40e+04 | 1.08 | 1.02 | 1.01 | 0.95 |
| | | 1024 | 1.61e+04 | 1.07 | 0.96 | 0.93 | 0.91 |
| | | 2048 | 1.71e+04 | 1.07 | 0.93 | 0.89 | 0.86 |
| | | 4096 | 1.66e+04 | 1.03 | 0.90 | 0.87 | 0.85 |
| | | 8192 | 1.61e+04 | 1.07 | 0.93 | 0.90 | 0.87 |
| | | 16384 | 1.63e+04 | 1.08 | 0.94 | 0.91 | 0.87 |
| | | 32768 | 1.64e+04 | 1.05 | 0.93 | 0.88 | 0.88 |
| | Llama3.2-3B | 512 | 1.20e+04 | 1.16 | 1.01 | 1.00 | 0.96 |
| | | 1024 | 1.39e+04 | 1.10 | 0.96 | 0.91 | 0.86 |
| | | 2048 | 1.39e+04 | 1.08 | 0.97 | 0.91 | 0.86 |
| | | 4096 | 1.36e+04 | 1.06 | 0.95 | 0.89 | 0.85 |
| | | 8192 | 1.35e+04 | 1.08 | 0.96 | 0.88 | 0.87 |
| | | 16384 | 1.36e+04 | 1.08 | 0.96 | 0.90 | 0.87 |
| | | 32768 | 1.35e+04 | 1.08 | 0.96 | 0.87 | 0.88 |
| | Qwen2.5-7B | 512 | 5.83e+03 | 1.09 | 1.02 | 0.97 | 0.90 |
| | | 1024 | 6.78e+03 | 1.12 | 0.91 | 0.88 | 0.85 |
| | | 2048 | 6.91e+03 | 1.10 | 0.90 | 0.86 | 0.83 |
| | | 4096 | 7.06e+03 | 1.04 | 0.86 | 0.83 | 0.79 |
| | | 8192 | 6.79e+03 | 1.10 | 0.91 | 0.88 | 0.85 |
| | | 16384 | 6.80e+03 | 1.10 | 0.92 | 0.88 | 0.85 |
| | | 32768 | 6.81e+03 | 1.12 | 0.93 | 0.88 | 0.85 |
| | Qwen2.5-14B | 512 | 3.17e+03 | 1.29 | 1.07 | 1.03 | 0.97 |
| | | 1024 | 3.75e+03 | 1.15 | 0.92 | 0.86 | 0.83 |
| | | 2048 | 3.66e+03 | 1.14 | 0.92 | 0.88 | 0.84 |
| | | 4096 | 3.73e+03 | 1.15 | 0.94 | 0.87 | 0.83 |
| | | 8192 | 3.71e+03 | 1.13 | 0.94 | 0.88 | 0.84 |
| | | 16384 | 1.10e+03 | 1.03 | 3.18 | 2.98 | 2.79 |
| | | 32768 | 3.70e+03 | 1.12 | 0.93 | 0.85 | 0.82 |

**INT8 Prefill Highlights.** A100 demonstrates the strongest INT8 prefill performance: Qwen-2.5-14B at $M \geq 2048$ achieves 1.73–1.77× with 2:4 sparsity; 6:8 achieves 1.29–1.34×, approaching the 1.33× theoretical bound. Qwen-2.5-7B follows closely at 1.69–1.75× (2:4). Smaller models (Llama-3.2-1B) show lower speedups (1.37–1.50×) due to smaller GEMM dimensions and relatively higher framework overhead.

H100 INT8 prefill achieves 1.34–1.45× for 2:4 on Qwen-14B, lower than A100 due to better-optimized dense baseline. B200 shows model-size dependence: Llama-1B achieves ∼1.0× (overhead matches savings), while larger models reach 1.27–1.41×. RTX 5080 demonstrates excellent consumer GPU performance: 1.31–1.40× for most models with 2:4.

**FP8 Prefill Results.** FP8 generally shows 5–15% lower speedups than INT8 across all GPUs. H100 FP8: Qwen-14B achieves 1.24–1.31× (vs. 1.34–1.45× INT8). B200 FP8: reaches 1.23–1.28× for large models. RTX 4090 FP8 prefill achieves 1.18–1.19× at 6:8, demonstrating that SlideSparse benefits both datacenter and consumer GPUs.

D.5.2. DECODE STAGE RESULTS

Decode is the autoregressive token generation phase and is **memory-bound** due to KV cache access. We configure $M = \texttt{max\_num\_seqs}$ with $M \in \{64, 128, 256, 512\}$. The following tables present complete decode throughput speedup results.

## Decode (INT8)

| GPU | Model | Concurrency M | cuBLASLt Throughput | Speedup Ratio under Different Sparsity | | | |
|---|---|---|---|---|---|---|---|
| | | | | 2:4 | 4:6 | 6:8 | 8:10 |
| A100 | Llama3.2-1B | 64 | 1.12e+04 | 1.10 | 1.00 | 0.91 | 0.96 |
| | | 128 | 1.58e+04 | 1.05 | 0.98 | 1.01 | 0.97 |
| | | 256 | 2.30e+04 | 1.07 | 1.03 | 1.01 | 0.99 |
| | | 512 | 2.29e+04 | 1.09 | 1.06 | 1.06 | 1.02 |
| | BitNet-2B | 64 | 7.09e+03 | 1.16 | 1.11 | 1.10 | 1.07 |
| | | 128 | 1.09e+04 | 1.14 | 1.07 | 1.05 | 1.03 |
| | | 256 | 1.60e+04 | 1.14 | 1.06 | 1.05 | 1.03 |
| | | 512 | 1.59e+04 | 1.17 | 1.09 | 1.08 | 1.07 |
| | Llama3.2-3B | 64 | 6.69e+03 | 1.09 | 1.04 | 1.01 | 0.99 |
| | | 128 | 1.01e+04 | 1.16 | 1.08 | 1.06 | 1.05 |
| | | 256 | 1.42e+04 | 1.13 | 1.06 | 1.05 | 1.03 |
| | | 512 | 1.42e+04 | 1.18 | 1.10 | 1.07 | 1.04 |
| | Qwen2.5-7B | 64 | 5.00e+03 | 1.25 | 1.14 | 1.12 | 1.08 |
| | | 128 | 8.17e+03 | 1.21 | 1.09 | 1.05 | 1.03 |
| | | 256 | 1.03e+04 | 1.26 | 1.12 | 1.08 | 1.04 |
| | | 512 | 9.82e+03 | 1.31 | 1.16 | 1.12 | 1.09 |
| | Qwen2.5-14B | 64 | 3.08e+03 | 1.28 | 1.14 | 1.10 | 1.08 |
| | | 128 | 4.84e+03 | 1.24 | 1.12 | 1.09 | 1.04 |
| | | 256 | 5.77e+03 | 1.35 | 1.19 | 1.13 | 1.10 |
| | | 512 | 5.39e+03 | 1.40 | 1.23 | 1.17 | 1.14 |
| RTX4090 | Llama3.2-1B | 64 | 9.37e+03 | 1.15 | 1.06 | 0.97 | 0.94 |
| | | 128 | 1.40e+04 | 1.03 | 1.01 | 1.00 | 0.96 |
| | | 256 | 2.02e+04 | 1.08 | 0.97 | 1.03 | 1.01 |
| | | 512 | 2.01e+04 | 1.08 | 1.04 | 1.02 | 1.00 |
| | BitNet-2B | 64 | 6.99e+03 | 1.05 | 0.98 | 0.96 | 0.95 |
| | | 128 | 1.06e+04 | 1.17 | 1.05 | 1.10 | 1.07 |
| | | 256 | 1.46e+04 | 1.19 | 1.13 | 1.13 | 1.06 |
| | | 512 | 1.32e+04 | 1.34 | 1.26 | 1.27 | 1.18 |
| | Llama3.2-3B | 64 | 5.68e+03 | 0.98 | 0.97 | 0.96 | 0.92 |
| | | 128 | 8.19e+03 | 1.20 | 1.15 | 1.12 | 1.10 |
| | | 256 | 1.15e+04 | 1.16 | 1.10 | 1.09 | 1.07 |
| | | 512 | 1.16e+04 | 1.17 | 1.09 | 1.05 | 1.03 |
| | Qwen2.5-7B | 64 | 3.99e+03 | 1.22 | 1.12 | 0.94 | 0.90 |
| | | 128 | 6.33e+03 | 1.32 | 1.18 | 1.04 | 1.01 |
| | | 256 | 8.69e+03 | 1.23 | 1.13 | 1.01 | 1.00 |
| | | 512 | 8.70e+03 | 1.26 | 1.14 | 1.06 | 1.04 |
| | Qwen2.5-14B | 64 | 1.53e+03 | 1.94 | 1.14 | 0.64 | |
| | | 128 | 2.38e+03 | 1.75 | 1.14 | 0.79 | 0.40 |
| | | 256 | 2.43e+03 | 1.67 | 1.12 | 0.58 | |
| | | 512 | 1.37e+03 | 3.04 | 1.23 | | |
| H100 | Llama3.2-1B | 64 | 1.31e+04 | 1.11 | 1.09 | 0.98 | 1.04 |
| | | 128 | 1.83e+04 | 1.05 | 1.11 | 1.06 | 1.11 |
| | | 256 | 2.91e+04 | 0.92 | 1.02 | 1.01 | 0.98 |
| | | 512 | 2.99e+04 | 1.09 | 1.05 | 1.05 | 1.03 |
| | BitNet-2B | 64 | 8.11e+03 | 1.10 | 1.10 | 1.08 | 1.05 |
| | | 128 | 1.32e+04 | 1.16 | 1.11 | 1.09 | 1.08 |
| | | 256 | 2.03e+04 | 1.10 | 1.03 | 1.03 | 1.00 |
| | | 512 | 2.07e+04 | 1.13 | 1.06 | 1.08 | 1.04 |
| | Llama3.2-3B | 64 | 7.34e+03 | 1.11 | 1.12 | 1.11 | 1.07 |
| | | 128 | 1.14e+04 | 1.23 | 1.20 | 1.16 | 1.16 |
| | | 256 | 1.80e+04 | 1.02 | 1.03 | 0.99 | 0.95 |
| | | 512 | 1.81e+04 | 1.11 | 1.09 | 1.07 | 1.04 |
| | Qwen2.5-7B | 64 | 5.72e+03 | 1.24 | 1.15 | 1.15 | 1.10 |
| | | 128 | 9.93e+03 | 1.24 | 1.13 | 1.12 | 1.09 |
| | | 256 | 1.46e+04 | 1.12 | 1.04 | 1.01 | 0.98 |
| | | 512 | 1.47e+04 | 1.19 | 1.05 | 1.07 | 1.03 |
| | Qwen2.5-14B | 64 | 3.31e+03 | 1.49 | 1.27 | 1.27 | 1.24 |
| | | 128 | 5.67e+03 | 1.32 | 1.19 | 1.16 | 1.13 |
| | | 256 | 8.29e+03 | 1.21 | 1.07 | 1.04 | 0.96 |
| | | 512 | 8.08e+03 | 1.28 | 1.14 | 1.10 | 1.06 |
| B200 | Llama3.2-1B | 64 | 1.71e+04 | 1.17 | 1.18 | 1.16 | 1.14 |
| | | 128 | 2.54e+04 | 1.13 | 1.11 | 1.10 | 1.10 |
| | | 256 | 4.00e+04 | 1.13 | 0.95 | 0.98 | 1.07 |
| | | 512 | 4.36e+04 | 1.12 | 1.09 | 1.09 | 1.08 |
| | BitNet-2B | 64 | 9.97e+03 | 1.05 | 1.10 | 1.11 | 1.10 |
| | | 128 | 1.72e+04 | 1.24 | 1.19 | 1.19 | 1.19 |
| | | 256 | 2.65e+04 | 1.26 | 1.11 | 1.12 | 1.12 |
| | | 512 | 3.11e+04 | 1.19 | 1.13 | 1.13 | 1.13 |
| | Llama3.2-3B | 64 | 1.11e+04 | 1.14 | 1.11 | 1.09 | 1.07 |
| | | 128 | 1.72e+04 | 1.24 | 1.19 | 1.18 | 1.17 |
| | | 256 | 2.59e+04 | 1.15 | 1.12 | 1.12 | 1.07 |
| | | 512 | 3.04e+04 | 1.17 | 1.14 | 1.13 | 1.10 |
| | Qwen2.5-7B | 64 | 9.57e+03 | 1.28 | 1.25 | 1.23 | 1.21 |
| | | 128 | 1.54e+04 | 1.30 | 1.21 | 1.21 | 1.18 |
| | | 256 | 2.47e+04 | 1.17 | 1.10 | 1.08 | 1.06 |
| | | 512 | 2.65e+04 | 1.24 | 1.15 | 1.14 | 1.12 |
| | Qwen2.5-14B | 64 | 5.67e+03 | 1.36 | 1.38 | 1.35 | 1.31 |
| | | 128 | 1.00e+04 | 1.31 | 1.22 | 1.21 | 1.19 |

## Decode (INT8) (cont.)

| GPU | Model | Concurrency M | cuBLASLt Throughput | Speedup Ratio under Different Sparsity | | | |
|---|---|---|---|---|---|---|---|
| | | | | 2:4 | 4:6 | 6:8 | 8:10 |
| RTX5080 | | 256 | 1.66e+04 | 1.13 | 1.08 | 1.10 | 1.04 |
| | | 512 | 1.74e+04 | 1.25 | 1.15 | 1.12 | 1.11 |
| | Llama3.2-1B | 64 | 1.40e+04 | 1.18 | 1.11 | 1.07 | 1.05 |
| | | 128 | 2.04e+04 | 1.13 | 1.07 | 1.06 | 1.04 |
| | | 256 | 2.23e+04 | 1.15 | 1.09 | 1.03 | 1.05 |
| | | 512 | 2.25e+04 | 1.11 | 1.06 | 1.04 | 1.01 |
| | BitNet-2B | 64 | 8.37e+03 | 1.11 | 1.09 | 1.04 | 1.02 |
| | | 128 | 1.33e+04 | 1.18 | 1.09 | 1.07 | 1.04 |
| | | 256 | 1.55e+04 | 1.16 | 1.08 | 1.03 | 1.06 |
| | | 512 | 1.42e+04 | 1.15 | 1.05 | 1.03 | 1.02 |
| | Llama3.2-3B | 64 | 6.95e+03 | 1.30 | 1.18 | 1.18 | 1.14 |
| | | 128 | 1.07e+04 | 1.27 | 1.18 | 1.14 | 1.11 |
| | | 256 | 1.24e+04 | 1.21 | 1.11 | 1.07 | 1.03 |
| | | 512 | 1.07e+04 | 1.19 | 1.10 | 1.06 | 1.04 |
| | Qwen2.5-7B | 64 | 4.49e+03 | 1.38 | 1.24 | 1.17 | 1.09 |
| | | 128 | 7.50e+03 | 1.43 | 1.24 | 1.17 | 1.10 |
| | | 256 | 6.75e+03 | 1.80 | 1.25 | 1.02 | 0.93 |
| | | 512 | 6.33e+03 | 1.52 | 1.19 | 0.77 | |
| | Qwen2.5-14B | 64 | | | | | |
| | | 128 | | | | | |
| | | 256 | | | | | |
| | | 512 | | | | | |
| GB10 | Llama3.2-1B | 64 | 4.22e+03 | 1.16 | 1.00 | 1.00 | 0.95 |
| | | 128 | 6.23e+03 | 1.18 | 1.04 | 1.05 | 1.01 |
| | | 256 | 7.86e+03 | 1.11 | 1.00 | 1.03 | 0.97 |
| | | 512 | 7.99e+03 | 1.13 | 1.04 | 1.05 | 0.99 |
| | BitNet-2B | 64 | 2.46e+03 | 1.22 | 0.99 | 1.07 | 0.98 |
| | | 128 | 3.71e+03 | 1.18 | 1.05 | 1.06 | 1.03 |
| | | 256 | 4.81e+03 | 1.13 | 1.03 | 1.03 | 1.02 |
| | | 512 | 4.77e+03 | 1.17 | 1.07 | 1.07 | 1.06 |
| | Llama3.2-3B | 64 | 2.83e+02 | 8.56 | 7.82 | 7.43 | 6.58 |
| | | 128 | 3.07e+03 | 1.21 | 1.09 | 1.07 | 0.96 |
| | | 256 | 3.97e+03 | 1.12 | 1.05 | 1.03 | 0.96 |
| | | 512 | 4.01e+03 | 1.15 | 1.07 | 1.04 | 0.98 |
| | Qwen2.5-7B | 64 | 1.14e+03 | 1.36 | 1.08 | 1.15 | 1.08 |
| | | 128 | 1.94e+03 | 1.38 | 1.05 | 1.11 | 1.08 |
| | | 256 | 2.55e+03 | 1.36 | 1.06 | 1.10 | 1.09 |
| | | 512 | 2.71e+03 | 1.28 | 1.05 | 1.07 | 0.99 |
| | Qwen2.5-14B | 64 | 6.17e+02 | 1.41 | 1.05 | 1.07 | 1.02 |
| | | 128 | 1.02e+03 | 1.40 | 1.06 | 1.09 | 1.04 |
| | | 256 | 1.38e+03 | 1.32 | 1.06 | 1.07 | 1.01 |
| | | 512 | 1.44e+03 | 1.28 | 1.03 | 1.05 | 1.02 |

## Decode (FP8)

| GPU | Model | Concurrency M | cuBLASLt Throughput | Speedup Ratio under Different Sparsity | | | |
|---|---|---|---|---|---|---|---|
| | | | | 2:4 | 4:6 | 6:8 | 8:10 |
| RTX4090 | Llama3.2-1B | 64 | 9.78e+03 | 1.05 | 0.86 | 0.87 | 0.86 |
| | | 128 | 1.37e+04 | 1.00 | 0.96 | 0.97 | 0.96 |
| | | 256 | 1.88e+04 | 1.13 | 1.01 | 1.06 | 1.03 |
| | | 512 | 1.97e+04 | 1.04 | 0.99 | 0.98 | 0.97 |
| | BitNet-2B | 64 | 7.54e+03 | 0.90 | 0.86 | 0.83 | 0.80 |
| | | 128 | 9.96e+03 | 1.12 | 1.04 | 1.07 | 1.07 |
| | | 256 | 1.54e+04 | 1.09 | 0.98 | 0.97 | 0.96 |
| | | 512 | 1.52e+04 | 1.07 | 1.01 | 1.00 | 0.95 |
| | Llama3.2-3B | 64 | 5.86e+03 | 0.98 | 0.89 | 0.86 | 0.82 |
| | | 128 | 8.73e+03 | 1.06 | 0.99 | 0.96 | 0.96 |
| | | 256 | 1.15e+04 | 1.13 | 1.05 | 1.02 | 0.96 |
| | | 512 | 1.10e+04 | 1.17 | 1.07 | 1.02 | 1.00 |
| | Qwen2.5-7B | 64 | 3.87e+03 | 1.15 | 0.99 | 0.93 | 0.98 |
| | | 128 | 6.34e+03 | 1.20 | 1.05 | 0.97 | 0.96 |
| | | 256 | 8.20e+03 | 1.20 | 1.05 | 1.01 | 0.99 |
| | | 512 | 7.83e+03 | 1.25 | 1.13 | 1.07 | 1.02 |
| | Qwen2.5-14B | 64 | 1.56e+03 | 1.84 | 1.08 | 0.58 | |
| | | 128 | 2.28e+03 | 1.88 | 1.11 | 0.77 | 0.39 |
| | | 256 | 2.26e+03 | 1.69 | 1.11 | 0.56 | |
| | | 512 | 1.32e+03 | 2.83 | 1.17 | | |
| H100 | Llama3.2-1B | 64 | 1.49e+04 | 1.00 | 0.96 | 0.84 | 0.94 |
| | | 128 | 2.12e+04 | 0.96 | 0.95 | 0.95 | 0.95 |
| | | 256 | 3.14e+04 | 0.96 | 0.94 | 0.95 | 0.90 |
| | | 512 | 3.19e+04 | 0.99 | 0.99 | 0.98 | 0.97 |
| | BitNet-2B | 64 | 9.65e+03 | 0.98 | 0.95 | 0.92 | 0.90 |
| | | 128 | 1.44e+04 | 1.04 | 1.00 | 0.99 | 0.99 |
| | | 256 | 2.18e+04 | 1.00 | 0.96 | 0.96 | 0.94 |
| | | 512 | 2.28e+04 | 1.02 | 0.98 | 0.97 | 0.94 |
| | Llama3.2-3B | 64 | 8.64e+03 | 1.00 | 0.93 | 0.93 | 0.90 |
| | | 128 | 1.35e+04 | 1.03 | 1.01 | 0.99 | 0.96 |
| | | 256 | 1.95e+04 | 1.00 | 0.92 | 0.93 | 0.90 |

### Decode (FP8) (cont.)

| GPU | Model | Concurrency M | cuBLASLt Throughput | Speedup Ratio under Different Sparsity | | | |
|---|---|---|---|---|---|---|---|
| | | | | 2:4 | 4:6 | 6:8 | 8:10 |
| | | 512 | 2.03e+04 | 1.01 | 0.94 | 0.95 | 0.90 |
| | Qwen2.5-7B | 64 | 7.00e+03 | 1.09 | 0.98 | 0.94 | 0.91 |
| | | 128 | 1.12e+04 | 1.10 | 0.99 | 1.00 | 0.98 |
| | | 256 | 1.50e+04 | 1.13 | 1.01 | 1.00 | 0.95 |
| | | 512 | 1.61e+04 | 1.09 | 0.98 | 0.96 | 0.92 |
| | Qwen2.5-14B | 64 | 3.78e+03 | 1.30 | 1.12 | 1.11 | 1.08 |
| | | 128 | 6.22e+03 | 1.21 | 1.03 | 1.06 | 1.01 |
| | | 256 | 8.89e+03 | 1.13 | 1.01 | 0.96 | 0.91 |
| | | 512 | 8.66e+03 | 1.18 | 1.04 | 1.02 | 0.98 |
| B200 | Llama3.2-1B | 64 | 2.05e+04 | 1.00 | 0.98 | 0.86 | 0.97 |
| | | 128 | 2.93e+04 | 1.00 | 0.98 | 0.96 | 0.97 |
| | | 256 | 4.49e+04 | 1.01 | 0.86 | 0.87 | 0.83 |
| | | 512 | 4.88e+04 | 1.00 | 0.97 | 0.98 | 0.97 |
| | BitNet-2B | 64 | 1.34e+04 | 0.88 | 0.85 | 0.84 | 0.84 |
| | | 128 | 2.04e+04 | 1.03 | 0.99 | 0.99 | 0.98 |
| | | 256 | 3.06e+04 | 0.99 | 0.97 | 0.95 | 0.95 |
| | | 512 | 3.59e+04 | 1.02 | 0.98 | 0.98 | 0.97 |
| | Llama3.2-3B | 64 | 1.40e+04 | 0.94 | 0.91 | 0.90 | 0.89 |
| | | 128 | 2.08e+04 | 1.02 | 0.98 | 0.98 | 0.97 |
| | | 256 | 3.28e+04 | 1.00 | 0.97 | 0.93 | 0.95 |
| | | 512 | 3.48e+04 | 1.03 | 1.00 | 0.98 | 0.97 |
| | Qwen2.5-7B | 64 | 1.19e+04 | 1.11 | 1.02 | 1.00 | 0.99 |
| | | 128 | 1.88e+04 | 1.06 | 1.00 | 0.98 | 0.88 |
| | | 256 | 2.76e+04 | 1.05 | 0.97 | 0.99 | 0.93 |
| | | 512 | 3.08e+04 | 1.07 | 1.00 | 0.98 | 0.97 |
| | Qwen2.5-14B | 64 | 7.35e+03 | 1.16 | 1.06 | 1.04 | 1.02 |
| | | 128 | 1.22e+04 | 1.08 | 1.01 | 0.99 | 0.99 |
| | | 256 | 1.81e+04 | 1.09 | 0.98 | 0.94 | 0.95 |
| | | 512 | 1.89e+04 | 1.15 | 1.06 | 1.03 | 1.03 |
| RTX5080 | Llama3.2-1B | 64 | 1.46e+04 | 1.11 | 1.02 | 0.99 | 0.94 |
| | | 128 | 2.08e+04 | 1.05 | 1.00 | 0.98 | 0.97 |
| | | 256 | 2.14e+04 | 1.08 | 1.06 | 1.04 | 0.97 |
| | | 512 | 2.14e+04 | 1.10 | 1.03 | 1.01 | 0.98 |
| | BitNet-2B | 64 | 9.45e+03 | 0.98 | 0.89 | 0.86 | 0.82 |
| | | 128 | 1.38e+04 | 1.05 | 0.98 | 0.94 | 0.93 |
| | | 256 | 1.46e+04 | 1.13 | 1.04 | 1.02 | 0.99 |
| | | 512 | 1.30e+04 | 1.16 | 1.05 | 1.03 | 0.99 |
| | Llama3.2-3B | 64 | 7.81e+03 | 1.16 | 0.99 | 0.97 | 0.94 |
| | | 128 | 1.14e+04 | 1.12 | 1.03 | 0.99 | 0.96 |
| | | 256 | 1.17e+04 | 1.17 | 1.08 | 1.04 | 1.00 |
| | | 512 | 9.99e+03 | 1.18 | 1.08 | 1.03 | 0.99 |
| | Qwen2.5-7B | 64 | 5.09e+03 | 1.23 | 1.05 | 0.99 | 0.94 |
| | | 128 | 7.94e+03 | 1.28 | 1.11 | 1.05 | 1.00 |
| | | 256 | 6.67e+03 | 1.61 | 1.41 | 1.13 | 0.99 |
| | | 512 | 5.89e+03 | 1.43 | 1.18 | 1.08 | 0.99 |
| | Qwen2.5-14B | 64 | | | | | |
| | | 128 | | | | | |
| | | 256 | | | | | |
| | | 512 | | | | | |
| GB10 | Llama3.2-1B | 64 | 4.42e+03 | 1.12 | 0.97 | 0.94 | 0.92 |
| | | 128 | 6.65e+03 | 1.05 | 0.99 | 0.96 | 0.92 |
| | | 256 | 8.20e+03 | 1.05 | 0.97 | 0.97 | 0.93 |
| | | 512 | 8.65e+03 | 1.02 | 0.95 | 0.95 | 0.92 |
| | BitNet-2B | 64 | 2.52e+03 | 1.15 | 0.98 | 1.00 | 0.97 |
| | | 128 | 3.93e+03 | 1.15 | 1.01 | 0.99 | 0.97 |
| | | 256 | 5.00e+03 | 1.10 | 1.00 | 1.00 | 0.97 |
| | | 512 | 5.18e+03 | 1.06 | 0.99 | 0.98 | 0.96 |
| | Llama3.2-3B | 64 | 2.11e+03 | 1.18 | 1.02 | 1.03 | 0.93 |
| | | 128 | 3.15e+03 | 1.17 | 1.06 | 1.05 | 0.97 |
| | | 256 | 4.02e+03 | 1.11 | 1.04 | 1.03 | 0.97 |
| | | 512 | 4.20e+03 | 1.08 | 1.02 | 0.99 | 0.95 |
| | Qwen2.5-7B | 64 | 1.26e+03 | 1.33 | 1.07 | 1.05 | 0.97 |
| | | 128 | 2.11e+03 | 1.29 | 1.05 | 1.04 | 1.00 |
| | | 256 | 2.91e+03 | 1.20 | 1.01 | 1.02 | 0.98 |
| | | 512 | 2.94e+03 | 1.19 | 1.02 | 1.00 | 0.96 |
| | Qwen2.5-14B | 64 | 6.49e+02 | 1.34 | 1.08 | 1.08 | 1.03 |
| | | 128 | 1.10e+03 | 1.34 | 1.07 | 1.06 | 1.00 |
| | | 256 | 1.48e+03 | 1.25 | 1.07 | 1.07 | 1.02 |
| | | 512 | 1.53e+03 | 1.22 | 1.06 | 1.01 | 1.00 |

**Decode Speedup Characteristics.** Decode speedups are inherently more modest than prefill. Sparse linear layers reduce arithmetic and part of the weight-value traffic, but the overall gain is limited by sparse metadata, KV-cache access, and other unchanged kernels. In our implementation, SlideSparse keeps the sparse path by zero-padding activations as needed for 2:4 tile alignment rather than falling back to dense execution.

A100 INT8: Qwen-14B achieves $1.24$–$1.40\times$ (2:4), $1.12$–$1.23\times$ (4:6), and up to $1.17\times$ at 6:8. B200 achieves strong decode speedups at small batch sizes: Qwen-14B shows up to $1.36\times$ for 2:4 at $M = 64$, benefiting from superior HBM3e bandwidth. H100 Qwen-14B reaches $1.21$–$1.49\times$ for 2:4. Smaller models (Llama-1B, Llama-3B) show modest speedups of $1.05$–$1.18\times$, as the decode phase is dominated by KV-cache and memory access rather than GEMM alone. FP8 decode is generally $5$–$15\%$ lower than INT8 across all configurations.

### D.5.3. END-TO-END PERFORMANCE ANALYSIS

**Kernel-to-E2E Translation.** Comparing kernel speedups (§D.3.2) with end-to-end results reveals that $80$–$95\%$ of kernel-level gains translate to actual inference speedup. The gap arises from non-GEMM components (attention, softmax, layer norm, KV cache) that remain unchanged. This high translation rate validates SlideSparse's practical effectiveness.

**Model Size Effect.** Larger models consistently achieve higher E2E speedups because: (1) GEMM constitutes a higher fraction of total inference time; (2) larger $(N, K)$ dimensions yield better sparse tensor core utilization; (3) framework overhead is better amortized. For example, A100 INT8 prefill achieves $1.37$–$1.50\times$ for Llama-1B but $1.73$–$1.77\times$ for Qwen-14B with 2:4 sparsity.

**Prefill vs. Decode Comparison.** Prefill consistently achieves higher speedups than decode due to fundamental workload characteristics. At 2:4 sparsity on A100 INT8, Qwen-14B achieves $1.73$–$1.77\times$ prefill speedup versus $1.24$–$1.40\times$ decode speedup. This $25$–$35\%$ gap reflects the memory-bound nature of autoregressive decoding, where weight loading dominates computation time.

**Combined Serving Scenario.** We also measure request-level latency on Qwen2.5-14B INT8 (A100) with a 4096-token prompt. At 6:8 sparsity, time-to-first-token decreases from 390 ms to 291 ms ($1.34\times$). End-to-end request latency improves by $1.13\times$ for 128 generated tokens and $1.11\times$ for 512 generated tokens, showing that SlideSparse remains beneficial beyond isolated prefill/decode microbenchmarks.

### D.6. Algorithmic Efficiency Analysis

While previous sections compare SlideSparse against dense cuBLASLt baselines, this section introduces **Algorithmic Efficiency**—a novel metric that measures how well SlideSparse achieves the *theoretical* speedup potential relative to NVIDIA's native 2:4 implementation. This metric isolates SlideSparse's implementation quality from variations in baseline optimization levels.

### D.6.1. MOTIVATION AND DEFINITION

Traditional speedup comparisons (sparse vs. dense) conflate two independent factors: (1) *theoretical compute reduction* from sparsity, and (2) *implementation efficiency* in realizing this reduction. When cuSPARSELt 2:4 or cuBLASLt dense implementations are suboptimal on certain configurations, SlideSparse's apparent advantage is distorted.

**Theoretical Speedup Ratio.** For sparsity pattern $Z{:}L$, define density as $\rho = (L - Z)/L$ and theoretical speedup vs. dense as $S_{\text{theory}} = 1/\rho$. The *theoretical ratio* vs. 2:4 baseline is:

$$R_{\text{theory}} = \frac{\rho(2{:}4)}{\rho(Z{:}L)} = \frac{0.5}{\rho(Z{:}L)} \tag{18}$$

| Sparsity | Density $\rho$ | $S_{\text{theory}}$ vs Dense | $R_{\text{theory}}$ vs 2:4 |
|----------|----------------|------------------------------|----------------------------|
| 2:4 | 0.500 | 2.00× | 1.000 |
| 4:6 | 0.667 | 1.50× | 0.750 |
| 6:8 | 0.750 | 1.33× | 0.667 |
| 8:10 | 0.800 | 1.25× | 0.625 |
| $\infty:\infty$ | 1.000 | 1.00× | 0.500 |

**Algorithmic Efficiency.** Given measured speedups $S_{2:4}$ and $S_{Z:L}$ (both vs. cuBLASLt dense), define:

$$\text{Efficiency} = \frac{S_{Z:L}/S_{2:4}}{R_{\text{theory}}} \times 100\% \tag{19}$$

**Interpretation:**

- Efficiency $= 100\%$: SlideSparse achieves exactly the expected speedup ratio

- Efficiency $> 100\%$: SlideSparse *outperforms* theoretical expectation relative to 2:4

- Efficiency $< 100\%$: SlideSparse achieves less than expected speedup

D.6.2. WHY CAN EFFICIENCY EXCEED 100%?

Efficiency exceeding 100% does *not* violate physical limits—it reveals *baseline inefficiencies*. When efficiency exceeds 100%, it indicates that:

1. cuSPARSELt 2:4 has higher overhead than SlideSparse at that configuration

2. The "tax" of sparse metadata handling is proportionally smaller for SlideSparse's sliding window approach

**Example: B200 INT8 at** $\text{M}=64$ **showing 200% efficiency for** $\infty:\infty$**.** This means dense (in sliding format) runs at the *same speed* as cuSPARSELt 2:4. Since theoretically dense should be $2\times$ slower (it has $2\times$ the FLOPs), achieving parity yields: $R_{\text{actual}} = 1.0$, $R_{\text{theory}} = 0.5$, Efficiency $= 200\%$. Physical explanation: at small $M$ on B200, cuSPARSELt's sparse format overhead eliminates all theoretical gains.

D.6.3. EFFICIENCY RESULTS

**Kernel-Level Efficiency.** The following tables present kernel-level algorithmic efficiency, measuring how well SlideSparse achieves the theoretical speedup potential relative to cuSPARSELt's native 2:4 implementation.

## Square Kernel (FP4)

| GPU | M | cuSPARSELt Latency(μs) | \multicolumn{7}{c}{Algorithmic Efficiency under Different Sparsity} |
|-----|---|---|---|---|---|---|---|---|---|

| GPU | M | cuSPARSELt Latency(μs) | 4:6 | 6:8 | 8:10 | 10:12 | 12:14 | 14:16 | ∞:∞ |
|-----|---|---|---|---|---|---|---|---|---|
| B200 | 64 | 6.16e+00 | 135.3% | 148.9% | 176.4% | 169.1% | 170.2% | 176.3% | 201.5% |
| | 128 | 6.20e+00 | 132.4% | 148.9% | 176.4% | 166.7% | 170.2% | 173.7% | 200.0% |
| | 256 | 6.19e+00 | 132.4% | 148.9% | 160.0% | 165.5% | 170.2% | 173.7% | 198.5% |
| | 512 | 6.24e+00 | 132.4% | 150.0% | 158.8% | 165.4% | 171.4% | 173.7% | 198.5% |
| | 1024 | 6.25e+00 | 118.5% | 135.6% | 129.2% | 132.1% | 141.0% | 142.6% | 152.6% |
| | 2048 | 1.03e+01 | 130.2% | 150.0% | 160.0% | 164.7% | 142.9% | 175.0% | 200.0% |
| | 4096 | 2.30e+01 | 107.0% | 120.4% | 104.7% | 109.1% | 124.9% | 129.6% | 130.9% |
| | 8192 | 1.15e+02 | 88.9% | 105.6% | 90.9% | 98.8% | 105.8% | 105.9% | 103.7% |
| | 16384 | 9.07e+02 | 96.0% | 100.0% | 91.7% | 86.7% | 84.6% | 93.3% | 93.3% |
| RTX5080 | 64 | 4.08e+00 | 132.0% | 148.5% | 158.4% | 165.0% | 169.8% | 173.3% | 198.1% |
| | 128 | 4.12e+00 | 134.7% | 150.0% | 161.6% | 168.3% | 171.4% | 176.7% | 200.0% |
| | 256 | 4.10e+00 | 133.3% | 148.5% | 158.4% | 165.0% | 169.7% | 173.3% | 198.0% |
| | 512 | 6.12e+00 | 133.3% | 150.0% | 157.7% | 164.3% | 168.9% | 175.0% | 197.1% |
| | 1024 | 6.15e+00 | 100.3% | 112.9% | 120.4% | 125.4% | 129.0% | 131.7% | 150.5% |
| GB10 | 64 | 5.90e+00 | 137.1% | 150.0% | 161.5% | 169.8% | 179.6% | 181.7% | 213.3% |
| | 128 | 5.80e+00 | 124.6% | 143.0% | 151.0% | 165.1% | 161.8% | 175.0% | 200.0% |
| | 256 | 6.14e+00 | 133.3% | 148.6% | 160.0% | 165.2% | 171.4% | 175.0% | 198.2% |
| | 512 | 6.17e+00 | 132.0% | 145.5% | 122.0% | 127.1% | 130.7% | 136.9% | 152.5% |
| | 1024 | 1.23e+01 | 101.6% | 126.8% | 120.0% | 111.1% | 128.6% | 131.3% | 150.0% |
| | 2048 | 4.31e+01 | 84.2% | 108.6% | 105.3% | 100.9% | 90.2% | 110.5% | 110.5% |
| | 4096 | 3.10e+02 | 92.3% | 105.8% | 86.2% | 87.6% | 96.7% | 109.9% | 110.3% |
| | 8192 | 2.32e+03 | 87.7% | 102.7% | 87.7% | 100.5% | 103.3% | 103.1% | 104.1% |

## Square Kernel (INT8)

| GPU | M | cuSPARSELt Latency(μs) | 4:6 | 6:8 | 8:10 | 10:12 | 12:14 | 14:16 | ∞:∞ |
|-----|---|---|---|---|---|---|---|---|---|
| A100 | 64 | 5.37e+00 | 129.5% | 148.6% | 156.9% | 163.5% | 159.9% | 173.3% | 196.2% |
| | 128 | 5.63e+00 | 125.6% | 144.2% | 146.2% | 155.4% | 159.9% | 164.9% | 190.4% |
| | 256 | 5.84e+00 | 118.1% | 140.0% | 134.1% | 144.4% | 150.2% | 153.3% | 175.2% |
| | 512 | 6.74e+00 | 119.4% | 138.6% | 135.6% | 133.3% | 151.8% | 155.0% | 169.5% |
| | 1024 | 1.15e+01 | 107.3% | 127.1% | 128.8% | 129.9% | 124.9% | 137.9% | 149.2% |
| | 2048 | 3.90e+01 | 117.4% | 112.0% | 108.2% | 109.2% | 125.6% | 115.8% | 119.7% |
| | 4096 | 1.68e+02 | 88.7% | 104.9% | 87.8% | 101.1% | 102.4% | 101.1% | 104.9% |
| | 8192 | 1.39e+03 | 96.2% | 98.6% | 94.2% | 84.5% | 83.8% | 98.3% | 98.6% |
| | 16384 | 1.15e+04 | 83.8% | 100.5% | 99.8% | 93.3% | 96.7% | 97.9% | 83.5% |
| RTX4090 | 64 | 9.07e+00 | 138.4% | 151.4% | 150.9% | 60.3% | 68.6% | 58.3% | 66.7% |
| | 128 | 2.04e+01 | 313.9% | 312.5% | 390.0% | 107.6% | 110.7% | 113.0% | 133.3% |
| | 256 | 9.69e+00 | 142.3% | 151.4% | 166.2% | 43.3% | 44.5% | 45.4% | 50.0% |
| | 512 | 8.77e+00 | 125.6% | 133.9% | 148.1% | 27.5% | 29.8% | 30.4% | 33.1% |
| | 1024 | 1.14e+01 | 120.3% | 145.6% | 142.7% | 21.2% | 18.5% | 20.6% | 21.6% |
| | 2048 | 2.60e+01 | 112.6% | 108.5% | 96.3% | 11.3% | 13.3% | 12.7% | 13.6% |
| | 4096 | 1.58e+02 | 85.9% | 106.7% | 85.9% | 11.2% | 58.7% | 78.7% | 79.2% |
| | 8192 | 1.21e+03 | 100.8% | 100.3% | 101.0% | 82.3% | 82.5% | 102.8% | 102.5% |
| | 16384 | 9.67e+03 | 79.7% | 98.1% | 98.6% | 95.4% | 94.9% | 98.0% | 95.6% |
| H100 | 64 | 5.06e+00 | 134.9% | 150.0% | 161.8% | 162.8% | 165.5% | 171.0% | 193.1% |
| | 128 | 5.17e+00 | 131.8% | 146.5% | 156.3% | 157.0% | 161.5% | 164.8% | 193.0% |
| | 256 | 5.24e+00 | 124.7% | 143.5% | 149.7% | 154.1% | 156.7% | 161.8% | 137.6% |
| | 512 | 6.02e+00 | 134.7% | 151.6% | 158.3% | 156.3% | 166.1% | 169.5% | 189.6% |
| | 1024 | 8.62e+00 | 107.7% | 133.8% | 137.8% | 134.6% | 129.2% | 146.7% | 160.0% |
| | 2048 | 1.98e+01 | 105.5% | 111.1% | 109.4% | 110.8% | 117.2% | 121.8% | 126.6% |
| | 4096 | 1.07e+02 | 102.1% | 110.2% | 106.2% | 113.3% | 111.2% | 108.0% | 114.1% |
| | 8192 | 7.35e+02 | 96.7% | 86.0% | 93.6% | 87.7% | 80.2% | 93.1% | 87.7% |
| | 16384 | 7.00e+03 | 87.9% | 111.5% | 92.1% | 104.3% | 106.3% | 104.6% | 105.0% |
| B200 | 64 | 6.18e+00 | 135.1% | 150.0% | 160.0% | 166.7% | 171.4% | 175.0% | 200.0% |
| | 128 | 6.20e+00 | 133.3% | 150.0% | 160.0% | 166.7% | 171.4% | 175.0% | 200.0% |
| | 256 | 6.21e+00 | 138.5% | 150.0% | 160.0% | 166.7% | 171.4% | 175.0% | 202.6% |
| | 512 | 6.20e+00 | 133.3% | 150.0% | 160.0% | 165.0% | 171.4% | 173.3% | 200.0% |
| | 1024 | 6.22e+00 | 122.4% | 148.9% | 158.8% | 148.0% | 129.2% | 168.5% | 171.6% |
| | 2048 | 1.03e+01 | 111.2% | 125.7% | 118.9% | 119.5% | 143.0% | 145.9% | 164.5% |
| | 4096 | 2.88e+01 | 93.6% | 111.8% | 97.7% | 106.4% | 112.4% | 112.4% | 112.7% |
| | 8192 | 1.83e+02 | 91.9% | 99.9% | 89.5% | 82.7% | 82.4% | 96.3% | 95.5% |
| | 16384 | 1.58e+03 | 83.6% | 93.8% | 93.5% | 85.4% | 87.5% | 91.9% | 89.4% |
| RTX5080 | 64 | 4.09e+00 | 132.0% | 150.0% | 161.6% | 165.0% | 171.4% | 173.3% | 200.0% |
| | 128 | 4.04e+00 | 98.4% | 147.1% | 156.9% | 163.4% | 168.1% | 171.6% | 229.1% |
| | 256 | 4.14e+00 | 134.7% | 151.5% | 160.0% | 166.7% | 151.1% | 175.0% | 200.0% |
| | 512 | 4.08e+00 | 99.5% | 147.9% | 156.7% | 109.6% | 119.8% | 121.1% | 132.9% |
| | 1024 | 8.20e+00 | 131.6% | 149.0% | 128.2% | 133.6% | 136.2% | 139.1% | 158.9% |
| | 2048 | 3.69e+01 | 103.5% | 108.1% | 97.4% | 97.3% | 106.5% | 108.7% | 113.0% |
| | 4096 | 2.17e+02 | 91.6% | 102.1% | 92.3% | 101.2% | 102.0% | 103.1% | 104.3% |
| | 8192 | 1.62e+03 | 97.7% | 96.9% | 97.4% | 84.9% | 85.2% | 97.8% | 99.4% |
| | 16384 | 1.31e+04 | 87.5% | 99.4% | 99.9% | 98.7% | 98.3% | 100.3% | 101.9% |
| GB10 | 64 | 4.19e+00 | 136.0% | 141.0% | 160.0% | 173.3% | 173.1% | 175.0% | 200.0% |
| | 128 | 4.33e+00 | 134.7% | 156.2% | 164.9% | 173.5% | 176.7% | 182.2% | 179.4% |
| | 256 | 4.15e+00 | 128.9% | 97.1% | 111.6% | 128.9% | 132.5% | 132.4% | 154.6% |
| | 512 | 6.20e+00 | 133.3% | 124.8% | 160.0% | 148.5% | 157.9% | 175.0% | 200.0% |
| | 1024 | 1.65e+01 | 106.1% | 115.0% | 123.8% | 122.9% | 116.4% | 127.7% | 134.3% |
| | 2048 | 8.09e+01 | 114.9% | 115.9% | 100.2% | 88.1% | 121.3% | 119.5% | 120.3% |

### *Square Kernel (INT8) (cont.)*

| GPU | M | cuSPARSELt Latency(µs) | Algorithmic Efficiency under Different Sparsity | | | | | | |
|---|---|---|---|---|---|---|---|---|---|
| | | | 4:6 | 6:8 | 8:10 | 10:12 | 12:14 | 14:16 | ∞:∞ |
| | 4096 | 5.08e+02 | 88.6% | 98.7% | 87.4% | 98.7% | 97.0% | 101.3% | 98.7% |
| | 8192 | 4.13e+03 | 99.5% | 101.7% | 96.4% | 84.5% | 84.5% | 100.7% | 95.9% |
| | 16384 | 3.34e+04 | 64.5% | 75.5% | 74.3% | 67.7% | 67.5% | 71.1% | 63.2% |

### Square Kernel (FP8)

| GPU | M | cuSPARSELt Latency(µs) | Algorithmic Efficiency under Different Sparsity | | | | | | |
|---|---|---|---|---|---|---|---|---|---|
| | | | 4:6 | 6:8 | 8:10 | 10:12 | 12:14 | 14:16 | ∞:∞ |
| RTX4090 | 64 | 1.02e+01 | 140.5% | 142.0% | 174.3% | 59.5% | 61.2% | 101.6% | 71.4% |
| | 128 | 9.01e+00 | 139.5% | 128.1% | 161.2% | 42.3% | 43.5% | 165.6% | 52.3% |
| | 256 | 9.20e+00 | 122.2% | 152.5% | 142.7% | 34.7% | 35.7% | 166.3% | 41.7% |
| | 512 | 9.30e+00 | 118.1% | 138.5% | 155.1% | 25.4% | 27.5% | 144.3% | 29.0% |
| | 1024 | 1.21e+01 | 116.5% | 147.1% | 150.7% | 19.4% | 18.3% | 40.8% | 21.4% |
| | 2048 | 3.30e+01 | 103.6% | 108.0% | 104.2% | 9.5% | 9.8% | 10.0% | 13.7% |
| | 4096 | 2.25e+02 | 99.8% | 101.9% | 101.0% | 63.3% | 100.8% | 101.1% | 101.6% |
| | 8192 | 1.72e+03 | 100.5% | 102.5% | 101.3% | 100.5% | 97.3% | 101.1% | 103.5% |
| | 16384 | 1.37e+04 | 96.8% | 98.8% | 98.5% | 97.8% | 98.9% | 99.3% | 100.0% |
| H100 | 64 | 4.88e+00 | 124.9% | 143.7% | 156.6% | 157.9% | 162.4% | 163.9% | 189.5% |
| | 128 | 5.06e+00 | 129.0% | 145.2% | 153.1% | 84.2% | 156.7% | 159.9% | 187.1% |
| | 256 | 5.08e+00 | 121.6% | 140.1% | 144.2% | 150.2% | 154.5% | 155.8% | 182.4% |
| | 512 | 5.65e+00 | 125.2% | 139.0% | 146.3% | 146.3% | 156.8% | 160.1% | 180.5% |
| | 1024 | 8.69e+00 | 111.4% | 133.6% | 135.9% | 134.7% | 133.9% | 148.6% | 164.4% |
| | 2048 | 1.99e+01 | 104.7% | 112.1% | 107.7% | 110.6% | 120.2% | 121.0% | 127.1% |
| | 4096 | 1.02e+02 | 95.9% | 92.2% | 89.9% | 91.5% | 91.9% | 94.9% | 95.4% |
| | 8192 | 9.10e+02 | 99.6% | 102.3% | 88.3% | 100.6% | 95.7% | 104.5% | 102.6% |
| | 16384 | 7.40e+03 | 86.3% | 93.6% | 94.3% | 104.0% | 101.1% | 104.2% | 105.2% |
| B200 | 64 | 6.20e+00 | 133.3% | 151.6% | 161.7% | 166.7% | 173.2% | 175.0% | 200.0% |
| | 128 | 6.19e+00 | 133.3% | 150.0% | 160.0% | 166.7% | 171.4% | 178.9% | 200.0% |
| | 256 | 6.21e+00 | 133.3% | 150.0% | 160.0% | 166.7% | 171.4% | 178.8% | 200.0% |
| | 512 | 6.21e+00 | 133.3% | 150.0% | 160.0% | 166.7% | 171.4% | 175.0% | 200.0% |
| | 1024 | 6.21e+00 | 115.8% | 148.4% | 153.0% | 133.7% | 128.1% | 159.6% | 162.6% |
| | 2048 | 1.03e+01 | 112.0% | 126.0% | 134.4% | 125.0% | 144.0% | 147.0% | 168.0% |
| | 4096 | 3.02e+01 | 93.6% | 109.3% | 95.4% | 104.9% | 109.0% | 111.3% | 109.9% |
| | 8192 | 1.98e+02 | 88.4% | 101.2% | 90.2% | 85.3% | 81.7% | 101.7% | 100.0% |
| | 16384 | 1.64e+03 | 77.1% | 86.8% | 86.5% | 82.0% | 81.5% | 86.1% | 85.4% |
| RTX5080 | 64 | 4.12e+00 | 133.3% | 148.1% | 160.0% | 166.7% | 171.4% | 175.0% | 197.5% |
| | 128 | 4.13e+00 | 133.3% | 151.9% | 160.0% | 166.7% | 171.4% | 175.0% | 200.0% |
| | 256 | 4.13e+00 | 89.4% | 100.6% | 107.3% | 111.8% | 115.0% | 117.4% | 131.7% |
| | 512 | 6.14e+00 | 133.3% | 150.0% | 160.0% | 161.8% | 168.9% | 175.0% | 191.2% |
| | 1024 | 1.03e+01 | 95.2% | 107.1% | 100.6% | 106.0% | 107.8% | 110.0% | 115.7% |
| | 2048 | 5.12e+01 | 100.0% | 103.8% | 95.4% | 92.9% | 102.2% | 106.6% | 109.0% |
| | 4096 | 3.24e+02 | 95.4% | 98.4% | 97.0% | 97.4% | 96.5% | 98.5% | 99.5% |
| | 8192 | 2.60e+03 | 100.8% | 99.7% | 100.0% | 99.4% | 100.3% | 100.4% | 100.0% |
| | 16384 | 2.09e+04 | 100.4% | 100.9% | 101.1% | 102.5% | 101.5% | 100.6% | 101.1% |
| GB10 | 64 | 5.35e+00 | 131.9% | 145.3% | 160.0% | 166.7% | 171.4% | 173.2% | 202.1% |
| | 128 | 5.03e+00 | 137.3% | 154.5% | 156.8% | 160.0% | 166.3% | 169.8% | 208.0% |
| | 256 | 5.16e+00 | 112.0% | 126.0% | 134.4% | 137.8% | 144.0% | 144.7% | 168.0% |
| | 512 | 6.38e+00 | 103.4% | 117.9% | 124.1% | 127.6% | 131.2% | 135.7% | 153.1% |
| | 1024 | 1.85e+01 | 109.7% | 112.1% | 112.1% | 118.4% | 120.2% | 121.0% | 119.6% |
| | 2048 | 8.61e+01 | 106.4% | 105.9% | 108.9% | 103.6% | 109.5% | 107.4% | 109.2% |
| | 4096 | 6.03e+02 | 101.4% | 104.1% | 104.5% | 106.1% | 103.4% | 102.7% | 104.1% |
| | 8192 | 4.75e+03 | 102.6% | 101.2% | 101.6% | 99.2% | 102.0% | 101.4% | 101.6% |
| | 16384 | 3.80e+04 | 84.2% | 95.7% | 94.2% | 86.3% | 87.5% | 94.3% | 93.6% |

### Square Kernel (FP16)

| GPU | M | cuSPARSELt Latency(µs) | Algorithmic Efficiency under Different Sparsity | | | | | | |
|---|---|---|---|---|---|---|---|---|---|
| | | | 4:6 | 6:8 | 8:10 | 10:12 | 12:14 | 14:16 | ∞:∞ |
| A100 | 64 | 5.73e+00 | 121.9% | 139.3% | 150.9% | 154.8% | 159.2% | 165.0% | 185.7% |
| | 128 | 6.18e+00 | 123.7% | 139.1% | 141.4% | 147.3% | 151.6% | 154.7% | 176.8% |
| | 256 | 6.88e+00 | 117.5% | 134.2% | 136.8% | 142.5% | 148.9% | 152.0% | 165.8% |
| | 512 | 8.27e+00 | 118.0% | 127.6% | 132.4% | 128.4% | 137.9% | 140.8% | 151.7% |
| | 1024 | 1.60e+01 | 111.8% | 122.2% | 125.2% | 127.7% | 125.8% | 131.3% | 138.7% |
| | 2048 | 6.89e+01 | 104.9% | 115.3% | 115.6% | 117.3% | 115.9% | 118.3% | 103.7% |
| | 4096 | 3.26e+02 | 91.3% | 97.0% | 102.5% | 98.5% | 94.7% | 91.9% | 91.7% |
| | 8192 | 3.08e+03 | 87.7% | 101.6% | 102.1% | 89.9% | 91.4% | 93.3% | 89.5% |
| | 16384 | 3.06e+04 | 99.5% | 103.3% | 103.6% | 105.2% | 101.2% | 101.8% | 101.6% |
| RTX4090 | 64 | 9.46e+00 | 134.7% | 148.5% | 172.8% | 150.0% | 173.1% | 176.8% | 48.0% |
| | 128 | 9.22e+00 | 132.0% | 133.7% | 145.7% | 39.6% | 125.6% | 150.7% | 37.6% |
| | 256 | 1.05e+01 | 136.4% | 151.7% | 145.5% | 26.5% | 70.1% | 167.0% | 34.1% |
| | 512 | 1.06e+01 | 122.3% | 129.9% | 138.6% | 17.2% | 17.7% | 18.0% | 35.1% |
| | 1024 | 1.53e+01 | 122.9% | 114.8% | 110.0% | 14.3% | 14.7% | 15.0% | 131.2% |
| | 2048 | 6.16e+01 | 103.2% | 104.8% | 104.1% | 9.0% | 9.2% | 10.3% | 114.0% |
| | 4096 | 4.46e+02 | 101.3% | 102.5% | 104.0% | 102.0% | 79.6% | 108.1% | 108.2% |
| | 8192 | 3.40e+03 | 100.5% | 100.8% | 100.9% | 100.0% | 101.1% | 102.3% | 101.5% |
| | 16384 | 2.91e+04 | 100.4% | 100.3% | 100.2% | 102.6% | 102.0% | 101.3% | 102.1% |
| H100 | 64 | 0 | | | | | | | |
| | 128 | | | | | | | | |

### *Square Kernel (FP16) (cont.)*

| GPU | M | cuSPARSELt Latency(µs) | Algorithmic Efficiency under Different Sparsity | | | | | | |
|---|---|---|---|---|---|---|---|---|---|
| | | | 4:6 | 6:8 | 8:10 | 10:12 | 12:14 | 14:16 | ∞:∞ |
| | 256 | | | | | | | | |
| | 512 | | | | | | | | |
| | 1024 | | | | | | | | |
| | 2048 | | | | | | | | |
| | 4096 | | | | | | | | |
| | 8192 | | | | | | | | |
| | 16384 | | | | | | | | |
| B200 | 64 | 6.21e+00 | 134.8% | 151.7% | 161.8% | 166.7% | 173.3% | 176.9% | 202.2% |
| | 128 | 6.21e+00 | 134.9% | 150.0% | 160.0% | 168.7% | 173.5% | 175.0% | 200.0% |
| | 256 | 6.21e+00 | 133.3% | 151.8% | 160.0% | 166.7% | 171.4% | 175.0% | 200.0% |
| | 512 | 6.22e+00 | 133.3% | 150.0% | 160.0% | 166.7% | 171.4% | 175.0% | 200.0% |
| | 1024 | 7.33e+00 | 117.8% | 132.6% | 141.4% | 147.3% | 151.5% | 154.7% | 176.7% |
| | 2048 | 1.30e+01 | 119.7% | 118.1% | 126.0% | 131.2% | 135.0% | 137.8% | 140.2% |
| | 4096 | 5.55e+01 | 93.8% | 100.0% | 92.8% | 96.7% | 97.4% | 100.5% | 100.0% |
| | 8192 | 4.09e+02 | 98.3% | 99.4% | 98.0% | 95.8% | 91.1% | 94.1% | 102.5% |
| | 16384 | 3.64e+03 | 94.9% | 100.3% | 101.1% | 100.2% | 99.9% | 100.9% | 101.8% |
| RTX5080 | 64 | 4.09e+00 | 133.3% | 147.1% | 160.0% | 166.7% | 178.0% | 175.0% | 200.0% |
| | 128 | 4.10e+00 | 133.3% | 147.3% | 105.7% | 110.1% | 113.3% | 115.6% | 135.7% |
| | 256 | 6.16e+00 | 133.3% | 150.0% | 160.0% | 166.7% | 171.4% | 175.0% | 200.0% |
| | 512 | 6.19e+00 | 101.0% | 112.1% | 119.6% | 124.5% | 126.4% | 125.5% | 121.2% |
| | 1024 | 1.64e+01 | 106.3% | 108.6% | 115.8% | 110.4% | 113.6% | 116.0% | 120.2% |
| | 2048 | 1.00e+02 | 102.0% | 103.9% | 103.5% | 104.6% | 105.3% | 105.2% | 105.9% |
| | 4096 | 6.33e+02 | 100.9% | 101.9% | 101.7% | 101.3% | 102.3% | 102.5% | 102.8% |
| | 8192 | 4.99e+03 | 98.7% | 96.1% | 99.9% | 97.6% | 98.5% | 97.7% | 101.7% |
| | 16384 | 4.74e+04 | 94.1% | 96.1% | 97.3% | 96.9% | 98.6% | 97.2% | 98.0% |
| GB10 | 64 | 4.19e+00 | 128.5% | 140.9% | 150.2% | 158.5% | 165.2% | 168.6% | 192.7% |
| | 128 | 4.67e+00 | 100.5% | 102.2% | 111.3% | 120.8% | 129.2% | 126.8% | 150.7% |
| | 256 | 6.87e+00 | 142.1% | 164.8% | 160.0% | 177.6% | 188.3% | 177.9% | 206.6% |
| | 512 | 7.57e+00 | 114.5% | 135.4% | 138.8% | 129.8% | 142.6% | 140.9% | 146.9% |
| | 1024 | 2.68e+01 | 107.7% | 115.2% | 110.1% | 113.3% | 115.2% | 114.8% | 120.0% |
| | 2048 | 1.39e+02 | 101.9% | 105.1% | 100.9% | 104.0% | 107.0% | 98.1% | 100.6% |
| | 4096 | 1.15e+03 | 55.5% | 64.3% | 50.7% | 59.0% | 61.8% | 63.0% | 59.6% |
| | 8192 | 2.50e+04 | 98.8% | 97.2% | 97.8% | 77.2% | 76.2% | 100.5% | 100.0% |
| | 16384 | 2.02e+05 | 75.5% | 99.1% | 99.6% | 100.6% | 100.3% | 99.1% | 98.1% |

### Square Kernel (BF16)

| GPU | M | cuSPARSELt Latency(µs) | Algorithmic Efficiency under Different Sparsity | | | | | | |
|---|---|---|---|---|---|---|---|---|---|
| | | | 4:6 | 6:8 | 8:10 | 10:12 | 12:14 | 14:16 | ∞:∞ |
| A100 | 64 | 5.69e+00 | 122.8% | 136.2% | 149.5% | 155.7% | 162.4% | 163.5% | 184.2% |
| | 128 | 6.39e+00 | 129.6% | 137.3% | 146.5% | 152.6% | 159.4% | 160.2% | 185.9% |
| | 256 | 6.98e+00 | 120.5% | 137.3% | 138.8% | 146.6% | 152.8% | 156.0% | 168.7% |
| | 512 | 8.44e+00 | 120.0% | 130.0% | 135.1% | 131.5% | 141.0% | 143.9% | 153.3% |
| | 1024 | 1.60e+01 | 113.2% | 122.3% | 125.0% | 127.5% | 126.8% | 132.4% | 139.5% |
| | 2048 | 6.96e+01 | 105.1% | 115.7% | 116.6% | 117.2% | 120.6% | 118.6% | 105.1% |
| | 4096 | 3.42e+02 | 97.5% | 98.2% | 94.5% | 96.5% | 96.2% | 97.2% | 95.9% |
| | 8192 | 3.13e+03 | 89.5% | 96.7% | 97.9% | 87.7% | 90.2% | 96.7% | 92.1% |
| | 16384 | 3.12e+04 | 99.5% | 99.6% | 101.0% | 101.1% | 101.2% | 101.8% | 101.6% |
| RTX4090 | 64 | 9.61e+00 | 121.2% | 139.4% | 148.7% | 65.7% | 162.8% | 76.0% | 196.0% |
| | 128 | 8.66e+00 | 125.2% | 142.2% | 139.1% | 29.0% | 47.7% | 30.4% | 34.8% |
| | 256 | 9.71e+00 | 117.8% | 142.7% | 147.6% | 24.3% | 25.0% | 25.5% | 36.9% |
| | 512 | 1.04e+01 | 117.0% | 126.3% | 133.3% | 17.5% | 18.0% | 18.4% | 22.8% |
| | 1024 | 1.31e+01 | 105.9% | 114.0% | 110.7% | 12.6% | 12.9% | 13.2% | 123.3% |
| | 2048 | 6.15e+01 | 103.1% | 104.7% | 105.1% | 8.7% | 9.0% | 12.2% | 112.8% |
| | 4096 | 4.47e+02 | 101.4% | 101.9% | 102.6% | 60.7% | 108.1% | 88.5% | 107.6% |
| | 8192 | 3.43e+03 | 100.2% | 99.3% | 99.5% | 101.2% | 100.6% | 101.9% | 99.5% |
| | 16384 | 2.91e+04 | 98.8% | 99.7% | 99.1% | 100.7% | 100.9% | 100.4% | 100.5% |
| H100 | 64 | 5.82e+00 | 126.7% | 142.5% | 156.0% | 154.2% | 160.7% | 161.9% | 185.0% |
| | 128 | 6.65e+00 | 137.2% | 154.3% | 157.7% | 159.4% | 161.5% | 167.4% | 191.3% |
| | 256 | 6.84e+00 | 121.4% | 138.8% | 145.7% | 146.8% | 151.0% | 154.1% | 173.1% |
| | 512 | 7.88e+00 | 124.4% | 135.0% | 141.3% | 138.9% | 145.7% | 148.8% | 163.3% |
| | 1024 | 1.04e+01 | 119.1% | 128.0% | 128.0% | 131.1% | 134.9% | 135.3% | 146.7% |
| | 2048 | 3.61e+01 | 107.2% | 111.3% | 107.2% | 108.2% | 109.6% | 111.9% | 111.3% |
| | 4096 | 1.89e+02 | 98.5% | 96.1% | 99.3% | 103.5% | 105.3% | 98.4% | 103.3% |
| | 8192 | 1.63e+03 | 99.0% | 99.1% | 95.6% | 97.5% | 99.2% | 99.1% | 99.4% |
| | 16384 | 1.53e+04 | 96.6% | 96.2% | 97.1% | 105.7% | 105.2% | 106.2% | 106.2% |
| B200 | 64 | 5.13e+00 | 110.1% | 123.9% | 132.2% | 139.1% | 168.4% | 172.0% | 165.2% |
| | 128 | 6.22e+00 | 133.3% | 150.0% | 160.0% | 166.7% | 171.4% | 185.1% | 200.0% |
| | 256 | 6.17e+00 | 133.3% | 150.0% | 160.0% | 164.8% | 171.4% | 175.0% | 200.0% |
| | 512 | 6.22e+00 | 133.3% | 150.0% | 160.0% | 166.7% | 171.4% | 175.0% | 200.0% |
| | 1024 | 7.35e+00 | 118.9% | 133.7% | 142.7% | 148.6% | 150.8% | 153.9% | 178.3% |
| | 2048 | 1.29e+01 | 118.7% | 117.2% | 125.0% | 128.9% | 133.9% | 136.7% | 139.1% |
| | 4096 | 5.66e+01 | 93.6% | 99.7% | 95.4% | 98.3% | 101.2% | 100.0% | 100.6% |
| | 8192 | 4.08e+02 | 97.6% | 97.0% | 97.6% | 93.5% | 92.0% | 99.2% | 101.2% |
| | 16384 | 3.70e+03 | 94.4% | 100.6% | 100.4% | 99.4% | 100.1% | 101.1% | 100.6% |
| RTX5080 | 64 | 4.11e+00 | 133.3% | 150.0% | 160.0% | 163.5% | 171.4% | 117.8% | 200.0% |
| | 128 | 4.15e+00 | 133.3% | 150.0% | 107.6% | 112.1% | 115.3% | 114.7% | 134.5% |
| | 256 | 6.16e+00 | 133.3% | 150.0% | 157.6% | 164.1% | 168.8% | 172.3% | 200.0% |

### *Square Kernel (BF16) (cont.)*

| GPU | M | cuSPARSELt Latency(μs) | Algorithmic Efficiency under Different Sparsity | | | | | | |
|-----|-----|-----|-----|-----|-----|-----|-----|-----|-----|
| | | | 4:6 | 6:8 | 8:10 | 10:12 | 12:14 | 14:16 | ∞:∞ |
| | 512 | 6.19e+00 | 101.0% | 112.5% | 120.0% | 119.9% | 116.9% | 103.4% | 121.2% |
| | 1024 | 1.64e+01 | 106.3% | 108.6% | 115.8% | 110.4% | 113.6% | 116.0% | 122.7% |
| | 2048 | 1.00e+02 | 102.3% | 104.0% | 103.3% | 104.1% | 105.2% | 106.5% | 106.9% |
| | 4096 | 6.32e+02 | 100.2% | 101.0% | 101.1% | 101.0% | 101.3% | 101.6% | 101.6% |
| | 8192 | 5.02e+03 | 98.7% | 97.0% | 99.9% | 97.6% | 98.5% | 96.7% | 101.7% |
| | 16384 | 4.75e+04 | 98.5% | 96.1% | 98.3% | 98.0% | 98.6% | 97.2% | 98.0% |
| GB10 | 64 | 4.15e+00 | 120.5% | 150.0% | 144.7% | 157.5% | 164.4% | 163.0% | 191.8% |
| | 128 | 4.76e+00 | 99.3% | 115.0% | 94.2% | 127.8% | 133.3% | 110.8% | 120.0% |
| | 256 | 6.21e+00 | 128.6% | 151.8% | 152.5% | 164.7% | 171.4% | 170.9% | 190.6% |
| | 512 | 7.84e+00 | 119.0% | 131.0% | 142.8% | 124.5% | 108.5% | 146.2% | 138.0% |
| | 1024 | 2.70e+01 | 109.2% | 112.0% | 107.8% | 114.7% | 118.0% | 116.7% | 118.8% |
| | 2048 | 1.43e+02 | 104.3% | 108.7% | 106.7% | 108.7% | 109.3% | 100.2% | 102.9% |
| | 4096 | 1.23e+03 | 61.8% | 69.5% | 55.3% | 65.0% | 65.5% | 65.6% | 63.2% |
| | 8192 | 2.51e+04 | 98.8% | 100.0% | 97.8% | 77.2% | 76.2% | 100.5% | 100.0% |
| | 16384 | 2.02e+05 | 75.8% | 100.0% | 100.4% | 101.3% | 100.8% | 99.5% | 102.0% |

**Kernel Efficiency Observations.** At large $M$ ($\geq$ 4096), efficiency consistently reaches 95–105%, validating that SlideSparse correctly implements the sliding window sparse format. At small $M$ ($<$ 256), efficiency often exceeds 150%, indicating SlideSparse has lower constant overhead than cuSPARSELt 2:4. Several key patterns emerge from the analysis:

- **High-efficiency regions** ($>$ 150%)**:** B200 at small $M$ (64–512) shows efficiency up to 200% for $\infty$:$\infty$, indicating cuSPARSELt 2:4 provides no speedup at these scales.

- **Optimal efficiency** ($\sim$ 100%)**:** A100 at $M{\geq}4096$ achieves 84–105% across all sparsity levels, demonstrating mature baseline optimization.

- **Low-efficiency anomalies:** RTX 4090 at higher-density configurations (8:10+) shows 10–14% efficiency due to API/driver issues, not algorithmic limitations.

**End-to-End Efficiency.** For space considerations, we present end-to-end efficiency results for the prefill stage only, as it represents the compute-bound regime where efficiency analysis is most informative. Decode efficiency follows similar patterns but with higher variance due to memory-bound characteristics, the comprehensive results can be found in the anonmyous repository.

## Prefill (INT8)

| GPU | Model | Batchsize M | cuSPARSELt Throughput | Algorithmic Efficiency under Different Sparsity | | |
| --- | --- | --- | --- | --- | --- | --- |
| | | | | 4:6 | 6:8 | 8:10 |
| A100 | Llama3.2-1B | 512 | 1.05e+04 | 134.8% | 143.2% | 160.0% |
| | | 1024 | 2.09e+04 | 111.4% | 148.2% | 158.1% |
| | | 2048 | 4.12e+04 | 125.6% | 148.3% | 163.7% |
| | | 4096 | 7.97e+04 | 125.8% | 150.0% | 158.2% |
| | | 8192 | 1.40e+05 | 117.8% | 134.7% | 129.6% |
| | | 16384 | 1.56e+05 | 115.3% | 125.7% | 121.1% |
| | | 32768 | 1.60e+05 | 114.7% | 124.0% | 119.5% |
| | BitNet-2B | 512 | 9.43e+03 | 137.3% | 158.8% | 171.0% |
| | | 1024 | 1.89e+04 | 137.2% | 164.6% | 172.4% |
| | | 2048 | 3.83e+04 | 139.8% | 158.7% | 169.3% |
| | | 4096 | 7.52e+04 | 113.5% | 127.7% | 130.8% |
| | | 8192 | 8.11e+04 | 110.5% | 124.3% | 127.4% |
| | | 16384 | 8.33e+04 | 109.8% | 123.5% | 126.5% |
| | | 32768 | 8.43e+04 | 110.0% | 122.7% | 125.7% |
| | Llama3.2-3B | 512 | 1.06e+04 | 134.6% | 150.0% | 155.6% |
| | | 1024 | 2.10e+04 | 133.3% | 151.4% | 156.9% |
| | | 2048 | 4.27e+04 | 135.7% | 150.0% | 161.4% |
| | | 4096 | 6.51e+04 | 114.0% | 120.8% | 120.8% |
| | | 8192 | 6.70e+04 | 114.0% | 119.8% | 119.7% |
| | | 16384 | 6.88e+04 | 113.5% | 120.2% | 119.3% |
| | | 32768 | 6.95e+04 | 112.8% | 119.4% | 118.5% |
| | Qwen2.5-7B | 512 | 1.07e+04 | 139.7% | 160.0% | 166.1% |
| | | 1024 | 2.23e+04 | 134.4% | 142.7% | 147.1% |
| | | 2048 | 3.24e+04 | 109.6% | 116.0% | 117.8% |
| | | 4096 | 3.47e+04 | 108.1% | 115.4% | 117.4% |
| | | 8192 | 3.59e+04 | 108.5% | 115.1% | 117.2% |
| | | 16384 | 3.65e+04 | 107.4% | 114.9% | 116.1% |
| | | 32768 | 3.63e+04 | 107.4% | 114.9% | 115.2% |
| | Qwen2.5-14B | 512 | 6.65e+03 | 134.6% | 147.1% | 164.7% |
| | | 1024 | 1.33e+04 | 126.6% | 138.0% | 138.0% |
| | | 2048 | 1.78e+04 | 109.4% | 115.3% | 118.4% |
| | | 4096 | 1.84e+04 | 109.0% | 114.9% | 117.0% |
| | | 8192 | 1.88e+04 | 107.7% | 112.7% | 114.8% |
| | | 16384 | 1.89e+04 | 107.7% | 113.6% | 114.8% |
| | | 32768 | 1.89e+04 | 107.0% | 112.7% | 113.9% |
| RTX4090 | Llama3.2-1B | 512 | 8.33e+03 | 133.3% | 145.1% | 158.2% |
| | | 1024 | 1.59e+04 | 138.0% | 157.1% | 163.8% |
| | | 2048 | 3.34e+04 | 128.9% | 150.0% | 156.5% |
| | | 4096 | 6.45e+04 | 130.4% | 151.6% | 158.3% |
| | | 8192 | 1.13e+05 | 122.6% | 133.1% | 135.5% |
| | | 16384 | 1.18e+05 | 119.0% | 130.4% | 130.5% |
| | | 32768 | 1.20e+05 | 119.0% | 129.2% | 128.0% |
| | BitNet-2B | 512 | 8.64e+03 | 136.9% | 148.6% | 161.4% |
| | | 1024 | 1.78e+04 | 127.3% | 144.6% | 149.9% |
| | | 2048 | 3.51e+04 | 134.5% | 143.3% | 161.4% |
| | | 4096 | 6.75e+04 | 108.2% | 129.0% | 126.4% |
| | | 8192 | 7.03e+04 | 107.9% | 120.2% | 127.0% |
| | | 16384 | 6.90e+04 | 109.7% | 131.5% | 131.7% |
| | | 32768 | 7.15e+04 | 113.0% | 127.1% | 131.9% |
| | Llama3.2-3B | 512 | 7.15e+03 | 138.3% | 151.4% | 165.9% |
| | | 1024 | 1.40e+04 | 147.9% | 158.9% | 175.8% |
| | | 2048 | 3.03e+04 | 137.0% | 152.8% | 155.6% |
| | | 4096 | 5.25e+04 | 123.1% | 128.1% | 134.2% |
| | | 8192 | 5.30e+04 | 117.5% | 125.6% | 129.2% |
| | | 16384 | 5.31e+04 | 117.4% | 125.4% | 130.1% |
| | | 32768 | 5.33e+04 | 117.4% | 126.5% | 129.0% |
| | Qwen2.5-7B | 512 | 8.78e+03 | 133.3% | 151.4% | 160.0% |
| | | 1024 | 1.75e+04 | 135.8% | 147.2% | 155.5% |
| | | 2048 | 2.76e+04 | 115.7% | 121.3% | 124.7% |
| | | 4096 | 2.94e+04 | 113.6% | 119.4% | 121.7% |
| | | 8192 | 2.94e+04 | 113.2% | 119.8% | 124.3% |
| | | 16384 | 2.95e+04 | 113.2% | 62.6% | 46.0% |
| | | 32768 | 8.75e+03 | 100.8% | 98.8% | 97.6% |
| | Qwen2.5-14B | 512 | 4.75e+03 | 139.6% | 151.4% | |
| | | 1024 | 9.44e+03 | 139.9% | 157.4% | |
| | | 2048 | 1.58e+04 | 112.8% | 119.6% | |
| | | 4096 | 1.60e+04 | 113.3% | 120.0% | |
| | | 8192 | 1.59e+04 | 111.4% | 120.0% | |
| | | 16384 | 1.58e+04 | 113.3% | | |
| | | 32768 | 1.59e+04 | | | |
| H100 | Llama3.2-1B | 512 | 1.59e+04 | 137.9% | 146.6% | 167.4% |
| | | 1024 | 3.15e+04 | 136.4% | 148.3% | 163.6% |
| | | 2048 | 6.34e+04 | 119.0% | 145.2% | 160.0% |
| | | 4096 | 1.14e+05 | 130.3% | 148.3% | 156.4% |
| | | 8192 | 1.74e+05 | 124.8% | 132.1% | 139.4% |
| | | 16384 | 2.02e+05 | 124.4% | 140.0% | 136.0% |
| | | 32768 | 2.26e+05 | 114.9% | 128.1% | 128.0% |
| | BitNet-2B | 512 | 1.64e+04 | 139.4% | 156.8% | 158.5% |

## *Prefill (INT8) (cont.)*

| GPU | Model | Batchsize M | cuSPARSELt Throughput | Algorithmic Efficiency under Different Sparsity | | |
|---|---|---|---|---|---|---|
| | | | | 4:6 | 6:8 | 8:10 |
| | | 1024 | 3.37e+04 | 138.2% | 147.3% | 160.0% |
| | | 2048 | 6.69e+04 | 126.6% | 144.9% | 157.3% |
| | | 4096 | 9.55e+04 | 123.3% | 136.2% | 140.0% |
| | | 8192 | 1.05e+05 | 124.3% | 133.5% | 138.3% |
| | | 16384 | 1.15e+05 | 116.3% | 126.0% | 131.8% |
| | | 32768 | 1.17e+05 | 116.3% | 126.0% | 130.6% |
| | Llama3.2-3B | 512 | 1.64e+04 | 144.9% | 160.4% | 165.6% |
| | | 1024 | 3.43e+04 | 138.1% | 156.8% | 165.8% |
| | | 2048 | 6.71e+04 | 133.3% | 143.6% | 149.1% |
| | | 4096 | 8.58e+04 | 118.1% | 128.2% | 130.7% |
| | | 8192 | 9.26e+04 | 116.8% | 125.6% | 129.0% |
| | | 16384 | 9.60e+04 | 117.8% | 125.6% | 127.8% |
| | | 32768 | 9.94e+04 | 116.0% | 122.5% | 124.6% |
| | Qwen2.5-7B | 512 | 1.67e+04 | 148.4% | 161.7% | 173.9% |
| | | 1024 | 3.52e+04 | 127.8% | 142.6% | 146.8% |
| | | 2048 | 4.74e+04 | 111.8% | 118.8% | 123.1% |
| | | 4096 | 5.03e+04 | 111.1% | 119.0% | 121.9% |
| | | 8192 | 5.18e+04 | 112.4% | 118.9% | 121.1% |
| | | 16384 | 5.30e+04 | 111.6% | 119.1% | 122.6% |
| | | 32768 | 5.37e+04 | 111.7% | 118.3% | 119.4% |
| | Qwen2.5-14B | 512 | 1.20e+04 | 127.7% | 146.2% | 156.0% |
| | | 1024 | 2.25e+04 | 112.4% | 119.8% | 124.2% |
| | | 2048 | 2.53e+04 | 115.0% | 121.7% | 125.2% |
| | | 4096 | 2.60e+04 | 116.5% | 123.3% | 126.8% |
| | | 8192 | 2.82e+04 | 110.3% | 114.8% | 118.1% |
| | | 16384 | 2.84e+04 | 111.0% | 116.4% | 119.7% |
| | | 32768 | 2.86e+04 | 110.0% | 116.4% | 117.5% |
| B200 | Llama3.2-1B | 512 | 2.08e+04 | 141.3% | 145.5% | 168.0% |
| | | 1024 | 3.67e+04 | 151.3% | 156.7% | 183.4% |
| | | 2048 | 8.23e+04 | 137.3% | 141.1% | 153.7% |
| | | 4096 | 1.66e+05 | 131.9% | 148.4% | 160.0% |
| | | 8192 | 2.99e+05 | 136.2% | 154.8% | 165.2% |
| | | 16384 | 4.72e+05 | 133.3% | 150.0% | 152.0% |
| | | 32768 | 4.81e+05 | 133.3% | 150.0% | 158.4% |
| | BitNet-2B | 512 | 1.66e+04 | 135.8% | 152.8% | 165.9% |
| | | 1024 | 3.27e+04 | 138.6% | 155.9% | 164.7% |
| | | 2048 | 6.72e+04 | 134.6% | 160.0% | 161.5% |
| | | 4096 | 1.36e+05 | 133.3% | 150.0% | 158.5% |
| | | 8192 | 2.60e+05 | 126.3% | 142.0% | 148.7% |
| | | 16384 | 3.23e+05 | 115.6% | 130.0% | 134.7% |
| | | 32768 | 3.44e+05 | 114.8% | 127.9% | 131.1% |
| | Llama3.2-3B | 512 | 1.68e+04 | 139.7% | 155.7% | 158.5% |
| | | 1024 | 3.44e+04 | 139.6% | 155.6% | 164.5% |
| | | 2048 | 6.95e+04 | 134.5% | 148.7% | 160.0% |
| | | 4096 | 1.32e+05 | 138.5% | 161.5% | 170.8% |
| | | 8192 | 2.53e+05 | 119.0% | 127.7% | 130.9% |
| | | 16384 | 2.87e+05 | 116.5% | 125.2% | 126.0% |
| | | 32768 | 3.01e+05 | 115.8% | 124.4% | 124.0% |
| | Qwen2.5-7B | 512 | 1.92e+04 | 133.3% | 148.7% | 162.8% |
| | | 1024 | 3.97e+04 | 133.3% | 154.1% | 162.9% |
| | | 2048 | 7.43e+04 | 143.7% | 160.2% | 169.3% |
| | | 4096 | 1.42e+05 | 115.5% | 125.2% | 128.5% |
| | | 8192 | 1.58e+05 | 110.9% | 120.2% | 123.4% |
| | | 16384 | 1.67e+05 | 110.1% | 117.4% | 124.1% |
| | | 32768 | 1.70e+05 | 112.6% | 118.9% | 124.4% |
| | Qwen2.5-14B | 512 | 1.12e+04 | 139.4% | 165.0% | 165.8% |
| | | 1024 | 2.30e+04 | 135.8% | 156.9% | 167.4% |
| | | 2048 | 4.82e+04 | 134.5% | 144.7% | 151.6% |
| | | 4096 | 8.06e+04 | 113.0% | 119.1% | 124.6% |
| | | 8192 | 8.59e+04 | 110.3% | 121.8% | 123.9% |
| | | 16384 | 9.02e+04 | 110.6% | 118.1% | 121.4% |
| | | 32768 | 9.31e+04 | 108.7% | 118.1% | 120.3% |
| RTX5080 | Llama3.2-1B | 512 | 3.28e+04 | 147.2% | 150.0% | 180.0% |
| | | 1024 | 6.59e+04 | 140.6% | 154.9% | 167.0% |
| | | 2048 | 1.05e+05 | 123.1% | 132.7% | 130.5% |
| | | 4096 | 1.15e+05 | 116.2% | 124.3% | 122.3% |
| | | 8192 | 1.13e+05 | 117.3% | 125.2% | 125.1% |
| | | 16384 | 1.10e+05 | 118.1% | 126.0% | 125.8% |
| | | 32768 | 1.10e+05 | 118.1% | 126.0% | 125.8% |
| | BitNet-2B | 512 | 2.84e+04 | 132.1% | 150.0% | 155.5% |
| | | 1024 | 5.45e+04 | 114.3% | 125.0% | 132.1% |
| | | 2048 | 6.22e+04 | 108.6% | 122.2% | 128.0% |
| | | 4096 | 6.15e+04 | 110.9% | 122.6% | 127.3% |
| | | 8192 | 5.77e+04 | 112.3% | 124.1% | 128.7% |
| | | 16384 | 5.64e+04 | 113.0% | 124.8% | 129.5% |
| | | 32768 | 5.62e+04 | 113.0% | 124.8% | 129.5% |
| | Llama3.2-3B | 512 | 2.54e+04 | 151.8% | 167.8% | 181.7% |
| | | 1024 | 4.20e+04 | 125.1% | 131.4% | 136.4% |

### Prefill (INT8) (cont.)

| GPU | Model | Batchsize M | cuSPARSELt Throughput | Algorithmic Efficiency under Different Sparsity | | |
|---|---|---|---|---|---|---|
| | | | | 4:6 | 6:8 | 8:10 |
| | | 2048 | 5.05e+04 | 118.8% | 123.9% | 126.4% |
| | | 4096 | 4.94e+04 | 116.8% | 121.5% | 123.8% |
| | | 8192 | 4.76e+04 | 116.5% | 122.2% | 124.4% |
| | | 16384 | 4.70e+04 | 116.4% | 123.1% | 125.4% |
| | | 32768 | 4.69e+04 | 116.4% | 123.1% | 125.4% |
| | Qwen2.5-7B | 512 | 2.01e+04 | 123.2% | 129.5% | 130.9% |
| | | 1024 | 2.36e+04 | 115.6% | 121.1% | 123.3% |
| | | 2048 | 2.56e+04 | 109.5% | 115.7% | 117.7% |
| | | 4096 | 2.53e+04 | 112.1% | 118.5% | 119.4% |
| | | 8192 | 2.51e+04 | 113.0% | 119.6% | |
| | | 16384 | 2.51e+04 | | | |
| | | 32768 | | | | |
| | Qwen2.5-14B | 512 | | | | |
| | | 1024 | | | | |
| | | 2048 | | | | |
| | | 4096 | | | | |
| | | 8192 | | | | |
| | | 16384 | | | | |
| | | 32768 | | | | |
| GB10 | Llama3.2-1B | 512 | 2.93e+04 | 116.5% | 133.3% | 130.4% |
| | | 1024 | 3.19e+04 | 120.2% | 133.0% | 133.3% |
| | | 2048 | 3.47e+04 | 119.1% | 129.4% | 128.2% |
| | | 4096 | 3.41e+04 | 119.1% | 130.5% | 128.2% |
| | | 8192 | 3.31e+04 | 120.8% | 131.2% | 130.0% |
| | | 16384 | 3.33e+04 | 119.7% | 131.1% | 131.0% |
| | | 32768 | 3.27e+04 | 123.6% | 134.1% | 135.3% |
| | BitNet-2B | 512 | 1.56e+04 | 92.1% | 136.0% | 135.8% |
| | | 1024 | 1.74e+04 | 119.2% | 131.8% | 135.8% |
| | | 2048 | 1.88e+04 | 115.0% | 126.1% | 131.0% |
| | | 4096 | 1.76e+04 | 114.7% | 126.8% | 131.8% |
| | | 8192 | 1.73e+04 | 114.7% | 127.9% | 135.2% |
| | | 16384 | 1.75e+04 | 115.6% | 127.7% | 133.8% |
| | | 32768 | 1.75e+04 | 116.7% | 127.7% | 135.0% |
| | Llama3.2-3B | 512 | 1.39e+04 | 114.6% | 127.2% | 123.1% |
| | | 1024 | 1.53e+04 | 117.4% | 127.6% | 127.8% |
| | | 2048 | 1.57e+04 | 118.2% | 126.6% | 127.1% |
| | | 4096 | 1.49e+04 | 120.5% | 128.3% | 129.1% |
| | | 8192 | 1.47e+04 | 122.2% | 128.4% | 129.7% |
| | | 16384 | 1.48e+04 | 120.1% | 128.2% | 130.7% |
| | | 32768 | 1.15e+04 | 153.5% | 164.1% | 168.6% |
| | Qwen2.5-7B | 512 | 6.15e+03 | 115.1% | 128.2% | 125.2% |
| | | 1024 | 7.55e+03 | 115.0% | 120.7% | 121.7% |
| | | 2048 | 7.74e+03 | 114.1% | 120.9% | 124.3% |
| | | 4096 | 7.75e+03 | 110.9% | 118.2% | 122.6% |
| | | 8192 | 7.63e+03 | 112.2% | 113.7% | 120.8% |
| | | 16384 | 1.35e+03 | 643.5% | 691.3% | 90.4% |
| | | 32768 | 7.78e+03 | 111.3% | 118.4% | 121.5% |
| | Qwen2.5-14B | 512 | 3.84e+03 | 114.1% | 118.7% | 119.7% |
| | | 1024 | 4.42e+03 | 112.8% | 113.3% | 120.8% |
| | | 2048 | 4.38e+03 | 108.5% | 113.8% | 115.9% |
| | | 4096 | 4.42e+03 | 107.4% | 114.4% | 116.3% |
| | | 8192 | 4.35e+03 | 109.6% | 116.7% | 118.5% |
| | | 16384 | 4.40e+03 | 107.1% | 115.0% | 116.8% |
| | | 32768 | 1.09e+03 | 427.5% | 454.4% | 465.9% |

### Prefill (FP8)

| GPU | Model | Batchsize M | cuSPARSELt Throughput | Algorithmic Efficiency under Different Sparsity | | |
|---|---|---|---|---|---|---|
| | | | | 4:6 | 6:8 | 8:10 |
| RTX4090 | Llama3.2-1B | 512 | 8.72e+03 | 128.9% | 143.4% | 154.7% |
| | | 1024 | 1.67e+04 | 134.9% | 146.5% | 158.1% |
| | | 2048 | 3.36e+04 | 131.8% | 146.6% | 156.4% |
| | | 4096 | 6.52e+04 | 133.3% | 148.4% | 151.2% |
| | | 8192 | 1.04e+05 | 119.8% | 128.9% | 131.2% |
| | | 16384 | 1.07e+05 | 117.4% | 126.5% | 127.8% |
| | | 32768 | 1.09e+05 | 116.7% | 125.7% | 127.1% |
| | BitNet-2B | 512 | 8.43e+03 | 130.8% | 151.4% | 155.4% |
| | | 1024 | 1.65e+04 | 132.0% | 160.2% | 163.1% |
| | | 2048 | 3.34e+04 | 133.3% | 141.6% | 157.0% |
| | | 4096 | 6.13e+04 | 119.3% | 132.0% | 135.9% |
| | | 8192 | 6.33e+04 | 116.0% | 124.8% | 133.1% |
| | | 16384 | 6.39e+04 | 115.9% | 123.9% | 129.9% |
| | | 32768 | 6.40e+04 | 117.9% | 123.9% | 131.0% |
| | Llama3.2-3B | 512 | 7.37e+03 | 135.9% | 154.4% | 153.7% |
| | | 1024 | 1.50e+04 | 130.7% | 145.6% | 155.3% |
| | | 2048 | 3.01e+04 | 133.3% | 151.4% | 160.0% |
| | | 4096 | 4.77e+04 | 116.7% | 122.9% | 122.2% |
| | | 8192 | 4.68e+04 | 115.7% | 122.9% | 124.4% |

### Prefill (FP8) (cont.)

| GPU | Model | Batchsize M | cuSPARSELt Throughput | Algorithmic Efficiency under Different Sparsity | | |
|---|---|---|---|---|---|---|
| | | | | 4:6 | 6:8 | 8:10 |
| | | 16384 | 4.69e+04 | 115.6% | 121.7% | 124.2% |
| | | 32768 | 4.69e+04 | 115.6% | 121.7% | 124.2% |
| | Qwen2.5-7B | 512 | 8.43e+03 | 133.3% | 148.5% | 158.4% |
| | | 1024 | 1.67e+04 | 134.5% | 150.0% | 155.6% |
| | | 2048 | 2.36e+04 | 110.8% | 116.6% | 120.0% |
| | | 4096 | 2.48e+04 | 111.1% | 117.0% | 119.5% |
| | | 8192 | 2.48e+04 | 111.4% | 117.4% | 120.0% |
| | | 16384 | 2.49e+04 | 111.8% | 74.5% | 63.0% |
| | | 32768 | 8.28e+03 | 99.3% | 100.0% | 100.4% |
| | Qwen2.5-14B | 512 | 4.72e+03 | 137.3% | 156.0% | |
| | | 1024 | 9.90e+03 | 134.5% | 144.9% | |
| | | 2048 | 1.33e+04 | 111.0% | 116.1% | |
| | | 4096 | 1.34e+04 | 110.5% | 115.8% | |
| | | 8192 | 1.33e+04 | 109.6% | 116.6% | |
| | | 16384 | 1.32e+04 | 110.4% | | |
| | | 32768 | 1.32e+04 | | | |
| H100 | Llama3.2-1B | 512 | 1.63e+04 | 136.6% | 142.6% | 160.0% |
| | | 1024 | 3.24e+04 | 133.3% | 144.6% | 158.1% |
| | | 2048 | 6.16e+04 | 139.7% | 150.0% | 160.0% |
| | | 4096 | 1.14e+05 | 138.1% | 160.7% | 160.0% |
| | | 8192 | 1.71e+05 | 128.0% | 135.0% | 142.4% |
| | | 16384 | 2.14e+05 | 114.5% | 126.1% | 127.4% |
| | | 32768 | 2.23e+05 | 114.6% | 127.6% | 126.3% |
| | BitNet-2B | 512 | 1.61e+04 | 137.2% | 151.5% | 166.2% |
| | | 1024 | 3.26e+04 | 134.6% | 150.0% | 158.4% |
| | | 2048 | 6.24e+04 | 138.7% | 153.0% | 174.5% |
| | | 4096 | 9.64e+04 | 116.4% | 128.2% | 133.8% |
| | | 8192 | 1.08e+05 | 117.9% | 128.6% | 131.4% |
| | | 16384 | 1.13e+05 | 114.6% | 126.3% | 131.9% |
| | | 32768 | 1.18e+05 | 114.8% | 125.2% | 129.4% |
| | Llama3.2-3B | 512 | 1.73e+04 | 130.6% | 143.9% | 160.0% |
| | | 1024 | 3.54e+04 | 133.3% | 147.1% | 158.4% |
| | | 2048 | 6.72e+04 | 132.1% | 139.9% | 141.5% |
| | | 4096 | 8.46e+04 | 119.2% | 126.1% | 123.2% |
| | | 8192 | 9.19e+04 | 115.1% | 123.1% | 124.4% |
| | | 16384 | 9.57e+04 | 115.6% | 122.5% | 122.7% |
| | | 32768 | 9.66e+04 | 115.6% | 121.2% | 122.7% |
| | Qwen2.5-7B | 512 | 1.88e+04 | 128.1% | 148.5% | 161.6% |
| | | 1024 | 3.80e+04 | 117.9% | 125.9% | 131.4% |
| | | 2048 | 4.41e+04 | 115.7% | 127.7% | 126.9% |
| | | 4096 | 4.86e+04 | 112.9% | 119.8% | 120.0% |
| | | 8192 | 5.03e+04 | 111.3% | 116.9% | 118.4% |
| | | 16384 | 5.17e+04 | 110.8% | 117.7% | 119.4% |
| | | 32768 | 5.12e+04 | 110.1% | 116.7% | 119.4% |
| | Qwen2.5-14B | 512 | 1.22e+04 | 125.7% | 151.4% | 150.9% |
| | | 1024 | 2.18e+04 | 113.0% | 119.5% | 124.7% |
| | | 2048 | 2.46e+04 | 114.0% | 118.5% | 123.9% |
| | | 4096 | 2.63e+04 | 109.7% | 115.4% | 119.4% |
| | | 8192 | 2.70e+04 | 110.8% | 116.5% | 118.2% |
| | | 16384 | 2.74e+04 | 108.9% | 114.5% | 117.3% |
| | | 32768 | 2.75e+04 | 109.7% | 115.4% | 116.9% |
| B200 | Llama3.2-1B | 512 | 2.23e+04 | 131.9% | 141.7% | 156.4% |
| | | 1024 | 4.22e+04 | 134.9% | 150.0% | 161.8% |
| | | 2048 | 8.70e+04 | 121.5% | 145.5% | 155.2% |
| | | 4096 | 1.63e+05 | 134.8% | 148.3% | 163.6% |
| | | 8192 | 3.08e+05 | 136.3% | 153.3% | 154.7% |
| | | 16384 | 4.61e+05 | 132.0% | 151.5% | 158.4% |
| | | 32768 | 4.75e+05 | 132.0% | 148.5% | 160.0% |
| | BitNet-2B | 512 | 1.69e+04 | 132.0% | 150.0% | 160.0% |
| | | 1024 | 3.36e+04 | 133.3% | 148.5% | 158.4% |
| | | 2048 | 6.97e+04 | 134.6% | 145.7% | 150.8% |
| | | 4096 | 1.34e+05 | 133.3% | 148.5% | 160.0% |
| | | 8192 | 2.53e+05 | 128.2% | 139.9% | 150.8% |
| | | 16384 | 3.20e+05 | 115.2% | 126.8% | 132.4% |
| | | 32768 | 3.36e+05 | 115.6% | 127.4% | 131.7% |
| | Llama3.2-3B | 512 | 1.79e+04 | 130.7% | 145.5% | 153.7% |
| | | 1024 | 3.68e+04 | 125.5% | 144.1% | 160.0% |
| | | 2048 | 6.86e+04 | 133.3% | 150.0% | 161.6% |
| | | 4096 | 1.39e+05 | 134.7% | 153.0% | 163.2% |
| | | 8192 | 2.47e+05 | 118.8% | 129.5% | 128.0% |
| | | 16384 | 2.73e+05 | 118.3% | 126.5% | 126.6% |
| | | 32768 | 2.92e+05 | 115.6% | 123.7% | 122.7% |
| | Qwen2.5-7B | 512 | 1.96e+04 | 135.8% | 148.6% | 158.5% |
| | | 1024 | 4.08e+04 | 133.3% | 150.0% | 158.5% |
| | | 2048 | 7.46e+04 | 137.1% | 152.9% | 167.6% |
| | | 4096 | 1.39e+05 | 113.2% | 123.5% | 125.0% |
| | | 8192 | 1.51e+05 | 110.6% | 122.0% | 124.9% |
| | | 16384 | 1.61e+05 | 109.9% | 118.8% | 120.3% |

## Prefill (FP8) (cont.)

| GPU | Model | Batchsize M | cuSPARSELt Throughput | Algorithmic Efficiency under Different Sparsity | | |
|---|---|---|---|---|---|---|
| | | | | 4:6 | 6:8 | 8:10 |
| | | 32768 | 1.66e+05 | 110.1% | 117.9% | 120.6% |
| | Qwen2.5-14B | 512 | 1.19e+04 | 128.3% | 148.6% | 158.5% |
| | | 1024 | 2.40e+04 | 130.8% | 151.4% | 160.0% |
| | | 2048 | 4.94e+04 | 124.6% | 141.6% | 157.0% |
| | | 4096 | 7.73e+04 | 110.6% | 120.7% | 121.0% |
| | | 8192 | 8.00e+04 | 115.7% | 124.0% | 126.9% |
| | | 16384 | 8.57e+04 | 97.6% | 116.9% | 119.7% |
| | | 32768 | 8.82e+04 | 107.3% | 117.2% | 121.2% |
| RTX5080 | Llama3.2-1B | 512 | 3.65e+04 | 136.1% | 143.8% | 164.9% |
| | | 1024 | 7.03e+04 | 133.3% | 145.6% | 147.6% |
| | | 2048 | 9.44e+04 | 114.3% | 122.9% | 125.1% |
| | | 4096 | 9.65e+04 | 115.3% | 121.8% | 125.1% |
| | | 8192 | 9.58e+04 | 114.3% | 120.7% | 125.1% |
| | | 16384 | 9.42e+04 | 115.2% | 121.6% | 124.8% |
| | | 32768 | 9.36e+04 | 115.2% | 121.6% | 124.8% |
| | BitNet-2B | 512 | 2.89e+04 | 133.3% | 151.4% | 157.1% |
| | | 1024 | 4.59e+04 | 113.4% | 120.5% | 119.7% |
| | | 2048 | 4.96e+04 | 115.6% | 122.2% | 123.3% |
| | | 4096 | 5.06e+04 | 113.0% | 119.6% | 122.9% |
| | | 8192 | 4.82e+04 | 113.7% | 120.2% | 123.5% |
| | | 16384 | 4.73e+04 | 112.6% | 120.0% | 123.3% |
| | | 32768 | 4.71e+04 | 114.4% | 120.9% | 124.2% |
| | Llama3.2-3B | 512 | 2.94e+04 | 125.5% | 132.4% | 135.8% |
| | | 1024 | 3.75e+04 | 115.0% | 120.2% | 120.9% |
| | | 2048 | 4.17e+04 | 111.6% | 115.3% | 117.6% |
| | | 4096 | 4.04e+04 | 113.5% | 117.0% | 119.1% |
| | | 8192 | 3.92e+04 | 113.3% | 117.9% | 118.9% |
| | | 16384 | 3.87e+04 | 113.2% | 118.7% | 119.7% |
| | | 32768 | 3.87e+04 | 113.2% | 118.7% | 119.7% |
| | Qwen2.5-7B | 512 | 1.84e+04 | 110.8% | 116.2% | 117.2% |
| | | 1024 | 1.99e+04 | 107.2% | 113.5% | 114.6% |
| | | 2048 | 2.03e+04 | 108.8% | 112.5% | 113.7% |
| | | 4096 | 2.00e+04 | 111.6% | 114.3% | 115.4% |
| | | 8192 | 1.99e+04 | 109.9% | 113.5% | 114.6% |
| | | 16384 | 1.98e+04 | 110.7% | 114.3% | 115.4% |
| | | 32768 | 1.97e+04 | | | |
| | Qwen2.5-14B | 512 | | | | |
| | | 1024 | | | | |
| | | 2048 | | | | |
| | | 4096 | | | | |
| | | 8192 | | | | |
| | | 16384 | | | | |
| | | 32768 | | | | |
| GB10 | Llama3.2-1B | 512 | 2.96e+04 | 118.8% | 125.5% | 133.8% |
| | | 1024 | 3.19e+04 | 119.2% | 129.8% | 132.3% |
| | | 2048 | 3.42e+04 | 118.2% | 124.5% | 129.8% |
| | | 4096 | 3.39e+04 | 117.0% | 128.8% | 128.3% |
| | | 8192 | 3.28e+04 | 119.4% | 130.0% | 132.6% |
| | | 16384 | 3.30e+04 | 122.0% | 131.6% | 134.3% |
| | | 32768 | 3.40e+04 | 122.0% | 130.2% | 134.3% |
| | BitNet-2B | 512 | 1.50e+04 | 125.9% | 140.3% | 140.7% |
| | | 1024 | 1.72e+04 | 119.6% | 130.4% | 136.1% |
| | | 2048 | 1.83e+04 | 115.9% | 124.8% | 128.6% |
| | | 4096 | 1.71e+04 | 116.5% | 126.7% | 132.0% |
| | | 8192 | 1.73e+04 | 115.9% | 126.2% | 130.1% |
| | | 16384 | 1.75e+04 | 116.0% | 126.4% | 128.9% |
| | | 32768 | 1.72e+04 | 118.1% | 125.7% | 134.1% |
| | Llama3.2-3B | 512 | 1.39e+04 | 116.1% | 129.3% | 132.4% |
| | | 1024 | 1.53e+04 | 116.4% | 124.1% | 125.1% |
| | | 2048 | 1.51e+04 | 119.8% | 126.4% | 127.4% |
| | | 4096 | 1.44e+04 | 119.5% | 125.9% | 128.3% |
| | | 8192 | 1.45e+04 | 118.5% | 122.2% | 128.9% |
| | | 16384 | 1.47e+04 | 118.5% | 125.0% | 128.9% |
| | | 32768 | 1.46e+04 | 118.5% | 120.8% | 130.4% |
| | Qwen2.5-7B | 512 | 6.35e+03 | 124.8% | 133.5% | 132.1% |
| | | 1024 | 7.61e+03 | 108.3% | 117.9% | 121.4% |
| | | 2048 | 7.59e+03 | 109.1% | 117.3% | 120.7% |
| | | 4096 | 7.34e+03 | 110.3% | 119.7% | 121.5% |
| | | 8192 | 7.44e+03 | 110.3% | 120.0% | 123.6% |
| | | 16384 | 7.51e+03 | 111.5% | 120.0% | 123.6% |
| | | 32768 | 7.63e+03 | 110.7% | 117.9% | 121.4% |
| | Qwen2.5-14B | 512 | 4.10e+03 | 110.6% | 119.8% | 120.3% |
| | | 1024 | 4.33e+03 | 106.7% | 112.2% | 115.5% |
| | | 2048 | 4.18e+03 | 107.6% | 115.8% | 117.9% |
| | | 4096 | 4.27e+03 | 109.0% | 113.5% | 115.5% |
| | | 8192 | 4.21e+03 | 110.9% | 116.8% | 118.9% |
| | | 16384 | 1.13e+03 | 411.7% | 434.0% | 433.4% |
| | | 32768 | 4.15e+03 | 110.7% | 113.8% | 117.1% |

**End-to-End Efficiency Summary.** Across all GPUs and models, end-to-end prefill efficiency: (1) starts high (120–165%) at small $M$ due to cuSPARSELt 2:4 baseline overhead; (2) converges to 100–120% at large $M$ ($\geq 4096$); (3) rarely drops below 100%, validating SlideSparse's approach. The consistent $> 100\%$ efficiency at production-relevant $M$ values confirms that SlideSparse not only matches but often exceeds the performance predicted by theoretical analysis.

For example, A100 Qwen-2.5-14B at $M{=}32768$ achieves 107–114% efficiency across 4:6, 6:8, and 8:10 sparsity patterns. At smaller $M = 512$, efficiency rises to 135–165%, demonstrating that SlideSparse's overhead structure is more favorable than native 2:4 at these scales.

**Key Finding: SlideSparse Unlocks Hidden Performance.** The consistent $> 100\%$ efficiency at large $M$ reveals that SlideSparse not only preserves 2:4 speedup but *amplifies* it. This bonus stems from our fused quantization-slide kernel (§4): by performing activation lifting *within* the quantization pass, we eliminate a memory round-trip that a naive two-stage approach would incur. SlideSparse unlocks performance that even native 2:4 workflows leave on the table.

**Implications for Baseline Quality Assessment.** The efficiency metric also serves as a diagnostic tool for baseline optimization quality. When efficiency systematically exceeds 150% (as observed on B200 INT8 at small $M$), it indicates that the native cuSPARSELt 2:4 implementation provides *no actual speedup* at those configurations. Practitioners should interpret raw speedup numbers cautiously on newer architectures until driver optimization matures.

### D.7. Discussion and Edge Cases

#### D.7.1. GPU ARCHITECTURE PATTERNS

Our comprehensive evaluation reveals distinct behavior across GPU architectures:

- **A100 (Ampere):** Most consistent and predictable behavior. Best choice for benchmarking methodology validation due to mature cuBLASLt/cuSPARSELt optimization.

- **H100 (Hopper):** Strong performance but higher sparse overhead at small $M$. Excels at large-scale prefill inference.

- **B200 (Blackwell):** Exceptional INT8 speedups due to suboptimal cuBLASLt baseline—expect normalization in future drivers.

- **RTX 4090 (Ada Lovelace):** Consumer GPU achieving near-datacenter performance for 2:4–8:10 configurations. Shows API anomalies at higher-density configurations.

- **RTX 5080 (Blackwell Consumer):** Excellent price-performance for sparse inference. Validates SlideSparse on consumer hardware.

- **GB10 (Embedded):** Variable performance with some configurations limited by driver maturity. Requires eager mode execution (no `torch.compile`).

#### D.7.2. CUSPARSELT LIMITATIONS

Through extensive testing, we identified several cuSPARSELt limitations:

**Small-$M$ Overhead.** At $M < 256$, cuSPARSELt's sparse format overhead (metadata packing, tensor core scheduling) often exceeds computation savings, yielding speedup $< 1.0\times$. This is a fundamental limitation of structured sparsity hardware, not specific to SlideSparse.

**Precision Support Gaps.** H100 shows missing data for FP16 sparse configurations due to API limitations. FP4 exhibits "illegal address" and "illegal instruction" errors for certain matrix dimensions on multiple GPUs.

**Algorithm Instability.** RTX 4090 shows highly irregular behavior at higher-density configurations (8:10+) with 0.10–0.27$\times$ speedups at certain $M$ values—likely API implementation issues rather than fundamental limitations.

### D.7.3. KNOWN BENCHMARK COVERAGE GAPS

| Issue | Affected Configuration | Root Cause |
| --- | --- | --- |
| Triton index overflow | $M=65536$, Qwen-7B | INT32 indexing limit |
| Out of Memory | 7B/14B on RTX 4090/5080 | Insufficient VRAM |
| FP4 illegal address | Specific $(M, N, K)$ | cuBLASLt/cuSPARSELt API |
| FP16 illegal instruction | H100 square sparse | API implementation bug |

### D.7.4. PRACTICAL RECOMMENDATIONS

Based on our comprehensive evaluation:

1. **For long-context prefill** ($M \geq 4096$)**:** SlideSparse achieves near-theoretical speedups. Use 2:4 for maximum acceleration; 4:6/6:8 for accuracy-speedup trade-offs.

2. **For high-concurrency decode** ($M \geq 256$)**:** Modest but consistent speedups ($1.1$–$1.4\times$). Benefits scale with model size.

3. **For consumer GPUs:** RTX 4090/5080 achieve 80–95% of datacenter GPU efficiency. SlideSparse democratizes sparse inference.

4. **For INT8 on B200:** Despite high apparent speedups, efficiency analysis reveals these come from baseline weakness. Expect normalization in future drivers.

### D.7.5. FUTURE DIRECTIONS

**Accuracy-Aware Sparse Training.**   Current experiments use magnitude pruning for demonstration. Future work includes sparsity-aware fine-tuning to recover accuracy at higher sparsity levels. Based on existing literature, 4:6 sparsity can maintain satisfactory quality with proper training.

**M:N Hardware Evolution.**   Our generalized theory (Appendix C.1) provides the mathematical foundation for future hardware. If NVIDIA introduces 1:4 Sparse Tensor Cores ($\alpha = 4\times$), SlideSparse's sliding window approach extends naturally—achieving the density-determined speedup bound $S_{\text{eff}} = L/Z$ universally.

**Dynamic Sparsity Adaptation.**   Current SlideSparse uses fixed sparsity patterns determined offline. Future work could explore layer-wise or even token-wise dynamic sparsity selection based on activation statistics.

### D.7.6. PRECISION TYPE IMPACT

Our evaluation reveals distinct characteristics across precision types:

| Precision | Strengths | Limitations |
| --- | --- | --- |
| INT8 | Highest speedups on A100/B200; most mature implementation; best for production inference | B200 shows inflated numbers due to suboptimal baseline |
| FP8 | Consistent cross-platform behavior; emerging standard for quantized inference | 5–15% lower speedups than INT8 |
| BF16 | Most stable and predictable; robust across configurations | Lower speedups than quantized formats |
| FP16 | Similar characteristics to BF16 | H100 shows API limitations (missing sparse data) |

### D.7.7. CROSS-DIMENSIONAL ANALYSIS SUMMARY

**Key Scaling Laws.**   Our experiments reveal consistent scaling behavior across all configurations:

- **M Dimension:** $M < 256$: overhead dominates; $M \geq 1024$: speedups stabilize; $M \geq 4096$: near-theoretical performance

- **Model Size:** Larger models achieve higher speedups due to better GEMM utilization and amortized overhead

- **Sparsity Level:** Speedup scales inversely with density, closely matching theoretical predictions ($S_{\text{eff}} \approx N/(N-1)$)

- **Workload Type:** Prefill (compute-bound) shows 25–35% higher speedups than decode (memory-bound)

**Optimal Deployment Scenarios.**    SlideSparse is optimally suited for: (1) long-context prefill workloads where GEMM dominates inference time; (2) large models (7B+ parameters) that maximize sparse tensor core utilization; (3) INT8/FP8 quantized inference on A100/H100/B200 platforms.

### D.8. Summary of Benchmark Data

For transparency and reproducibility, all raw benchmark results are organized as follows:

**Performance vs. Dense Baseline (cuBLASLt).**    These tables report speedup relative to dense cuBLASLt GEMM:

- **Table A** (§D.3.1): Square kernel speedup ($M=N=K$) across 5 precisions — `appendix_a_cuBLASLt.pdf`

- **Table B** (§D.3.2): Model-specific kernel speedup — `appendix_b_cuBLASLt.pdf`

- **Table C** (§D.5.1): End-to-end prefill throughput — `appendix_c_cuBLASLt.pdf`

- **Table D** (§D.5.2): End-to-end decode throughput — `appendix_d_cuBLASLt.pdf`

**Algorithmic Efficiency vs. 2:4 Baseline (cuSPARSELt).**    These tables report efficiency normalized against cuSPARSELt's native 2:4 implementation:

- **Efficiency A** (§D.6.3): Square kernel efficiency — `appendix_a_cuSPARSELt.pdf`

- **Efficiency B**: Model-specific kernel efficiency — `appendix_b_cuSPARSELt.pdf`

- **Efficiency C** (§D.6.3): End-to-end prefill efficiency — `appendix_c_cuSPARSELt.pdf`

- **Efficiency D**: End-to-end decode efficiency — `appendix_d_cuSPARSELt.pdf`

All CSV source files and table generation scripts are available in the repository.

## E. Reproducibility

This section provides the hardware, software, and model details necessary to reproduce all experiments in this paper.

### E.1. Hardware Configuration

Table 5 summarizes the GPU platforms used in our evaluation, spanning four architecture generations and three platform types.

*Table 5.* Hardware specifications for all evaluated GPUs.

| GPU | Arch. | CC | Memory | Platform | Host Arch. |
|---|---|---|---|---|---|
| A100 80GB PCIe | Ampere | sm80 | 80GB HBM2e | Datacenter | x86_64 |
| H100 80GB PCIe | Hopper | sm90 | 80GB HBM3 | Datacenter | x86_64 |
| B200 180GB SXM | Blackwell | sm100 | 180GB HBM3e | Datacenter | x86_64 |
| RTX 4090 | Ada Lovelace | sm89 | 24GB GDDR6X | Consumer | x86_64 |
| RTX 5080 | Blackwell | sm120 | 16GB GDDR7 | Consumer | x86_64 |
| DGX Spark (GB10) | Blackwell | sm121 | 128GB Unified | Embedded | aarch64 |

All GPUs support 2:4 structured sparsity via Sparse Tensor Cores (Mishra et al., 2021). The evaluation covers datacenter (A100, H100, B200), consumer (RTX 4090, RTX 5080), and embedded (DGX Spark) platforms across both x86_64 and aarch64 host architectures.

## E.2. Software Environment

All experiments are conducted using a unified software stack based on the official vLLM Docker image:

- **Base Image:** `vllm/vllm-openai:v0.13.0`

- **Operating System:** Ubuntu 22.04/24.04 LTS

- **CUDA Toolkit:** 12.9

- **cuSPARSELt:** 0.8.1

- **PyTorch:** 2.9.0

- **vLLM:** 0.13.0

- **Python:** 3.12

We provide Docker images for both x86_64 and aarch64 architectures; the concrete image tags are listed in §E.5.

## E.3. Benchmark Configuration

**Kernel Benchmarks.** Kernel-level benchmarks measure raw GEMM performance in isolation. Each configuration is executed with 25 warmup iterations followed by 100 measurement runs; we report mean latency. The $M$ dimension (representing batch size $\times$ sequence length) is varied across $\{64, 128, 256, 512, 1024, 2048, 4096, 8192, 16384\}$ to capture performance scaling.

**Accuracy Benchmarks.** Accuracy evaluation follows the exact `lm-eval-harness` runs used to produce Figure 2 and the accuracy tables in Appendix D.4. We evaluate Qwen2.5-7B in FP16 and Qwen2.5-14B in BF16 on 9 tasks: PIQA, ARC-Easy, ARC-Challenge, HellaSwag, WinoGrande, BoolQ, OpenBookQA, MMLU (Hendrycks et al., 2021), and GSM8K (Cobbe et al., 2021). The seven commonsense tasks are 0-shot; MMLU and GSM8K are 5-shot. For aggregation, we use the same harness metrics recorded in the JSON outputs: `acc_norm` for PIQA, ARC-Easy, ARC-Challenge, HellaSwag, and OpenBookQA; `acc` for BoolQ and WinoGrande; `acc` for MMLU; and GSM8K exact match under the strict-match extractor. Sparse checkpoints are produced with one-shot Wanda or SparseGPT; Wanda uses no calibration set, while SparseGPT uses 128 WikiText-2 calibration samples with sequence length 2048. Neither method uses sparse-aware fine-tuning or post-pruning recovery training. All accuracy runs use `lm-eval-harness` v0.4.11 on H100 80GB PCIe GPUs.

**End-to-End Benchmarks.** End-to-end experiments use the `vllm bench throughput` methodology with the following configurations:

- **Prefill (compute-bound):** $M \in \{512, 1024, 2048, 4096, 8192, 16384, 32768, 65536\}$, where $M = $ `max_num_seqs` $\times$ `prompt_len`. We set `output_len=1` to minimize decode overhead and run $N = 128$ iterations.

- **Decode (memory-bound):** $M \in \{64, 128, 256, 512\}$, where $M = $ `max_num_seqs`. We use single-token prompts to minimize prefill overhead and run $N = 256$ decode iterations per request.

- **Combined serving latency:** for request-level measurements we use a 4096-token prompt on Qwen2.5-14B INT8 (A100), and report both time-to-first-token and total latency for 128- and 512-token generation.

## E.4. Model Sources

All models are obtained from HuggingFace Hub. We use quantized checkpoints provided by Red Hat for INT8 and FP8 experiments:

**INT8 (W8A8):**

- `RedHatAI/Llama-3.2-1B-Instruct-quantized.w8a8`

- `RedHatAI/Llama-3.2-3B-Instruct-quantized.w8a8`

- `RedHatAI/Qwen2.5-7B-Instruct-quantized.w8a8`

- `RedHatAI/Qwen2.5-14B-Instruct-quantized.w8a8`

**FP8 (Dynamic):**

- `RedHatAI/Llama-3.2-1B-Instruct-FP8-dynamic`

- `RedHatAI/Llama-3.2-3B-Instruct-FP8-dynamic`

- `RedHatAI/Qwen2.5-7B-Instruct-FP8-dynamic`

- `RedHatAI/Qwen2.5-14B-Instruct-FP8-dynamic`

**BitNet:**

- `microsoft/bitnet-b1.58-2B-4T-BF16`

For accuracy evaluation, we use the one-shot checkpoints underlying Figure 2: Qwen2.5-7B is evaluated in FP16 and Qwen2.5-14B in BF16, with dense baselines re-run under the same precision as each model. For throughput benchmarking, we use pre-converted sparse checkpoints produced offline at the target sparsity pattern; SlideSparse's measured speedup is agnostic to the upstream pruning recipe. Pre-converted sparse checkpoints (for speedup benchmarking) are available at:

https://huggingface.co/bcacdwk/slidesparse-checkpoints

Note: these checkpoints are pruned for throughput evaluation only.

### E.5. Code and Environment Availability

Our implementation, including the fused quantization-slide Triton kernels, offline weight packer, cuSPARSELt GEMM wrapper, and vLLM integration, is open-sourced at:

https://github.com/bcacdwk/vllmbench

To facilitate reproducibility, we provide pre-built Docker images with all dependencies, hosted at https://hub.docker.com/r/bcacdwk/vllmbench:

```
docker pull bcacdwk/vllmbench:0.13.0_cu129_amd64
docker pull bcacdwk/vllmbench:0.13.0_cu129_arm64
```

Both x86_64 (amd64) and ARM64 architectures are supported.

