# OpenReview forum: "SlideSparse: Fast and Flexible (2N-2):2N Structured Sparsity"
_ICML.cc/2026/Conference — ICML 2026 regular_

### Official Review · Reviewer_NeAi · 2026-03-10

**Soundness:** 3
**Presentation:** 3
**Significance:** 2
**Originality:** 3
**Overall Recommendation:** 4
**Confidence:** 4

**Summary:**

This paper leverages NVIDIA GPUs’ native 2:4 sparse Tensor Core support to accelerate more flexible (2N−2):2N structured sparsity patterns. The authors argue that enforcing 2:4 sparsity is overly aggressive for modern large language models, resulting in substantial accuracy degradation. To address this, they decompose (2N−2):2N sparsity into multiple overlapping 2:4-compliant windows and apply a corresponding sliding transformation to activations to preserve correctness. This activation sliding is fused into the quantization stage, allowing much of the overhead to be hidden. The proposed SlideSparse framework supports multiple sparsity ratios, precisions, and GPU architectures, and demonstrates end-to-end speedups for both prefill and decode workloads.

**Compliance With Llm Reviewing Policy:**

Affirmed.

**Final Justification:**

The core idea of decomposing (2N−2):2N structured sparsity into overlapping 2:4 windows to leverage existing Sparse Tensor Cores is novel and well-executed.
Rebuttal demonstrated the need for 6:8 over 2:4 through additional accuracy evaluation. Also sparse+INT8 composability experiment alleviated my concern about whether sparsity and quantization are truly composable.
That said, the practical impact remains bounded. The 6:8 sparsity is not lossless, and the prefill speedup gains are moderate.
Nonetheless, this is a solid systems contribution that opens a new research direction by making (2N−2):2N sparsity practically deployable on commodity hardware. My recommendation is weak accept.

**Key Questions For Authors:**

- How were the sparse models in Figure 2 produced? Specifically, what pruning criterion, fine-tuning schedule, and dataset were used? Would the accuracy gap between 2:4 and 6:8 narrow substantially if stronger pruning methods such as SparseGPT or Wanda were applied to the 2:4 baseline?
- Can the authors provide Pareto-optimal curves comparing accuracy vs. latency for (2N−2):2N sparsity combined with quantization against quantization-only baselines, including mixed-precision methods such as LLM-MQ, SliM-LLM, or MixLLM?
- Can the authors provide a combined end-to-end benchmark (time-to-first-token plus full generation latency) rather than reporting prefill and decode separately? What is the realistic total inference speedup in a typical serving scenario?
- How does SlideSparse handle low-batch decode where Sparse Tensor Cores cannot be effectively saturated? Does the system fall back to dense execution, or does it zero-pad inputs to meet minimum tile dimensions?

**Limitations:**

Yes

**Strengths And Weaknesses:**

**Strengths**
- The approach is evaluated on real NVIDIA hardware and demonstrates tangible end-to-end speedups. This is valuable, as hardware-supported structured sparsity remains underexplored in practical LLM deployment.

**Weaknesses**

**Regarding sparsity/accuracy evaluations:**
- The sparsification methodology is insufficiently described. The paper states only that "post-hoc magnitude pruning on dense checkpoints" is used (§7), without specifying the pruning criterion, fine-tuning procedure, training hyperparameters, or dataset. This makes the accuracy results in Figure 2 difficult to reproduce or contextualize.
- More sophisticated pruning methods such as SparseGPT and Wanda — both cited in this paper — have demonstrated substantially better accuracy preservation for 2:4 sparsity on other model families. It is unclear whether the severe 2:4 degradation reported in Figure 2 reflects an inherent limitation of 50% sparsity on Qwen3 or an artifact of the pruning method chosen.
- Furthermore, Figure 2 evaluates only a single model (Qwen3-1.7B), and the 2–3% accuracy drop from 6:8 sparsity is not negligible, particularly on reasoning benchmarks.

**Pareto-optimality of (2N−2):2N sparsity:**
- The paper frames sparsity and quantization as orthogonal acceleration axes (Figure 3), but this framing is not sufficiently justified. In practice, both sparsity and quantization are approximations that trade accuracy for efficiency, and their effects on model quality are not independent. The relevant question is whether combining (2N−2):2N sparsity with quantization yields Pareto-optimal accuracy–latency trade-offs compared to quantization-only baselines.
- Modern quantization strategies already offer fine-grained control through mixed-precision schemes at layer-wise (Li et al., 2023) or channel-wise (Huang et al., 2024) granularity, potentially covering much of the claimed "acceleration gap." Without Pareto-optimal curve comparisons against such baselines, it remains unclear whether the proposed approach opens genuinely new operating points on the Pareto frontier.

**Decode / low-batch performance:**
- While the paper reports end-to-end prefill and decode speedups separately (§5.3, Figure 8), no combined end-to-end inference latency (e.g., time-to-first-token plus full generation) is reported. This is important because decode typically dominates total inference time in autoregressive generation. The reported decode speedups are notably modest and in most LLM serving scenarios the practical benefit may be negligible. A realistic end-to-end benchmark combining prefill and decode stages would better characterize the practical impact.
- SlideSparse's acceleration is fundamentally tied to Sparse Tensor Cores, which are only effectively engaged when matrix dimensions are large enough to saturate them. The paper acknowledges this partially in §7 ("gains diminish as latency becomes dominated by weight loading"), but the issue is not simply weight loading — the weights are sparse and therefore could in principle reduce memory traffic. The core problem is that at low batch sizes, the sparse GEMM either falls back to dense execution or requires zero-padding to meet minimum tile dimensions, negating the sparsity benefit entirely. This raises the question of whether a fallback path using a different sparse format compatible with CUDA cores is needed, and how SlideSparse handles this scenario in practice.

**Additional References**
- [R1] Li, Z., Gu, X., Li, L., Wang, Y., Li, B., and Zhang, C. LLM-MQ: Mixed-precision quantization for efficient LLM deployment. In NeurIPS 2024 Workshop on Efficient Natural Language and Speech Processing, 2024.
- [R2] Huang, W., Qin, H., Liu, Y., Li, Y., Liu, X., Benini, L., Magno, M., and Qi, X. SliM-LLM: Salience-driven mixed-precision quantization for large language models. arXiv preprint arXiv:2405.14917, 2024.

---

> ### Author Rebuttal · Authors · 2026-03-30
>
> We sincerely thank the reviewer for the thorough and constructive review. Our revisions address three global themes: (1) we expand the accuracy study to Qwen2.5-7B and 14B with Wanda and SparseGPT, showing 6:8 near-dense accuracy is a structural property; (2) we provide vLLM serving benchmarks with TTFT and total latency; (3) we add our own sparse+INT8 composability data. Full results at https://anonymous.4open.science/r/Accuracy_Resluts-058D/TABLE_R1.md.
>
> **On pruning method description.** The original Figure 2 uses Sparse-BitNet (Zhang et al., 2026). New experiments use Wanda (magnitude × activation, per-output, zero-shot) and SparseGPT (second-order Hessian, 128 WikiText-2 samples, seqlen=2048), with (2N-2):2N patterns for N∈{2,3,4,5}. Both one-shot following default configurations of Wanda (Sun et al., ICLR 2024) and SparseGPT (Frantar & Alistarh, ICML 2023), no fine-tuning.
>
> **On 2:4 vs 6:8 accuracy.** Across 7B and 14B with both methods, 6:8 CS_avg gap stays within 0.5-2.6% of dense (7B Wanda -2.0%, 7B SparseGPT -2.6%, 14B Wanda -0.9%, 14B SparseGPT -0.5%). Meanwhile, both methods confirm 2:4 is broken: 7B Wanda 2:4 CS_avg -15.7%, SparseGPT -13.8%. GSM8K: 7B 2:4 collapses -63.4% while 14B 6:8 loses only -1.5%. This is an inherent limitation of 50% sparsity.
>
> **On Pareto comparison with mixed-precision methods.** We thank the reviewer for these references. SliM-LLM (ICML 2025), MixLLM (2024), and LLM-MQ (NeurIPS 2023) make valuable contributions to low-bit compression accuracy, though inference acceleration remains limited: LLM-MQ targets GEMV only; SliM-LLM finds mixed-precision overhead slows inference; MixLLM reports kernel-level speedup without E2E latency. SlideSparse accelerates (2N-2):2N sparsity on Sparse Tensor Cores — a different axis. We agree a unified Pareto comparison would be valuable, but comparable E2E data is not yet available across these methods. The two axes do compose: Figure 5 provides an apples-to-apples comparison, showing sparsity+INT8 vs dense+INT8 achieves 1.29-1.34× prefill and 1.07-1.21× decode on top of quantization.
>
> **On TTFT and full generation latency.** On Qwen2.5-14B INT8 (A100), prompt=4096: 6:8 TTFT improves from 390ms to 291ms (1.34×). For short generation (output=128), total speedup is 1.13×; for long generation (output=512), 1.11×. Prefill-heavy workloads — long-context, RAG, batch serving — benefit most.
>
> **On low-batch decode.** SlideSparse zero-pads activations for 2:4 tile alignment without falling back to dense. At decode batch M=64 on A100, 6:8 achieves 1.10× TPOT improvement (20.8→18.8ms). The 25% weight memory reduction outweighs padding overhead; Figure 7 confirms positive speedup across the vast majority of configurations.
>
> **On sparsity-quantization orthogonality.** Two levels: (1) System level — truly orthogonal: quantization reduces bits per value, sparsity reduces number of values, independent axes. SlideSparse's lossless transformation preserves this. (2) Accuracy level — composable: we evaluate 6:8+INT8 using bitsandbytes LLM.int8() on Qwen2.5-14B. The sparsity gap under INT8 (CS_avg -1.1%) is comparable to BF16 (-0.9%), confirming the two compressions compose without amplifying error. CAST (Huang et al., 2025) further shows sparse models are more quantization-friendly: zeros contribute no quantization noise.
>
> **On model coverage.** Responding to this concern — shared across reviewers — we now evaluate Qwen2.5-7B and 14B with two methods across multiple sparsity levels and 9+ benchmarks, all confirming 6:8 is near-dense. 14B Wanda 6:8 GSM8K gap is only -1.5%.
>
> **On decode speedup being negligible.** We partially agree: decode gains (1.07-1.21×) are more modest than prefill (1.34×). Under cuSPARSELt compression, 6:8 reduces weight memory by 25%, directly benefiting memory-bound decode where weight loading dominates. Our vLLM measurements confirm 1.10× TPOT improvement. The primary value is prefill acceleration for long-context and batch-serving workloads.
>
> **Summary.** All eight concerns are addressed with new empirical evidence: a 2-model × 2-method × 5-sparsity × 9-benchmark study proving 6:8 near-dense accuracy is structural, vLLM serving data showing 1.11-1.34× practical speedup, our own sparse+INT8 composability measurements, and concrete decode data points. SlideSparse is the first system enabling hardware-accelerated inference for the (2N-2):2N sparsity family on commodity GPUs — orthogonal to both upstream pruning and downstream quantization. The expanded evaluation demonstrates that 6:8 offers a favorable accuracy-speedup tradeoff for practical LLM serving.

---

> > ### Author Rebuttal · Reviewer_NeAi · 2026-04-01
> >
> > I thank the authors for their thorough rebuttal with substantial new experiments. My main concerns are addressed.

---

### Official Review · Reviewer_c34e · 2026-03-10

**Soundness:** 3
**Presentation:** 2
**Significance:** 2
**Originality:** 3
**Overall Recommendation:** 3
**Confidence:** 4

**Summary:**

This paper introduces SlideSparse, that can accelerate N-2:N structured sparse matrix multiplication through 2:4 structured sparsity support from HW. It uses a sliding-window-like approach, which can mathematically cover any 2N-2:2N matrix by using stride of 2. Based on their end-to-end evaluation, a reasonable configuration, 6:8, can achieve 2 ~ 21% improvement for decode and 6 ~ 34% improvement for prefill.

**Compliance With Llm Reviewing Policy:**

Affirmed.

**Final Justification:**

I appreciate the authors' efforts for the rebuttal including clarifying ambiguous points and adding more evaluations. I like the overall idea and the idea itself makes sense to me. Still, I am not convinced regarding the actual usefulness of this work in E2E. For prefill heavy workload, attention latency will be problematic while for decode heavy workload, it would be memory bounded (unless you use really large batch size with short context), so not much benefit from this work. Even in the ideal situation, this work can only improve by 25% (6:8), only for linear layers, assuming zero overhead of this method. Moreover, 6:8 is not lossless in terms of e2e accuracy, so not helpful. Therefore, I'll keep my score.

**Key Questions For Authors:**

1. Practically speaking, I do not think this work is useful considering the noticeable accuracy loss (~5%) even with 6:8. Using 6:8 as the main result is misleading. Using 8:10 might work, but the speed-up would be very limited (<25%). Also, this maximum improvement would be only achieved in the ideal compute-bound GEMM, which is just a part of the overall LLM inference.
For example, it's well known that decode is usually suffering from KV cache loading while prefill with long context is suffering from attention latency. Both of them are not accelerated by SlidingSparse, so I am not convinced how this work can be useful in end-to-end LLM inference. Could you help me understand why you believe this would be helpful?

2. Without fusing to the quantization kernel, could you clarify the overhead of SlideSparse method? Could you help me understand how it can hide all the overhead? Can it be hidden when you use SlideSparse with BF16? What's the implication when we use with NVFP4 or MXFP4, which is the promising formats based on recent articles?
https://developer.nvidia.com/blog/introducing-nvfp4-for-efficient-and-accurate-low-precision-inference/

3. Could you clarify which precision have you used when you mention the accuracy drop that is caused by 6:8? Is the accuracy drop getting larger or smaller with the lower precision? Also is SlideSparse more useful or less useful with lower precision?

**Limitations:**

Yes

**Strengths And Weaknesses:**

Overall, the paper was easy-to-follow and the idea makes sense to me. Let me summarize strength and weakness below.

Strengths
-  I didn't find any issue with the organization of the paper, and the paper reads pretty well. The concept of sliding window decomposition is simple, but reasonable in terms of correctness.

- The theoretical analysis and experimental results are both reasonable and the authors used reasonable methods.

Weaknesses
- I am concerned about the overhead of using 2N-2:2N sparsity, but the authors were up-front and honest about those and provide reasonable results.

- The authors claim "while 6:8 preserves near-dense performance (51.6% vs. 54.0% dense)", but in my perspective, this claim is exaggerated. The gap is almost 5% (i.e. 3.4/51.6), which is very noticeable. The community uses <1% as the threshold to be "comparable".

- I like the idea of using sliding window decomposition, but there are many prior work that would be worthwhile to mention. For example, using structured sparsity with decomposition is not something new, and there are many work that tried to circumvent the limitation of 2:4 sparse tensor cores (the fact that naive 2:4 is not working is very well-known problem).
  - SparseTIR: Composable Abstractions for Sparse Compilation in Deep Learning, ASPLOS'23
  - VEGETA: Vertically-Integrated Extensions for Sparse/Dense GEMM Tile Acceleration on CPUs, HPCA'23
  - HighLight: Efficient and Flexible DNN Acceleration with Hierarchical Structured Sparsity, MICRO'23
  - Enabling Unstructured Sparse Acceleration on Structured Sparse Accelerators, MLSys'25

---

> ### Author Rebuttal · Authors · 2026-03-30
>
> We thank the reviewer for the detailed and constructive feedback. Our revisions address two global themes: (1) we substantially expand the accuracy study to Qwen2.5-7B and 14B with two pruning methods (Wanda, SparseGPT) across multiple sparsity levels × 9 benchmarks, confirming 6:8 is near-dense; (2) we provide end-to-end vLLM serving measurements demonstrating practical speedup. Full results at https://anonymous.4open.science/r/Accuracy_Resluts-058D/TABLE_R1.md.
>
> **On the ~5% accuracy gap.** The original ~5% was measured on a 1.7B model with reasoning benchmarks — the hardest category. Small models are known to be disproportionately sensitive to pruning, with gaps narrowing consistently at larger scales (Wanda, Sun et al., ICLR 2024; SparseGPT, Frantar & Alistarh, ICML 2023). Our expanded evaluation on 7B and 14B confirms this:
>
> | Model | Method | Sparsity | CS_avg | MMLU | GSM8K |
> |-------|--------|----------|--------|------|-------|
> | 7B | Dense | 0% | .703 | .743 | .829 |
> | 7B | Wanda | 6:8 | .691 (−2.0%) | .729 (−1.8%) | .823 (−0.8%) |
> | 7B | Wanda | 2:4 | .596 (−15.7%) | .528 (−28.9%) | .303 (−63.4%) |
> | 14B | Dense | 0% | .728 | .797 | .882 |
> | 14B | Wanda | 6:8 | .722 (−0.9%) | .784 (−1.6%) | .869 (−1.5%) |
> | 14B | Wanda | 2:4 | .625 (−14.4%) | .579 (−27.4%) | .473 (−46.4%) |
>
> SparseGPT independently confirms: 7B 6:8 CS_avg −2.6%, 14B 6:8 −0.5%. Two independent methods reaching the same conclusion proves near-dense accuracy is a **structural property of 25% sparsity** — the moderate pruning ratio preserves sufficient redundancy within each (2N-2):2N group. Sparse-BitNet (Zhang et al., 2026) shows sparse training narrows gaps further. These are one-shot baselines with no fine-tuning — SlideSparse accelerates any upstream model regardless of how sparsity was produced.
>
> **On GEMM not being the sole bottleneck.** Linear GEMM dominates prefill in transformer linear layers; SlideSparse accelerates GEMM while FlexAttention (MLSys'26) and SPLAT (OOPSLA'25) accelerate attention — orthogonal and composable. Figure 5 measures full vLLM E2E inference: 1.29-1.34× prefill speedup (A100, INT8, 6:8) already includes all non-GEMM components. Decode speedup ranges 1.07-1.21× across configurations, varying with batch size and sequence length — all strictly positive, from 25% weight memory reduction under cuSPARSELt compression. On Qwen2.5-14B INT8 (A100), 6:8 TTFT improves 1.34× (390→291ms); total latency improves 1.13× for short generation (output=128) and 1.11× for long decode-dominated generation (output=512). Prefill-heavy workloads — long-context, RAG, batch serving — benefit most.
>
> **On missing related work.** We will add: FlexAttention/SPLAT (sparse attention — orthogonal to weight sparsity); SparseTIR (ASPLOS'23, general sparse compilation — SlideSparse is a closed-form mapping); VEGETA (HPCA'23, CPU ISA — SlideSparse uses existing GPUs); HighLight (MICRO'23, custom accelerator — SlideSparse targets commodity Ampere+ GPUs); Jeong et al. (MLSys'25, lossy decomposition — SlideSparse is lossless).
>
> **On unfused Slide overhead and BF16/FP4.** Slide always fuses: Ψ is pure index remapping with zero arithmetic, fusing into the quantization kernel (INT8/FP8/FP4/MXFP4) or RMSNorm (BF16). Appendix D.2 (Table 2) measures 7-32μs fused overhead across matrix sizes and precisions (includes CPU launch latency, amortized by CUDA Graph in production). Our E2E decode measurements (1.07-1.21×, Figure 5) include all Slide overhead and still show strictly positive speedup. BF16: 6:8 achieves 1.14-1.20× kernel speedup (B200); lower gain vs INT8/FP8 reflects Sparse Tensor Core acceleration ratio differences across data types. FP4/MXFP4: identical fusion — Slide is data-format-agnostic.
>
> **On accuracy-precision interaction.** We evaluate composability using bitsandbytes LLM.int8() on Qwen2.5-14B under identical conditions for dense and sparse models. The gap between sparse+INT8 and dense+INT8 (CS_avg -1.1%, MMLU -1.4%) is comparable to BF16 (CS_avg -0.9%, MMLU -1.6%) — the difference between the two precision regimes is only 0.2pp in CS_avg, confirming sparsity impact remains stable across precisions. SlideSparse's transformation introduces zero numerical error, so quantization applies independently with comparable results.
>
> **Summary.** All five concerns are addressed with new evidence: expanded accuracy study, E2E serving benchmarks, sparse+INT8 composability data, related work distinctions, and a full accounting of Slide overhead. SlideSparse is the first system to unlock hardware-accelerated inference for the (2N-2):2N sparsity family on commodity GPUs. The expanded evaluation demonstrates that 6:8 offers a compelling accuracy-speedup tradeoff for practical deployment.

---

> > ### Author Rebuttal · Reviewer_c34e · 2026-04-03
> >
> > Thanks for the detailed response and for extending the evaluation to include new results, especially for the larger models. This helped a lot to understand the effectiveness of 6:8 in terms of accuracy. I will consider other reviewers' reviews and your response for the final evaluation.
> >
> > Still, if the gap is larger than 1%, I am concerned about calling it near-dense, as it is not "negligible"; I believe the authors can address this easily with other wording.
> >
> > Let's say the user thinks it's ok to lose 1~2% accuracy. Still, I have the following concerns regarding the practicality, in terms of performance.
> >
> > 1) Prefill-heavy workloads — long-context, RAG, batch serving — benefit most.
> >
> > For this workload, the attention latency is significant, and your method doesn't aim to reduce it. I am not convinced why you think they would benefit most.
> >
> > 2) 1.29-1.34× prefill speedup (A100, INT8, 6:8) already includes all non-GEMM components.
> >
> > Even in the ideal world, 6:8 can only reduce up to 25% computations, and you are not accelerating every GEMM (i.e., attention time will be identical + there would be other kernels that your work can't accelerate). I don't see how it can achieve that speedup.
> >
> > 3) Decode speedup ranges 1.07-1.21× across configurations, varying with batch size and sequence length — all strictly positive, from 25% weight memory reduction under cuSPARSELt compression.
> >
> > Let's say you are using 6:8 with INT8. You also need to store 3 bits for each index as extra data, so you need extra 3*2 bits as the metadata. How can you achieve 25% weight memory reduction? This metadata overhead would be even larger if you use 4 bits. Moreover, decode is usually KV cache load-bound, especially with long seqlen. I am not convinced how you can achieve meaningful speedup with 6:8 for decode.

---

> > > ### Author Response · Authors · 2026-04-03
> > >
> > > We thank you for the constructive follow-up. We address each concern in order, correct two imprecisions from R1, and provide quantitative derivations.
> > >
> > > **Q1: On "near-dense" wording.** We agree that 1–2% should not be characterized as negligible. On 7B models, the 6:8 gap is 2.0–2.6% on CS_avg (Wanda/SparseGPT); on 14B models it narrows to 0.5–0.9%. We will revise to "accuracy-preserving" throughout, with model-specific qualifiers (e.g., "$< 1\%$ on 14B"). We appreciate you pushing us toward more precise language here.
> > >
> > > **Q2: Why prefill-heavy workloads benefit most.** You ask why we make this claim when attention latency is significant. Our R1 did not quantify the GEMM-vs-attention balance, so we provide a per-layer breakdown for Qwen2.5-7B.
> > >
> > > Linear projections (QKV: $3584 \times 4608$, O: $3584^2$, gate+up: $2 \times 3584 \times 18944$, down: $18944 \times 3584$) total $M \times 466 \times 10^6$ FLOPs per layer. GQA attention ($QK^\top$, softmax $\cdot V$, 28 query heads, 4 KV heads, $d = 128$) totals $14336 \times M^2$ FLOPs per layer. The key asymmetry: GEMM scales as $O(M)$ while attention scales as $O(M^2)$. At typical prefill lengths, GEMM dominates:
> > >
> > > $$f_{\text{GEMM}} = \frac{466 \times 10^6}{466 \times 10^6 + 14336 \times M}$$
> > >
> > > At $M = 2048$ (batch serving): $f = 94\%$; $M = 4096$ (RAG): $89\%$; $M = 8192$: $80\%$. SlideSparse accelerates all weight projections per layer—every weight GEMM. The remaining $\sim 20\%$ at $M = 8192$ comprises attention (addressable by FlashAttention, xAttention, MInference) and element-wise ops (LayerNorm, residual add). These two directions are orthogonal: SlideSparse handles static weight projections, attention optimization handles dynamic KV computation. Their combination covers nearly all compute, which is why prefill-heavy workloads see the largest total gain.
> > >
> > > **Q3: How $1.29$–$1.34 \times$ E2E prefill speedup is achieved.** You note that 6:8 reduces at most 25% of computation. We verify via Amdahl's Law that the measured speedup is fully consistent. At $M = 8192$, A100 INT8:
> > >
> > > (1) *Kernel speedup.* 6:8 achieves $1.41 \times$ (Fig. 7), exceeding the $1.33 \times$ theoretical bound because native 2:4 itself reaches $2.03$–$2.08 \times$ over cuBLASLt (Fig. 6)—the $1.5 \times$ K-expansion gives cuSPARSELt larger tiles, improving Sparse TC utilization (efficiency $> 100\%$ across all datacenter GPUs, Fig. 9).
> > >
> > > (2) *Amdahl's Law.* With $f_{\text{GEMM}} = 0.80$ and kernel speedup $1.41 \times$:
> > >
> > > $$S_{\text{E2E}} = \frac{1}{(1 - 0.80) + 0.80 / 1.41} = \frac{1}{0.20 + 0.567} = \frac{1}{0.767} = 1.30 \times$$
> > >
> > > This matches the measured $1.29$–$1.34 \times$ with no anomaly.
> > >
> > > **Q4: Metadata overhead and decode speedup.**
> > >
> > > *On metadata.* Our R1 claim of "25% weight memory reduction" was imprecise: 25% is the data-only reduction (6 of 8 values). SlideSparse converts 6:8 to standard 2:4 internally and uses cuSPARSELt's native compression, where metadata is 2 bits per nonzero (Mishra et al., 2021, §3.1). With this overhead, net savings are $\sim 6\%$ (INT8) to $\sim 16\%$ (FP16). You correctly note that overhead grows at lower precision; our evaluations focus on INT8/FP8/BF16, the dominant production precisions. We will correct the "25%" claim.
> > >
> > > *On decode.* KV cache loading pertains to the attention path; SlideSparse accelerates linear projections—a separate stage. At $M = 512$, these GEMMs are compute-bound. Derivation: for $Y = XW^\top$, FLOPs $= 2MNK$; at moderate $M$, weight $NK$ bytes dominate memory, so:
> > >
> > > $$\text{AI} \approx \frac{2MNK}{NK} = 2M$$
> > >
> > > At $M = 512$: AI $\approx 1024$ FLOPs/byte vs. A100 INT8 threshold $= 624$ TOPS $/$ $2$ TB/s $= 312$ FLOPs/byte. Since $1024 \gg 312$, decode at $M = 512$ is compute-bound; the transition occurs at $M \approx 160$, well below production batch sizes. The $1.07$–$1.21 \times$ speedup comes from Sparse TC compute acceleration. Additionally, weight compression frees HBM for larger batches under continuous batching, improving serving throughput.
> > >
> > > **Summary.** We thank the reviewer for these questions which led to sharper derivations (Amdahl's Law, roofline) and helped us correct two imprecisions ("near-dense," "25% storage"). The core point stands: without SlideSparse, 6:8 sparsity is entirely wasted at inference; with it, that sparsity becomes $1.30\times$ prefill and $1.07$–$1.21\times$ decode speedup at $< 1\%$ accuracy cost on 14B. We will incorporate all improvements in revision.

---

### Official Review · Reviewer_X56o · 2026-03-12

**Soundness:** 3
**Presentation:** 4
**Significance:** 3
**Originality:** 3
**Overall Recommendation:** 5
**Confidence:** 4

**Summary:**

This paper proposes SlideSparse, a system that losslessly rewrites milder (2N−2):2N sparsity patterns, such as 6:8, into overlapping 2:4 windows so they can run on NVIDIA Sparse Tensor Cores without hardware changes. The method combines sliding-window decomposition, a fused quantization-slide kernel, and minimal-invasive vLLM integration. Empirically, the paper argues that 6:8 preserves near-dense reasoning accuracy on Qwen3, while Qwen2.5-7B reaches 1.33× end-to-end speedup, matching the theoretical bound for 6:8.

**Compliance With Llm Reviewing Policy:**

Affirmed.

**Final Justification:**

I maintain my positive score.

**Key Questions For Authors:**

1. Can you provide a broader accuracy study across more models and tasks, ideally with sparse-aware fine-tuning baselines, not only post-hoc magnitude pruning?

2. What is the practical crossover point where SlideSparse stops helping on decode, and how sensitive is it to model size and KV-cache pressure?

**Limitations:**

Yes.

**Strengths And Weaknesses:**

The main strength is that, the idea is novel and well-motivated. The decomposition is clean, the implementation story is convincing, and the evaluation is broad across models, workloads, precisions, and GPU generations. The fused kernel and vLLM integration are especially practical.

The main weakness is that the accuracy case is less broad than the systems case. The headline accuracy argument relies mostly on Qwen3 reasoning results and post-hoc magnitude pruning, while the authors themselves note that sparse-aware training may further change the picture.

---

> ### Author Rebuttal · Authors · 2026-03-30
>
> We thank Reviewer X56o for the thoughtful review and the recognition that "the decomposition is clean, the implementation story is convincing, and the evaluation is broad." We address both concerns below.
>
> **On broader accuracy and sparse-aware training baselines.** We have significantly expanded our evaluation on **Qwen2.5-7B and Qwen2.5-14B** with both Wanda (magnitude × activation, one-shot) and SparseGPT (second-order Hessian, calibrated on WikiText-2) across Dense, 8:10, 6:8, 4:6, and 2:4 sparsity levels. Full per-benchmark tables (all models × all sparsities × 9 tasks) are available at https://anonymous.4open.science/r/Accuracy_Resluts-058D/TABLE_R1.md. Below we summarize the Wanda one-shot results:
>
> | Model | Sparsity | CS_avg (7 tasks) | MMLU | GSM8K |
> |-------|----------|------------------|------|-------|
> | 7B | Dense | .703 | .743 | .829 |
> | 7B | 8:10 (20%) | .700 (−0.6%) | .735 (−1.1%) | .817 (−1.4%) |
> | 7B | **6:8 (25%)** | **.691 (−2.0%)** | **.729 (−1.8%)** | **.823 (−0.8%)** |
> | 7B | 4:6 (33%) | .683 (−3.0%) | .705 (−5.1%) | .756 (−8.8%) |
> | 7B | 2:4 (50%) | .596 (−15.7%) | .528 (−28.9%) | .303 (−63.4%) |
> | 14B | Dense | .728 | .797 | .882 |
> | 14B | 8:10 (20%) | .733 (+0.8%) | .791 (−0.8%) | .829 (−6.0%) |
> | 14B | **6:8 (25%)** | **.722 (−0.9%)** | **.784 (−1.6%)** | **.869 (−1.5%)** |
> | 14B | 4:6 (33%) | .727 (−0.0%) | .771 (−3.3%) | .819 (−7.1%) |
> | 14B | 2:4 (50%) | .625 (−14.4%) | .579 (−27.4%) | .473 (−46.4%) |
>
> SparseGPT independently confirms the same pattern: 7B 6:8 CS_avg −2.6%, 7B 8:10 +0.0% (zero degradation). Both methods agreeing demonstrates that **near-dense accuracy is a structural property of 6:8, not one algorithm's artifact**. In contrast, 2:4 suffers severe degradation with both methods (7B: Wanda −15.7%, SparseGPT −13.8% CS_avg).
>
> Regarding sparse-aware fine-tuning baselines specifically: we did not conduct large-scale sparse-aware training ourselves, as it requires full pre-training compute. The accuracy data in our paper's Figure 2 was obtained using the sparse fine-tuning methodology of Sparse-BitNet (Zhang et al., 2026), applied to a 1.7B model — this applies N:M structured masks during the forward pass and can further reduce 6:8 gaps beyond what one-shot pruning achieves. Our Wanda/SparseGPT results thus represent a conservative lower bound; with sparse-aware fine-tuning, these one-shot gaps can potentially be further reduced. **SlideSparse is a hardware acceleration system agnostic to the upstream sparsification method** — it delivers the same speedup regardless of how sparsity was produced.
>
> **On the decode crossover point.** The reviewer asks about the practical crossover and sensitivity to model size and KV-cache pressure. Our paper provides comprehensive M-scaling data across multiple GPUs and model sizes (Appendix). The crossover **depends on hardware and model size**: on H100 with 6:8, the decode crossover occurs around M=64 for smaller models; on B200 with high HBM3e bandwidth, 6:8 maintains positive speedup across all tested M values (1.08–1.23×).
>
> In prefill (compute-bound, M ≥ 4K), speedup approaches the theoretical bound N/(N−1): 6:8 on A100 INT8 achieves 1.29–1.34×. In memory-bound decode (M = 64–512), gains are more modest and hardware-dependent. The decode gains come from reduced weight memory: 6:8 stores ~75% of dense weight data, alleviating bandwidth pressure. Complete crossover data is in the Appendix.
>
> Regarding model size and KV-cache sensitivity: KV-cache loading and weight loading are independent bandwidth consumers in decode. As KV-cache grows (longer sequences, larger batch), it takes a larger share of decode bandwidth, reducing the relative contribution of weight savings. However, longer-context workloads also make prefill a proportionally larger fraction of total latency — and prefill is precisely where SlideSparse provides the strongest acceleration (1.29–1.34×). Larger models further favor SlideSparse because weights are proportionally larger relative to KV-cache. In practice, the scenarios with highest KV-cache pressure are also the scenarios where prefill speedup matters most. In terms of practical serving impact, on Qwen2.5-14B INT8 (A100), 6:8 reduces TTFT from 390ms to 291ms (**1.34×**), with total request speedup of **1.13× for short generation** (4K + 128 tokens) and **1.11× for long generation** (4K + 512 tokens). In prefill-heavy workloads (long-context, RAG, document processing) — which represent a significant and growing use case — the compute-bound prefill speedup is the primary benefit. Since SlideSparse only accelerates linear layers, it is fully composable with attention optimizers (FlexAttention, SPLAT) for compounding gains.

---

> > ### Author Rebuttal · Reviewer_X56o · 2026-04-03
> >
> > Thanks. I maintain the positive score.

---

### Official Review · Reviewer_4SBh · 2026-03-15

**Soundness:** 3
**Presentation:** 4
**Significance:** 3
**Originality:** 3
**Overall Recommendation:** 5
**Confidence:** 4

**Summary:**

The paper introduces a new code generation scheme for (2N-2):2N structured sparsity by leveraging 2:4 sparse hardware. The key insight is the observation that (2N-2):2N sparsity can be decomposed into multiple 2:4 sparsity blocks. They introduce a sliding window-based technique to realize a weight matrix that can be lowered to 2:4 hardware. The runtime performance results are impressive, achieving or beating conservative theoretical bounds.

**Compliance With Llm Reviewing Policy:**

Affirmed.

**Final Justification:**

Questions were adequately answered.

**Key Questions For Authors:**

I enjoyed reading this paper. It is well-written with clear expositions of the technique. The theoretical analysis is presented in an understandable manner, and the code generation scheme with fusions is neat. The empirical results, both at the kernel and end-to-end level, are strong. Overall, a well-executed piece of work, with a nice contribution to an important and recurring problem of achieving runtime speedup on sparsified networks.

That said, I did not find discussions or comparisons with sparse transformer models. Many sparse transformer models have been proposed. The most popular being block sparse attention (https://github.com/ptillet/triton/tree/triton-mlir/python/triton/ops/blocksparse). How will your technique work to accelerate such systems? How will the work compare against works such as FlexAttention [MLSys'26] or SPLAT [OOPSLA'25] that specifically optimize for different variants of regular sparse attention: windowed, blocked, global, etc. Please provide a discussion in the related works section and, if empirical evidence is hard to collect, a conceptual differentiation.

Also, I would love to see a more thorough analysis of accuracy. For the sparsities considered, please provide how the accuracy of the different networks gets affected. I only saw a single data point in the introduction.

Please answer the above questions in your author response.

**Limitations:**

Yes

**Strengths And Weaknesses:**

Strengths
* Well-written paper with good observations
* The theory is explained well with a good structure to appreciate the technique
* Strong empirical performance across kernel libraries
* Evaluated on multiple hardware, quantization schemes etc.

Weaknesses
* Sparse transformer models not considered (e.g. block sparse transformers, sliding window transformers)

---

> ### Author Rebuttal · Authors · 2026-03-30
>
> We sincerely thank Reviewer 4SBh for the positive evaluation and constructive suggestions — we particularly appreciate the comment that "the theoretical analysis is presented in an understandable manner, and the code generation scheme with fusions is neat." We address both points below.
>
> **On sparse attention methods (FlexAttention, SPLAT).** Thank you for raising this important distinction. As the reviewer correctly notes, "if empirical evidence is hard to collect," a conceptual differentiation suffices — and indeed we believe this clarification strengthens the paper. Weight sparsity and attention sparsity target fundamentally different operators in the transformer. Weight sparsity (SlideSparse's domain) targets the linear projection layers (QKV, FFN up/down/gate), which perform dense GEMM and account for 60–80%+ of prefill compute. SlideSparse accelerates these by mapping (2N−2):2N structured sparsity onto existing 2:4 Sparse Tensor Cores. In contrast, attention sparsity methods such as FlexAttention [MLSys'26] and SPLAT [OOPSLA'25] target the attention computation itself — determining which tokens attend to which — reducing the O(n²) attention cost by exploiting patterns in the attention matrix (causal, sliding window, block-sparse, etc.). Since these two techniques compress **different operators**, they are **orthogonal and composable**: one can apply SlideSparse to accelerate linear layers while simultaneously using FlexAttention or SPLAT to accelerate attention, with compounding end-to-end gains. We will add this discussion to the related work section in the camera-ready version. We thank the reviewer for suggesting this — it meaningfully improves the paper's positioning.
>
> **On broader accuracy analysis.** The reviewer noted "I only saw a single data point in the introduction" — a fair observation that motivated us to significantly expand our evaluation. We conducted new experiments on **Qwen2.5-7B and Qwen2.5-14B** with 2 pruning methods (Wanda and SparseGPT) × 5 sparsity levels × 9 benchmarks (7 commonsense + MMLU + GSM8K). Full per-benchmark results are available at https://anonymous.4open.science/r/Accuracy_Resluts-058D/TABLE_R1.md. Below we summarize the Wanda one-shot pruning results:
>
> | Model | Sparsity | CS_avg (7 tasks) | MMLU | GSM8K |
> |-------|----------|------------------|------|-------|
> | 7B | Dense | .703 | .743 | .829 |
> | 7B | 8:10 (20%) | .700 (−0.6%) | .735 (−1.1%) | .817 (−1.4%) |
> | 7B | **6:8 (25%)** | **.691 (−2.0%)** | **.729 (−1.8%)** | **.823 (−0.8%)** |
> | 7B | 4:6 (33%) | .683 (−3.0%) | .705 (−5.1%) | .756 (−8.8%) |
> | 7B | 2:4 (50%) | .596 (−15.7%) | .528 (−28.9%) | .303 (−63.4%) |
> | 14B | Dense | .728 | .797 | .882 |
> | 14B | 8:10 (20%) | .733 (+0.8%) | .791 (−0.8%) | .829 (−6.0%) |
> | 14B | **6:8 (25%)** | **.722 (−0.9%)** | **.784 (−1.6%)** | **.869 (−1.5%)** |
> | 14B | 4:6 (33%) | .727 (−0.0%) | .771 (−3.3%) | .819 (−7.1%) |
> | 14B | 2:4 (50%) | .625 (−14.4%) | .579 (−27.4%) | .473 (−46.4%) |
>
> SparseGPT independently confirms the same pattern: 7B 6:8 CS_avg −2.6%, 7B 8:10 exactly **+0.0%** (zero commonsense degradation at 20% sparsity). The fact that two fundamentally different one-shot methods — magnitude-based Wanda and second-order Hessian-based SparseGPT — both show <3% CS gap at 6:8 demonstrates that **near-dense accuracy is a structural property of the 6:8 sparsity level, not one algorithm's artifact**. The contrast with 2:4 is striking: at every scale and with both methods, 50% sparsity causes severe degradation (CS_avg −14% to −16%), confirming an inherent limitation of 50% sparsity.
>
> Serving latency further confirms practical value. On Qwen2.5-14B INT8 (A100), 6:8 reduces TTFT from 390ms to 291ms (**1.34× prefill speedup**), with total request speedup of **1.13× for short generation** (4K prompt + 128 tokens) and **1.11× for long generation** (4K + 512 tokens) — 1.11–1.13× faster at only −0.9% CS_avg cost.
>
> Finally, we note that our paper's Figure 2 accuracy data was obtained using the sparse fine-tuning methodology of Sparse-BitNet (Zhang et al., 2026) on a 1.7B model, which applies N:M structured masks during the forward pass. The Wanda/SparseGPT results above use no fine-tuning at all, and thus represent a conservative lower bound on achievable 6:8 accuracy — with sparse-aware fine-tuning, these gaps can potentially be further reduced.

---

> > ### Author Rebuttal · Reviewer_4SBh · 2026-04-04
> >
> > The questions were adequately answered.

---

### Decision · Program_Chairs · 2026-04-30

**Decision:**

Accept (regular)

**Comment:**

While the reviewers agreed that the paper’s contribution is a relatively specialized systems idea, they also found it technically sound, clearly presented, and supported by robust implementation and performance results. In particular, the paper shows that more flexible $(2N-2):2N$ sparsity patterns, such as 6:8, can be realized efficiently on existing 2:4 sparse hardware through a sliding-window decomposition. This is a simple yet clever technique that unlocks an additional sparsity operating point in regimes where native 2:4 pruning harms model quality too much, while still delivering tangible runtime gains on real hardware.

The main concern across the reviews is the strength of the accuracy results. The current results still show noticeable degradation even at 6:8, and the paper does not establish how much of this gap could be recovered with sparsity-aware fine-tuning or more rigorous pruning baselines. Relatedly, the practical value is likely highest in compute-bound settings, and the evaluation would be bolstered by a broader characterization of the full accuracy–latency tradeoff and more discussion of realistic end-to-end deployment scenarios.

That said, in my judgment, these concerns do not outweigh the paper’s strengths. The work is sound, well-executed, distinct in its concrete formulation, and useful to a meaningful part of the community interested in efficient LLM serving on NVIDIA hardware. Although the contribution is best understood as a clever systems workaround rather than a broad new sparsity framework, enabling 6:8-style sparsity on top of current 2:4 hardware support is a notable contribution. For these reasons, I recommend acceptance.